# Time-Reversed Dissipation Induces Duality Between Minimizing Gradient Norm and Function Value

**Jaeyeon Kim**
Seoul National University
kjy011102@snu.ac.kr

**Asuman Ozdaglar**
MIT EECS
asuman@mit.edu

**Chanwoo Park**
MIT EECS
cpark97@mit.edu

**Ernest K. Ryu**
Seoul National University
ernestryu@snu.ac.kr

## Abstract

In convex optimization, first-order optimization methods efficiently minimizing function values have been a central subject study since Nesterov's seminal work of 1983. Recently, however, Kim and Fessler's OGM-G and Lee et al.'s FISTA-G have been presented as alternatives that efficiently minimize the gradient magnitude instead. In this paper, we present H-duality, which represents a surprising one-to-one correspondence between methods efficiently minimizing function values and methods efficiently minimizing gradient magnitude. In continuous-time formulations, H-duality corresponds to reversing the time dependence of the dissipation/friction term. To the best of our knowledge, H-duality is different from Lagrange/Fenchel duality and is distinct from any previously known duality or symmetry relations. Using H-duality, we obtain a clearer understanding of the symmetry between Nesterov's method and OGM-G, derive a new class of methods efficiently reducing gradient magnitudes of smooth convex functions, and find a new composite minimization method that is simpler and faster than FISTA-G.

## 1 Introduction

Since Nesterov's seminal work of 1983 [37], accelerated first-order optimization methods that efficiently reduce *function values* have been central to the theory and practice of large-scale optimization and machine learning. In 2012, however, Nesterov initiated the study of first-order methods that efficiently reduce *gradient magnitudes* of convex functions [41]. In convex optimization, making the function value exactly optimal is equivalent to making the gradient exactly zero, but reducing the function-value suboptimality below a threshold is not equivalent to reducing the gradient magnitude below a threshold. This line of research showed that accelerated methods for reducing function values, such as Nesterov's FGM [37], the more modern OGM [26], and the accelerated composite optimization method FISTA [11] are not optimal for reducing gradient magnitude, and new optimal alternatives, such as OGM-G [29] and FISTA-G [31], were presented.

These new accelerated methods for reducing gradient magnitudes are understood far less than those for minimizing function values. However, an interesting observation of symmetry, described in Section 2, was made between these two types of methods, and it was conjectured that this symmetry might be a key to understanding the acceleration mechanism for efficiently reducing gradient magnitude.

**Contribution.** We present a surprising one-to-one correspondence between methods efficiently minimizing function values and methods efficiently minimizing gradient magnitude. We call this correspondence H-duality and formally establish a duality theory in both discrete- and continuous-time dynamics. Using H-duality, we obtain a clearer understanding of the symmetry between FGM/OGM and OGM-G, derive a new class of methods efficiently reducing gradient magnitudes, and find a new composite minimization method that is simpler and faster than FISTA-G, the prior state-of-the-art in efficiently reducing gradient magnitude in the composite minimization setup.

37th Conference on Neural Information Processing Systems (NeurIPS 2023).

## 1.1 Preliminaries and Notation

Given $f\colon \mathbb{R}^d \to \mathbb{R}$, write $f_\star = \inf_{x\in\mathbb{R}^d} f(x) \in (-\infty, \infty)$ for the minimum value and $x_\star \in \operatorname{argmin}_{x\in\mathbb{R}^n} f(x)$ for a minimizer, if one exists. Throughout this paper, we assume $f_\star \neq -\infty$, but we do not always assume a minimizer $x_\star$ exists. Given a differentiable $f\colon \mathbb{R}^d \to \mathbb{R}$ and a pre-specified value of $L > 0$, we define the notation

$$[x, y] := f(y) - f(x) + \langle \nabla f(y), x - y \rangle$$

$$[\![x, y]\!] := f(y) - f(x) + \langle \nabla f(y), x - y \rangle + \frac{1}{2L} \|\nabla f(x) - \nabla f(y)\|^2$$

$$[\![x, \star]\!] := f_\star - f(x) + \frac{1}{2L} \|\nabla f(x)\|^2$$

for $x, y \in \mathbb{R}^d$. A differentiable function $f\colon \mathbb{R}^d \to \mathbb{R}$ is convex if the convexity inequality $[x, y] \leq 0$ holds for all $x, y \in \mathbb{R}^d$. For $L > 0$, a function $f\colon \mathbb{R}^d \to \mathbb{R}$ is $L$-smooth convex if it is differentiable and the cocoercivity inequality $[\![x, y]\!] \leq 0$ holds for all $x, y \in \mathbb{R}^d$ [42]. If $f$ has a minimizer $x_\star$, then $[\![x, \star]\!] = [\![x, x_\star]\!]$, but the notation $[\![x, \star]\!]$ is well defined even when a minimizer $x_\star$ does not exist. If $f$ is $L$-smooth convex, then $[\![x, \star]\!] \leq 0$ holds for all $x \in \mathbb{R}^d$ [42].

Throughout this paper, we consider the duality between the following two problems.

(P1) Efficiently reduce $f(x_N) - f_\star$ assuming $x_\star$ exists and $\|x_0 - x_\star\| \leq R$.

(P2) Efficiently reduce $\frac{1}{2L}\|\nabla f(y_N)\|^2$ assuming $f_\star > -\infty$ and $f(y_0) - f_\star \leq R$.

Here, $R \in (0, \infty)$ is a parameter, $x_0$ and $y_0$ denote initial points of methods for (P1) and (P2), and $x_N$ and $y_N$ denote outputs of methods for (P1) and (P2).

Finally, the standard gradient descent (GD) with stepsize $h$ is

$$x_{i+1} = x_i - \frac{h}{L}\nabla f(x_i), \quad i = 0, 1, \dots. \tag{GD}$$

## 1.2 Prior works

Classically, the goal of optimization methods is to reduce the function value efficiently. In the smooth convex setup, Nesterov's fast gradient method (FGM) [37] achieves an accelerated $\mathcal{O}(1/N^2)$-rate, and the optimized gradient method (OGM) [26] improves this rate by a factor of 2, which is, in fact, exactly optimal [18].

On the other hand, Nesterov initiated the study of methods for reducing the gradient magnitude of convex functions [41] as such methods help us understand non-convex optimization better and design faster non-convex machine learning methods. For smooth convex functions, (GD) achieves a $\mathcal{O}((f(x_0) - f_\star)/N)$-rate on the squared gradient magnitude [34, Proposition 3.3.1], while (OGM-G) achieves an accelerated $\mathcal{O}((f(x_0) - f_\star)/N^2)$-rate [29], which matches a lower bound and is therefore optimal [35, 36]. Interestingly, (OGM) and (OGM-G) exhibit an interesting hint of symmetry, as we detail in Section 2, and the goal of this work is to derive a more general duality principle from this observation.

In the composite optimization setup, iterative shrinkage-thresholding algorithm (ISTA) [13, 45, 16, 14] achieves a $\mathcal{O}(\|x_0 - x_\star\|^2/N)$-rate on function-value suboptimality, while the fast iterative shrinkage-thresholding algorithm (FISTA) [11] achieves an accelerated $\mathcal{O}(\|x_0 - x_\star\|/N^2)$-rate. On the squared gradient mapping norm, FISTA-G achieves $\mathcal{O}((F(x_0) - F_\star)/N^2)$-rate [31], which is optimal [35, 36]. Analysis of an accelerated method often uses the estimate sequence technique [38, 8, 39, 9, 40, 32] or a Lyapunov analysis [37, 11, 50, 10, 53, 1, 5, 6, 7, 43]. In this work, we focus on the Lyapunov analysis technique, as it is simpler and more amenable to a continuous-time view.

The notion of duality is fundamental in many branches of mathematics, including optimization. Lagrange duality [46, 47, 12], Wolfe duality [57, 15, 49, 33], and Fenchel–Rockacheller duality [20, 46] are related (arguably equivalent) notions that consider a pairing of primal and dual optimization problems. The recent gauge duality [21, 22, 3, 58] and radial duality [24, 23] are alternative notions of duality for optimization problems. Attouch–Théra duality [4, 48] generalizes Fenchel–Rockacheller to the setup of monotone inclusion problems. In this work, we present H-duality, which is a notion of duality for optimization *algorithms*, and it is, to the best of our knowledge, distinct from any previously known duality or symmetry relations.

## 2 H-duality

In this section, we will introduce H-duality, state the main H-duality theorem, and provide applications. Let $N \geq 1$ be a pre-specified iteration count. Let $\{h_{k,i}\}_{0 \leq i < k \leq N}$ be an array of (scalar) stepsizes and identify it with a lower triangular matrix $H \in \mathbb{R}^{N \times N}$ via $H_{k+1,i+1} = h_{k+1,i}$ if $0 \leq i \leq k \leq N-1$ and $H_{k,i} = 0$ otherwise. An $N$-step Fixed Step First Order Method (FSFOM) with $H$ is

$$x_{k+1} = x_k - \frac{1}{L} \sum_{i=0}^{k} h_{k+1,i} \nabla f(x_i), \qquad \forall\, k = 0, \ldots, N-1 \tag{1}$$

for any initial point $x_0 \in \mathbb{R}^d$ and differentiable $f$. For $H \in \mathbb{R}^{N \times N}$, define its *anti-transpose* $H^A \in \mathbb{R}^{N \times N}$ with $H^A_{i,j} = H_{N-j+1,N-i+1}$ for $i, j = 1, \ldots, N$. We call [FSFOM with $H^A$] the **H-dual** of [FSFOM with $H$].

### 2.1 Symmetry between OGM and OGM-G

Let $f$ be an $L$-smooth convex function. Define the notation $z^+ = z - \frac{1}{L} \nabla f(z)$ for $z \in \mathbb{R}^d$. The accelerated methods OGM [19, 26] and OGM-G [29] are

$$x_{k+1} = x_k^+ + \frac{\theta_k - 1}{\theta_{k+1}}(x_k^+ - x_{k-1}^+) + \frac{\theta_k}{\theta_{k+1}}(x_k^+ - x_k) \tag{OGM}$$

$$y_{k+1} = y_k^+ + \frac{(\theta_{N-k}-1)(2\theta_{N-k-1}-1)}{\theta_{N-k}(2\theta_{N-k}-1)}(y_k^+ - y_{k-1}^+) + \frac{2\theta_{N-k-1}-1}{2\theta_{N-k}-1}(y_k^+ - y_k) \tag{OGM-G}$$

for $k = 0, \ldots, N-1$, where $\{\theta_i\}_{i=0}^N$ are defined as $\theta_0 = 1$, $\theta_{i+1}^2 - \theta_{i+1} = \theta_i^2$ for $0 \leq i \leq N-2$, and $\theta_N^2 - \theta_N = 2\theta_{N-1}^2$.[1] (OGM) and (OGM-G) are two representative accelerated methods for the setups (P1) and (P2), respectively. As a surface-level symmetry, the methods both access the $\{\theta_i\}_{i=0}^N$ sequence, but (OGM-G) does so in a reversed ordering [29]. There turns out to be a deeper-level symmetry: (OGM) and (OGM-G) are H-duals of each other, i.e., $H_{\mathrm{OGM}}^A = H_{\mathrm{OGM\text{-}G}}$. The proof structures of (OGM) and (OGM-G) also exhibit symmetry. We can analyze (OGM) with the Lyapunov function

$$\mathcal{U}_k = \frac{L}{2}\|x_0 - x_\star\|^2 + \sum_{i=0}^{k-1} u_i [\![x_i, x_{i+1}]\!] + \sum_{i=0}^{k}(u_i - u_{i-1})[\![x_\star, x_i]\!] \tag{2}$$

for $-1 \leq k \leq N$ with $\{u_i\}_{i=0}^N = (2\theta_0^2, \ldots, 2\theta_{N-1}^2, \theta_N^2)$ and $u_{-1} = 0$. Since $[\![\cdot, \cdot]\!] \leq 0$ and $\{u_i\}_{i=0}^N$ is a positive monotonically increasing sequence, $\{\mathcal{U}_k\}_{k=-1}^N$ is dissipative, i.e., $\mathcal{U}_N \leq \mathcal{U}_{N-1} \leq \cdots \leq \mathcal{U}_0 \leq \mathcal{U}_{-1}$. So

$$\theta_N^2\left(f(x_N) - f_\star\right) \leq \theta_N^2\left(f(x_N) - f_\star\right) + \frac{L}{2}\|x^\star - x_0 + z\|^2 \stackrel{(\bullet)}{=} \mathcal{U}_N \leq \mathcal{U}_{-1} = \frac{L\|x_0 - x^\star\|^2}{2},$$

where $z = \sum_{i=0}^N \frac{u_i - u_{i-1}}{L}\nabla f(x_i)$. The justification of $(\bullet)$ is the main technical challenge of this analysis, and it is provided in Appendix B.2. Dividing both sides by $\theta_N^2$, we conclude the rate

$$f(x_N) - f_\star \leq \frac{1}{\theta_N^2}\frac{L}{2}\|x_0 - x^\star\|^2.$$

Likewise, we can analyze (OGM-G) with the Lyapunov function

$$\mathcal{V}_k = v_0\left(f(y_0) - f_\star + [\![y_N, \star]\!]\right) + \sum_{i=0}^{k-1} v_{i+1}[\![y_i, y_{i+1}]\!] + \sum_{i=0}^{k-1}(v_{i+1} - v_i)[\![y_N, y_i]\!] \tag{3}$$

for $0 \leq k \leq N$ with $\{v_i\}_{i=0}^N = \left(\frac{1}{\theta_N^2}, \frac{1}{2\theta_{N-1}^2}, \ldots, \frac{1}{2\theta_0^2}\right)$. Similarly, $\{\mathcal{V}_k\}_{k=0}^N$ is dissipative, so

$$\frac{1}{2L}\|\nabla f(y_N)\|^2 \stackrel{(\circ)}{=} \mathcal{V}_N \leq \mathcal{V}_0 = \frac{1}{\theta_N^2}\left(f(y_0) - f_\star\right) + \frac{1}{\theta_N^2}[\![y_N, \star]\!] \leq \frac{1}{\theta_N^2}\left(f(y_0) - f_\star\right).$$

---

[1]Throughout this paper, we use the convention of denoting iterates of a given "primal" FSFOM as $x_k$ while denoting the iterates of the H-dual FSFOM as $y_k$.

Again, the justification of $(\circ)$ is the main technical challenge of this analysis, and it is provided in Appendix B.2. The crucial observations are **(i)** $u_i = 1/v_{N-i}$ for $0 \le i \le N$ and **(ii)** the convergence rates share the identical factor $1/\theta_N^2 = 1/u_N = v_0$. Interestingly, a similar symmetry relation holds between method pairs $[(\text{OBL-F}_\flat), (\text{OBL-G}_\flat)][44]$ and $[(\text{GD}), (\text{GD})]$, which we discuss later in Section 2.4.

## 2.2 H-duality theorem

The symmetry observed in Section 2.1 is, in fact, not a coincidence. Suppose we have $N$-step FSFOMs with $H$ and $H^A$. We denote their iterates as $\{x_i\}_{i=0}^N$ and $\{y_i\}_{i=0}^N$. For clarity, note that $\{u_i\}_{i=0}^N$ are free variables and can be appropriately chosen for the convergence rate analysis. For the FSFOM with $H$, define $\{\mathcal{U}_k\}_{k=-1}^N$ with the general form (2) with $u_{-1} = 0$. If $0 = u_{-1} \le u_0 \le u_1 \le \cdots \le u_N$, then $\{\mathcal{U}_k\}_{k=-1}^N$ is monotonically nonincreasing (dissipative). Assume we can show

$$u_N(f(x_N) - f_\star) \le \mathcal{U}_N \quad (\forall\, x_0, x_\star, \nabla f(x_0), \ldots, \nabla f(x_N) \in \mathbb{R}^d). \tag{C1}$$

To clarify, since $\{x_i\}_{i=0}^N$ lies within $\text{span}\{x_0, \nabla f(x_0), \ldots, \nabla f(x_N)\}$, the $\mathcal{U}_N$ depends on $\left(x_0, x_\star, \{\nabla f(x_i)\}_{i=0}^N, \{u_i\}_{i=0}^N, H\right)$. If (C1) holds, the FSFOM with $H$ exhibits the convergence rate

$$u_N(f(x_N) - f_\star) \le \mathcal{U}_N \le \cdots \le \mathcal{U}_{-1} = \frac{L}{2}\|x_0 - x_\star\|^2. \tag{4}$$

For the FSFOM with $H^A$, define $\{\mathcal{V}_k\}_{k=0}^N$ with the general form (3). Also, note that $\{v_i\}_{i=0}^N$ are free variables and can be appropriately chosen for the convergence rate analysis. If $0 \le v_0 \le v_1 \le \cdots \le v_N$, then $\{\mathcal{V}_k\}_{k=0}^N$ is monotonically nonincreasing (dissipative). Assume we can show

$$\frac{1}{2L}\|\nabla f(y_N)\|^2 \le \mathcal{V}_N \quad (\forall\, y_0, \nabla f(y_0), \ldots, \nabla f(y_N) \in \mathbb{R}^d,\ f_\star \in \mathbb{R}). \tag{C2}$$

To clarify, since $\{y_i\}_{i=0}^N$ lies within $\text{span}\{y_0, \nabla f(y_0), \ldots, \nabla f(y_N)\}$, the $\mathcal{V}_N$ depends on $\left(y_0, \{\nabla f(y_i)\}_{i=0}^N, f_\star, \{v_i\}_{i=0}^N, H^A\right)$. If (C2) holds, the FSFOM with $H^A$ exhibits the convergence rate

$$\frac{1}{2L}\|\nabla f(y_N)\|^2 \le \mathcal{V}_N \le \cdots \le \mathcal{V}_0 = v_0\left(f(y_0) - f_\star\right) + v_0[\![y_N, \star]\!] \le v_0\left(f(y_0) - f_\star\right). \tag{5}$$

We now state our main H-duality theorem, which establishes a correspondence between the two types of bounds for the FSFOMs induced by $H$ and $H^A$.

**Theorem 1.** Consider sequences of positive real numbers $\{u_i\}_{i=0}^N$ and $\{v_i\}_{i=0}^N$ related through $v_i = \frac{1}{u_{N-i}}$ for $i = 0, \ldots, N$. Let $H \in \mathbb{R}^{N \times N}$ be lower triangular. Then,

$$\left[\text{(C1) is satisfied with } \{u_i\}_{i=0}^N \text{ and } H\right] \quad \Leftrightarrow \quad \left[\text{(C2) is satisfied with } \{v_i\}_{i=0}^N \text{ and } H^A\right].$$

Theorem 1 provides a sufficient condition that ensures an FSFOM with $H$ with a convergence guarantee on $(f(x_N) - f_\star)$ can be H-dualized to obtain an FSFOM with $H^A$ with a convergence guarantee on $\|\nabla f(y_N)\|^2$. To the best of our knowledge, this is the first result establishing a symmetrical relationship between (P1) and (P2). Section 2.3 provides a proof outline of Theorem 1.

## 2.3 Proof outline of Theorem 1

Define

$$\mathbf{U} := \mathcal{U}_N - u_N(f(x_N) - f_\star) - \frac{L}{2}\left\|x_\star - x_0 + \frac{1}{L}\sum_{i=0}^N (u_i - u_{i-1})\nabla f(x_i)\right\|^2$$

$$\mathbf{V} := \mathcal{V}_N - \frac{1}{2}\|\nabla f(y_N)\|^2.$$

Expanding $\mathbf{U}$ and $\mathbf{V}$ reveals that all function value terms are eliminated and only quadratic terms of $\{\nabla f(x_i)\}_{i=0}^N$ and $\{\nabla f(y_i)\}_{i=0}^N$ remain. Now, (C1) and (C2) are equivalent to the conditions

$$\left[\mathbf{U} \ge 0, \quad \forall\, \left(\nabla f(x_0), \ldots, \nabla f(x_N) \in \mathbb{R}^d\right)\right], \qquad \left[\mathbf{V} \ge 0 \quad \forall\, \left(\nabla f(y_0), \ldots, \nabla f(y_N) \in \mathbb{R}^d\right)\right],$$

respectively. Next, define $g_x = \left[\nabla f(x_0)|\nabla f(x_1)|\ldots|\nabla f(x_N)\right] \in \mathbb{R}^{d\times(N+1)}$ and $g_y = \left[\nabla f(y_0)|\nabla f(y_1)|\ldots|\nabla f(y_N)\right] \in \mathbb{R}^{d\times(N+1)}$. We show that there is $\mathcal{S}(H,u)$ and $\mathcal{T}(H^A,v) \in \mathbb{S}^{N+1}$ such that

$$\mathbf{U} = \text{Tr}\left(g_x \mathcal{S}(H,u) g_x^\mathsf{T}\right), \qquad \mathbf{V} = \text{Tr}\left(g_y \mathcal{T}(H^A,v) g_y^\mathsf{T}\right).$$

Next, we find an explicit invertible matrix $M(u) \in \mathbb{R}^{(N+1)\times(N+1)}$ such that $\mathcal{S}(H,u) = \mathcal{M}(u)^\mathsf{T}\mathcal{T}(H^A,v)\mathcal{M}(u)$. Therefore,

$$\text{Tr}\left(g_x \mathcal{S}(H,u) g_x^\mathsf{T}\right) = \text{Tr}\left(g_y \mathcal{T}(H^A,v) g_y^\mathsf{T}\right)$$

with $g_y = g_x \mathcal{M}(u)^\mathsf{T}$ and we conclude the proof. This technique of considering the quadratic forms of Lyapunov functions as a trace of matrices is inspired by the ideas from the Performance Estimation Problem (PEP) literature [19, 55]. The full proof is given in Appendix A.

## 2.4 Verifying conditions for H-duality theorem

In this section, we illustrate how to verify conditions (C1) and (C2) through examples. Detailed calculations are deferred to Appendices B.1 and B.2.

**Example 1.** For (OGM) and (OGM-G), the choice

$$\{u_i\}_{i=0}^N = (2\theta_0^2,\ldots,2\theta_{N-1}^2,\theta_N^2), \qquad \{v_i\}_{i=0}^N = \left(\frac{1}{\theta_N^2},\frac{1}{2\theta_{N-1}^2},\ldots,\frac{1}{2\theta_0^2}\right)$$

leads to

$$\mathbf{U} = 0, \qquad \mathbf{V} = 0.$$

Therefore, (C1) and (C2) hold.

**Example 2.** Again, define $z^+ = z - \frac{1}{L}\nabla f(z)$ for $z \in \mathbb{R}^d$. Consider the FSFSOMs [44]

$$
\begin{aligned}
x_{k+1} &= x_k^+ + \frac{k}{k+3}\left(x_k^+ - x_{k-1}^+\right) + \frac{k}{k+3}\left(x_k^+ - x_k\right) \quad k = 0,\ldots,N-2 \\
x_N &= x_{N-1}^+ + \frac{N-1}{2(\gamma+1)}\left(x_{N-1}^+ - x_{N-2}^+\right) + \frac{N-1}{2(\gamma+1)}\left(x_{N-1}^+ - x_{N-1}\right)
\end{aligned}
\tag{OBL-F$_\flat$}
$$

and

$$
\begin{aligned}
y_1 &= y_0^+ + \frac{N-1}{2(\gamma+1)}\left(y_0^+ - y_{-1}^+\right) + \frac{N-1}{2(\gamma+1)}\left(y_0^+ - y_0\right) \\
y_{k+1} &= y_k^+ + \frac{N-k-1}{N-k+2}\left(y_k^+ - y_{k-1}^+\right) + \frac{N-k-1}{N-k+2}\left(y_k^+ - y_k\right) \quad k = 1,\ldots,N-1
\end{aligned}
\tag{OBL-G$_\flat$}
$$

where $y_{-1}^+ = y_0$, $x_{-1}^+ = x_0$ and $\gamma = \sqrt{N(N+1)/2}$. It turns out that (OBL-F$_\flat$) and (OBL-G$_\flat$) are H-duals of each other. The choice

$$\{u_i\}_{i=0}^N = \left(\frac{1\cdot 2}{2},\ldots,\frac{N(N+1)}{2},\gamma^2+\gamma\right), \quad \{v_i\}_{i=0}^N = \left(\frac{1}{\gamma^2+\gamma},\frac{2}{N(N+1)},\ldots,\frac{2}{1\cdot 2}\right)$$

leads to

$$\mathbf{U} = \sum_{i=0}^N \frac{u_i - u_{i-1}}{2L}\|\nabla f(x_i)\|^2, \quad \mathbf{V} = \frac{v_0}{2L}\|\nabla f(y_N)\|^2 + \sum_{i=0}^{N-1}\frac{v_{i+1}-v_i}{2L}\|\nabla f(y_i) - \nabla f(y_N)\|^2$$

where $u_{-1} = 0$. Since $\mathbf{U}$ and $\mathbf{V}$ are expressed as a sum of squares, (C1) and (C2) hold.

**Example 3.** Interestingly, (GD) is a self-dual FSFOM in the H-dual sense. For the case $h = 1$, the choice

$$\{u_i\}_{i=0}^N = \left(\ldots,\frac{(2N+1)(i+1)}{2N-i},\ldots,2N+1\right), \qquad \{v_i\}_{i=0}^N = \left(\frac{1}{2N+1},\ldots,\frac{N+i}{(2N+1)(N-i+1)},\ldots\right)$$

leads to

$$\mathbf{U} = \sum_{0\le i,j\le N}\frac{s_{ij}}{L}\langle\nabla f(x_i),\nabla f(x_j)\rangle, \qquad \mathbf{V} = \sum_{0\le i,j\le N}\frac{t_{ij}}{L}\langle\nabla f(y_i),\nabla f(y_j)\rangle$$

for some $\{s_{ij}\}$ and $\{t_{ij}\}$ stated precisely in Appendix B.2. $\mathbf{V} \ge 0$ can be established by showing that the $\{t_{ij}\}$ forms a diagonally dominant and hence positive semidefinite matrix [29]. $\mathbf{U} \ge 0$ can be established with a more elaborate argument [19], but that is not necessary; $\mathbf{V} \ge 0$ implies (C2), and, by Theorem 1, this implies (C1).

## 2.5 Applications of the H-duality theorem

**A family of gradient reduction methods.** Parameterized families of accelerated FSFOMs for reducing function values have been presented throughout the extensive prior literature. Such families generalize Nesterov's method and elucidate the essential algorithmic component that enables acceleration. For reducing gradient magnitude, however, there are only four accelerated FSFOMs (OGM-G), (OBL-G$_\flat$), and M-OGM-G [60], and [17, Lemma 2.6]. Here, we construct a simple parameterized family of accelerated FSFOMs for reducing gradient magnitude by H-dualizing an accelerated FSFOM family for reducing function values.

Let $\{t_i\}_{i=0}^N$ and $\{T_i\}_{i=0}^N$ be sequences positive real numbers satisfying $t_i^2 \leq 2T_i = 2\sum_{j=0}^i t_j$ for $0 \leq i \leq N-1$ and $t_N^2 \leq T_N = \sum_{j=0}^N t_j$. Consider a family of FSFOMs

$$x_{k+1} = x_k^+ + \frac{(T_k - t_k)t_{k+1}}{t_k T_{k+1}}\left(x_k^+ - x_{k-1}^+\right) + \frac{(t_k^2 - T_k)t_{k+1}}{t_k T_{k+1}}\left(x_k^+ - x_k\right) \tag{6}$$

for $k = 0, 1, \ldots, N-1$, where $x_{-1}^+ = x_0$. This family coincides with the GOGM of [28], and it exhibits the rate [28, Theorem 5]

$$f(x_N) - f_\star \leq \frac{1}{T_N}\frac{L}{2}\|x_0 - x_\star\|^2,$$

which can be established from (2) with $u_i = T_i$ for $0 \leq i \leq N$.

**Corollary 1.** The H-dual of (6) is

$$y_{k+1} = y_k^+ + \frac{T_{N-k-1}(t_{N-k-1} - 1)}{T_{N-k}(t_{N-k} - 1)}\left(y_k^+ - y_{k-1}^+\right) + \frac{(t_{N-k}^2 - T_{N-k})(t_{N-k-1} - 1)}{T_{N-k}(t_{N-k} - 1)}\left(y_k^+ - y_k\right)$$

for $k = 0, \ldots, N-1$, where $y_{-1}^+ = y_0$, and it exhibits the rate

$$\frac{1}{2L}\|\nabla f(y_N)\|^2 \leq \frac{1}{T_N}\left(f(y_0) - f_\star\right).$$

*Proof outline.* By Theorem 1, (C2) holds with $v_i = 1/T_{N-i}$ for $0 \leq i \leq N$. We then use (5). □

When $T_i = t_i^2$ for $0 \leq i \leq N$, the FSFOM (6) reduces to Nestrov's FGM [37] and its H-dual is, to the best of our knowledge, a new method without a name. If $t_i^2 = 2T_i$ for $0 \leq i \leq N-1$ and $t_N^2 = T_N$, (6) reduces to (OGM) and its H-dual is (OGM-G). If $t_i = i + 1$ for $0 \leq i \leq N-1$ and $t_N = \sqrt{N(N+1)/2}$, (6) reduces to (OBL-F$_\flat$) and its H-dual is (OBL-G$_\flat$).

**Gradient magnitude rate of** (GD). For gradient descent (GD) with stepsize $h$, the $H$ matrix is the identity matrix scaled by $h$, and the H-dual is (GD) itself, i.e., (GD) is self-dual. For $0 < h \leq 1$, the rate $f(x_N) - f_\star \leq \frac{1}{2Nh+1}\frac{L}{2}\|x_0 - x_\star\|^2$, originally due to [19], can be established from (2) with $\{u_i\}_{i=0}^N = \left(\ldots, \frac{(2Nh+1)(i+1)}{2N-i}, \ldots, 2Nh+1\right)$. Applying Theorem 1 leads to the following.

**Corollary 2.** Consider (GD) with $0 < h \leq 1$ applied to an $L$-smooth convex $f$. For $N \geq 1$,

$$\frac{1}{2L}\|\nabla f(x_N)\|^2 \leq \min\left(\frac{f(x_0) - f_\star}{2Nh+1}, \frac{L\|x_0 - x_\star\|^2}{2(2\lfloor\frac{N}{2}\rfloor h+1)(2\lceil\frac{N}{2}\rceil h+1)}\right).$$

To the best of our knowledge, Corollary 2 is the tightest rate on gradient magnitude for (GD) for the general step size $0 < h < 1$, and it matches [53, Theorem 3] for $h = 1$.

**Resolving conjectures of $\mathcal{A}^\star$-optimality of** (OGM-G) **and** (OBL-F$_\flat$)**.** The prior work of [44] defines the notion of $\mathcal{A}^\star$-optimality, a certain restricted sense of optimality of FSFOMs, and shows that (OGM) and (OBL-F$_\flat$) are $\mathcal{A}^\star$-optimal under a certain set of relaxed inequalities. On the other hand, $\mathcal{A}^\star$-optimality of (OGM-G) and (OBL-G$_\flat$) are presented as conjectures. Combining Theorem 1 and the $\mathcal{A}^\star$-optimality of (OGM) and (OBL-F$_\flat$) resolves these conjectures; (OGM-G) and (OBL-G$_\flat$) are $\mathcal{A}^\star$-optimal.

## 2.6 Intuition behind energy functions: Lagrangian Formulation

Ome might ask where the energy functions (2) and (3) came from. In this section, we provide an intuitive explanation of these energy functions using the QCQP and its Lagrangian. Consider an FSFOM (1) given with a lower triangular matrix $H \in \mathbb{R}^{N \times N}$ with function $f$, resulting in the sequence $\{x_i\}_{i=0}^N$. To analyze the convergence rate of the function value, we formulate the following optimization problem:

$$\sup_f \quad f(x_N) - f_\star$$

$$\text{subject to} \quad f \colon \mathbb{R}^n \to \mathbb{R} \text{ is } L\text{-smooth convex}, \quad \|x_0 - x_\star\|^2 \leq R^2.$$

However, this optimization problem is not solvable, as $f$ is a functional variable. To address this, [19, 55] demonstrated its equivalence to a QCQP:

$$\sup_f \quad f(x_N) - f_\star$$

$$\text{subject to} \quad [\![x_i, x_j]\!] \leq 0 \quad [i, j] \in [-1, \dots, N]^2, \quad \|x_0 - x_\star\|^2 \leq R^2$$

where $x_{-1} \colon = x_\star$. To clarify, the optimization variables are $\{\nabla f(x_i), f(x_i)\}_{i=0}^N$, $f_\star$, $x_0$, and $x_\star$ since $\{x_i\}_{i=0}^N$ lies within $\text{span}\{x_0, \nabla f(x_0), \dots, \nabla f(x_N)\}$. We consider a relaxed optimization problem as [26]:

$$\sup_f \quad f(x_N) - f_\star$$

$$\text{subject to} \quad [\![x_i, x_{i+1}]\!] \leq 0, \quad i = 0, 1, \dots, N - 1, \quad [\![x_\star, x_i]\!] \leq 0, \quad i = 0, 1, \dots, N,$$

$$\|x_0 - x_\star\|^2 \leq R^2$$

Now, consider the Lagrangian function and the convex dual.

$$\mathcal{L}_1(f, \{a_i\}_{i=0}^{N-1}, \{b_i\}_{i=0}^N, \alpha) = -f(x_N) + f_\star + \sum_{i=0}^{N-1} a_i [\![x_i, x_{i+1}]\!] + \sum_{i=0}^N b_i [\![x_\star, x_i]\!] + \alpha \|x_0 - x_\star\|^2 - \alpha R^2$$

where $\{a_i\}_{i=0}^{N-1}$, $\{b_i\}_{i=0}^N$, and $\alpha$ are dual variables which are nonnegative. Considering $a_{-1} = 0$ and $a_N = 1$, the infimum of $\mathcal{L}_1$ equals $-\infty$ unless $b_i = a_i - a_{i-1}$ for $0 \leq i \leq N$. Therefore, by introducing $u_N = \frac{L}{2\alpha}$ and $u_i = a_i u_N$ for $0 \leq i \leq N$, the convex dual problem can be simplified as follows:

$$\inf_{\{u_i\}_{i=0}^N} \quad -\frac{LR^2}{2u_N}$$

$$\text{s.t.} \quad \inf_{x_0, x_\star, \{\nabla f(x_i)\}_{i=0}^N} -u_N(f(x_N) - f_\star) + \frac{L}{2}\|x_0 - x_\star\|^2 + \sum_{i=0}^{N-1} u_i [\![x_i, x_{i+1}]\!] + \sum_{i=0}^N (u_i - u_{i-1}) [\![x_\star, x_i]\!] \geq 0$$

$$\{u_i\}_{i=0}^N \text{ are nonnegative and nondecreasing.}$$

If the above two constraints holds for $\{u_i\}_{i=0}^N$, we have $f(x_N) - f_\star \leq \frac{L}{2u_N} R^2$. This understanding motivates the introduction of (2) and (C1). We can perform a similar analysis on the gradient norm minimization problem with the relaxed optimization problem as follows:

$$\sup_f \quad \frac{1}{2L}\|\nabla f(y_N)\|^2$$

$$\text{subject to} \quad [\![y_i, y_{i+1}]\!] \leq 0, \quad i = 0, 1, \dots, N - 1, \quad [\![y_N, y_i]\!] \leq 0, \quad i = 0, 1, \dots, N - 1,$$

$$[\![y_N, \star]\!] \leq 0, \quad f(y_0) - f_\star \leq R.$$

Finally, we note that although (2) and (3) both originate from the relaxed optimization problems, they have been commonly employed to achieve the convergence analysis. The function value convergence rate of OGM[26], FGM[37], G-OGM[28], GD[19], OBL-F$_b$[44] can be proved by using (2) with appropriate $\{u_i\}_{i=0}^N$. The gradient norm convergence rate of OGM-G[29], OBL-G$_b$[44], M-OGM-G[60], and [17, Lemma 2.6] can be proved by using (3) with appropriate $\{v_i\}_{i=0}^N$. We also note that recent works [2, 25] do not employ (2) to achieve the convergence rate, particularly for gradient descent with varying step sizes.

# 3 H-duality in continuous time

We now establish a continuous-time analog of the H-duality theorem. As the continuous-time result and, especially, its proof is much simpler than its discrete-time counterpart, the results of this section serve as a vehicle to convey the key ideas more clearly. Let $T > 0$ be a pre-specified terminal time. Let $H(t, s)$ be an appropriately integrable[2] real-valued kernel with domain $\{(t, s) \,|\, 0 < s < t < T\}$. We define a Continuous-time Fixed Step First Order Method (C-FSFOM) with $H$ as

$$X(0) = x_0, \quad \dot{X}(t) = -\int_0^t H(t, s) \nabla f(X(s)) \, ds, \quad \forall \, t \in (0, T) \tag{7}$$

for any initial point $x_0 \in \mathbb{R}^d$ and differentiable $f$. Note, the Euler discretization of C-FSFOMs (7) corresponds to FSFOMs (1). The notion of C-FSFOMs has been considered previously in [30].

Given a kernel $H(t, s)$, analogously define its *anti-transpose* as $H^A(t, s) = H(T - s, T - t)$. We call [C-FSFOM with $H^A$] the **H-dual** of [C-FSFOM with $H$]. In the special case $H(t, s) = e^{\gamma(s) - \gamma(t)}$ for some function $\gamma(\cdot)$, the C-FSFOMs with $H$ and its H-dual have the form

$$\ddot{X}(t) + \dot{\gamma}(t)\dot{X}(t) + \nabla f(X(t)) = 0 \qquad \text{(C-FSFOM with } H(t, s) = e^{\gamma(s) - \gamma(t)})$$
$$\ddot{Y}(t) + \dot{\gamma}(T - t)\dot{Y}(t) + \nabla f(Y(t)) = 0 \qquad \text{(C-FSFOM with } H^A(t, s))$$

Interestingly, friction terms with $\gamma'$ have time-reversed dependence between the H-duals, and this is why we refer to this phenomenon as time-reversed dissipation.

## 3.1 Continuous-time H-duality theorem

For the C-FSFOM with $H$, define the energy function

$$\mathcal{U}(t) = \frac{1}{2}\|X(0) - x_\star\|^2 + \int_0^t u'(s)[x_\star, X(s)]ds \tag{8}$$

for $t \in [0, T]$ with differentiable $u \colon (0, T) \to \mathbb{R}$. If $u'(\cdot) \geq 0$, then $\{\mathcal{U}(t)\}_{t \in [0,T]}$ is dissipative. Assume we can show

$$u(T)\,(f(X(T)) - f_\star) \leq \mathcal{U}(T) \quad (\forall\, X(0), x_\star, \{\nabla f(X(s))\}_{s \in [0,T]} \in \mathbb{R}^d). \tag{C3}$$

Then, the C-FSFOM with $H$ exhibits the convergence rate

$$u(T)\,(f(X(T)) - f_\star) \leq \mathcal{U}(T) \leq \mathcal{U}(0) = \frac{1}{2}\|X(0) - x_\star\|^2.$$

For the C-FSFOM with $H^A$, define the energy function

$$\mathcal{V}(t) = v(0)\big(f(Y(0)) - f(Y(T))\big) + \int_0^t v'(s)[Y(T), Y(s)]ds \tag{9}$$

for $t \in [0, T]$ with differentiable $v \colon (0, T) \to \mathbb{R}$. If $v'(\cdot) \geq 0$, then $\{\mathcal{V}(t)\}_{t \in [0,T]}$ is dissipative. Assume we can show

$$\frac{1}{2}\|\nabla f(Y(T))\|^2 \leq \mathcal{V}(T) \quad (\forall\, Y(0), \{\nabla f(Y(s))\}_{s \in [0,T]} \in \mathbb{R}^d). \tag{C4}$$

Then, the C-FSFOM with $H^A$ exhibits the convergence rate

$$\frac{1}{2}\|\nabla f(Y(T))\|^2 \leq \mathcal{V}(T) \leq \mathcal{V}(0) = v(0)\,(f(Y(0)) - f(Y(T))) \leq v(0)\,(f(Y(0)) - f_\star).$$

**Theorem 2** (informal)**.** Consider differentiable functions $u, v \colon (0, T) \to \mathbb{R}$ related through $v(t) = \frac{1}{u(T-t)}$ for $t \in [0, T]$. Assume certain regularity conditions (specified in Appendix C.2). Then,

$$[\text{(C3) is satisfied with } u(\cdot) \text{ and } H] \quad \Leftrightarrow \quad \left[\text{(C4) is satisfied with } v(\cdot) \text{ and } H^A\right].$$

The formal statement of 2 and its proof are given in Appendix C.2. Loosely speaking, we can consider Theorem 2 as the limit of Theorem 1 with $N \to \infty$.

---

[2]In this paper, we avoid analytical and measure-theoretic details and focus on convergence (rather than existence) results of the continuous-time dynamics.

## 3.2 Verifying conditions for H-duality theorem

As an illustrative example, consider the case $H(t,s) = \frac{s^r}{t^r}$ for $r \geq 3$ which corresponds to an ODE studied in the prior work [50, 51]. For the C-FSFOM with $H$, the choice $u(t) = \frac{t^2}{2(r-1)}$ for the dissipative energy function $\{\mathcal{U}(t)\}_{t=0}^{T}$ of (8) leads to

$$\mathcal{U}(T) - u(T)\left(f(X(T)) - f_\star\right) = \frac{\left\|T\dot{X}(T) + 2(X(T) - x_\star)\right\|^2 + 2(r-3)\|X(T) - x_\star\|^2}{4(r-1)} + \int_0^T \frac{(r-3)s}{2(r-1)}\left\|\dot{X}(s)\right\|^2 ds.$$

For the C-FSFOM with $H^A$, the choice $v(t) = \frac{1}{u(T-t)} = \frac{2(r-1)}{(T-t)^2}$ for the dissipative energy function $\{\mathcal{V}(t)\}_{t=0}^{T}$ of (9) leads to

$$\mathcal{V}(T) - \frac{1}{2}\|\nabla f(Y(T))\|^2 = \frac{2(r-1)(r-3)\|Y(0) - Y(T)\|^2}{T^4} + \int_0^T \frac{2(r-1)(r-3)\left\|(T-s)\dot{Y}(s) + 2(Y(s) - Y(T))\right\|^2}{(T-s)^5} ds.$$

Since the right-hand sides are expressed as sums/integrals of squares, they are nonnegative, so (C3) and (C4) hold. (By Theorem 2, verifying (C3) implies (C4) and vice versa.) The detailed calculations are provided in Appendix C.1.

## 3.3 Applications of continuous-time H-duality theorem

The C-FSFOM (7) with $H(t,s) = \frac{Cp^2 s^{2p-1}}{t^{p+1}}$ recovers

$$\ddot{X}(t) + \frac{p+1}{t}\dot{X}(t) + Cp^2 t^{p-2}\nabla f(X(t)) = 0,$$

an ODE considered in [56]. The rate $f(X(T)) - f_\star \leq \frac{1}{2CT^p}\|X(0) - x_\star\|^2$ can be established from (8) with $u(t) = Ct^p$. The C-FSFOM with $H^A$ can be expressed as the ODE

$$\ddot{Y}(t) + \frac{2p-1}{T-t}\dot{Y}(t) + Cp^2(T-t)^{p-2}\nabla f(Y(t)) = 0. \tag{10}$$

By Theorem 2, using (9) with $v(t) = \frac{1}{C(T-t)^p}$ leads to the rate

$$\frac{1}{2}\|\nabla f(Y(T))\|^2 \leq \frac{1}{CT^p}\left(f(Y(0)) - f_\star\right).$$

Note that the continuous-time models of (OGM) and (OGM-G), considered in [51], are special cases of this setup with $p = 2$ and $C = 1/2$. The detailed derivation and well-definedness of the ODE are presented in Appendix C.3.

# 4 New method efficiently reducing gradient mapping norm: (SFG)

In this section, we introduce a novel algorithm obtained using the insights of Theorem 1. Consider minimizing $F(x) := f(x) + g(x)$, where $f: \mathbb{R}^d \to \mathbb{R}$ is $L$-smooth convex with $0 < L < \infty$ and $g: \mathbb{R}^d \to \mathbb{R} \cup \{\infty\}$ is a closed convex proper function. Write $F_\star = \inf_{x \in \mathbb{R}^n} F(x)$ for the minimum value. For $\alpha > 0$, define the $\alpha$-*proximal gradient step* as

$$y^{\oplus,\alpha} = \underset{z \in \mathbb{R}^n}{\operatorname{argmin}}\left(f(y) + \langle\nabla f(y), z - y\rangle + g(z) + \frac{\alpha L}{2}\|z - y\|^2\right) = \operatorname{Prox}_{\frac{g}{\alpha L}}\left(y - \frac{1}{\alpha L}\nabla f(y)\right).$$

Consider FSFOMs defined by a lower triangular matrix $H = \{h_{k,i}\}_{0 \leq i < k \leq N}$ as follows:

$$x_{k+1} = x_k - \sum_{i=0}^{k}\alpha h_{k+1,i}\left(x_i - x_i^{\oplus,\alpha}\right), \quad \forall k = 0, \ldots, N-1.$$

When $g = 0$, this reduces to (1). FISTA [11], FISTA-G [31] and GFPGM [27] are instances of this FSFOM with $\alpha = 1$. In this section, we present a new method for efficiently reducing the gradient mapping norm. This method is faster than the prior state-of-the-art FISTA-G [31] by a constant factor of $5.28$ while having substantially simpler coefficients.

**Theorem 3.** Consider the method

$$y_{k+1} = y_k^{\oplus,4} + \frac{(N-k+1)(2N-2k-1)}{(N-k+3)(2N-2k+1)} \left( y_k^{\oplus,4} - y_{k-1}^{\oplus,4} \right) + \frac{(4N-4k-1)(2N-2k-1)}{6(N-k+3)(2N-2k+1)} \left( y_k^{\oplus,4} - y_k \right)$$

$$y_N = y_{N-1}^{\oplus,4} + \frac{3}{10} \left( y_{N-1}^{\oplus,4} - y_{N-2}^{\oplus,4} \right) + \frac{3}{40} \left( y_{N-1}^{\oplus,4} - y_{N-1} \right) \qquad \text{(SFG)}$$

for $k = 0, \ldots, N-2$, where $y_{-1}^{\oplus,4} = y_0$. This method exhibits the rate

$$\min_{v \in \partial F(y_N^{\oplus,4})} \|v\|^2 \le 25L^2 \left\| y_N - y_N^{\oplus,4} \right\|^2 \le \frac{50L}{(N+2)(N+3)} \left( F(y_0) - F_\star \right).$$

We call this method *Super FISTA-G* (SFG), and in Appendix D.3, we present a further general parameterized family (SFG-family). To derive (SFG-family), we start with the parameterized family GFPGM [27], which exhibits an accelerated rate on function values, and expresses it as FSFOMs with $H$. We then obtain the FSFOMs with $H^A + C$, where $C$ is a lower triangular matrix satisfying certain constraints. We find that the appropriate H-dual for the composite setup is given by this $H^A + C$, rather than $H^A$. We provide the proof of Theorem 3 in Appendix D.2.

(SFG) is an instance of (SFG-family) with simple rational coefficients. Among the family, the optimal choice has complicated coefficients, but its rate has a leading coefficient of $46$, which is slightly smaller than the $50$ of (SFG). We provide the details Appendix D.4.

## 5    Conclusion

In this work, we defined the notion of H-duality and formally established that the H-dual of an optimization method designed to efficiently reduce function values is another method that efficiently reduces gradient magnitude. For optimization algorithms, the notion of equivalence, whether informal or formal [59], is intuitive and standard. For optimization problems, the notion of equivalence is also standard, but the beauty of convex optimization is arguably derived from the elegant duality of optimization problems. In fact, there are many notions of duality for spaces, problems, operators, functions, sets, etc. However, the notion of duality for algorithms is something we, the authors, are unfamiliar with within the context of optimization, applied mathematics, and computer science. In our view, the significance of this work is establishing the first instance of a duality of algorithms. The idea that an optimization algorithm is an abstract mathematical object that we can take the dual of opens the door to many interesting questions. In particular, exploring for what type of algorithms the H-dual or a similar notion of duality makes sense is an interesting direction for future work.

## Acknowledgments and Disclosure of Funding

We thank Soonwon Choi, G. Bruno De Luca, and Eva Silverstein for sharing their insights as physicists. We thank Jaewook J. Suh for reviewing the manuscript and providing valuable feedback. We are grateful to Jungbin Kim and Jeongwhan Lee for their valuable discussions at the outset of this work. J.K. and E.K.R. were supported by the National the Samsung Science and Technology Foundation (Project Number SSTF-BA2101-02). C.P. acknowledges support from the Xianhong Wu Fellowship, the Korea Foundation for Advanced Studies, and the Siebel Scholarship. A.O. acknowledges support from the MIT-IBM Watson AI Lab grant 027397-00196, Trustworthy AI grant 029436-00103, and the MIT-DSTA grant 031017-00016. Finally, we also thank anonymous reviewers for giving thoughtful comments.

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

# A    Proof of Theorem 1

**Reformulate** (C1) **and** (C2) **into U and** $V$**.**    In this paragraph, we will show

$$(\text{C1}) \quad \Leftrightarrow \quad \Big[ \mathbf{U} \geq 0, \quad \big( \forall \, \nabla f(x_0), \dots, \nabla f(x_N) \in \mathbb{R}^d \big) \Big]$$

and

$$(\text{C2}) \quad \Leftrightarrow \quad \Big[ \mathbf{V} \geq 0, \quad \big( \forall \, \nabla f(y_0), \dots, \nabla f(y_N) \in \mathbb{R}^d \big) \Big].$$

Recall the definition of **U** and **V**.

$$\mathbf{U} \colon = \mathcal{U}_N - u_N (f(x_N) - f_\star) - \frac{L}{2} \Big\| x_\star - x_0 + \frac{1}{L} \sum_{i=0}^{N} (u_i - u_{i-1}) \nabla f(x_i) \Big\|^2 \tag{11}$$

$$\mathbf{V} \colon = \mathcal{V}_N - \frac{1}{2} \| \nabla f(y_N) \|^2 . \tag{12}$$

First we calculate $\mathcal{U}_N - u_N \left( f(x_N) - f_\star \right)$.

$$\mathcal{U}_N - u_N \left( f(x_N) - f_\star \right)$$

$$= \frac{L}{2} \| x_0 - x_\star \|^2 + \sum_{i=0}^{N} (u_i - u_{i-1}) \left( f(x_i) - f_\star + \langle \nabla f(x_i), x_\star - x_i \rangle + \frac{1}{2L} \| \nabla f(x_i) \|^2 \right)$$

$$+ \sum_{i=0}^{N-1} u_i \left( f(x_{i+1}) - f(x_i) + \langle \nabla f(x_{i+1}), x_i - x_{i+1} \rangle + \frac{1}{2L} \| \nabla f(x_i) - \nabla f(x_{i+1}) \|^2 \right) - u_N \left( f(x_N) - f_\star \right)$$

$$\overset{(\circ)}{=} \frac{L}{2} \| x_0 - x_\star \|^2 + \sum_{i=0}^{N} (u_i - u_{i-1}) \left( \langle \nabla f(x_i), x_\star - x_i \rangle + \frac{1}{2L} \| \nabla f(x_i) \|^2 \right)$$

$$+ \sum_{i=0}^{N-1} u_i \left( \langle \nabla f(x_{i+1}), x_i - x_{i+1} \rangle + \frac{1}{2L} \| \nabla f(x_i) - \nabla f(x_{i+1}) \|^2 \right)$$

$$= \frac{L}{2} \Big\| x_0 - x_\star + \sum_{i=0}^{N} \frac{u_i - u_{i-1}}{L} \nabla f(x_i) \Big\|^2 - \frac{1}{2L} \Big\| \sum_{i=0}^{N} (u_i - u_{i-1}) \nabla f(x_i) \Big\|^2$$

$$+ \sum_{i=0}^{N} (u_i - u_{i-1}) \left( \langle \nabla f(x_i), x_0 - x_i \rangle + \frac{1}{2L} \| \nabla f(x_i) \|^2 \right)$$

$$+ \sum_{i=0}^{N-1} u_i \left( \langle \nabla f(x_{i+1}), x_i - x_{i+1} \rangle + \frac{1}{2L} \| \nabla f(x_i) - \nabla f(x_{i+1}) \|^2 \right).$$

Note that all function value terms are deleted at $\circ$. Therefore,

$$\mathbf{U} = - \frac{1}{2L} \Big\| \sum_{i=0}^{N} (u_i - u_{i-1}) \nabla f(x_i) \Big\|^2 + \sum_{i=0}^{N} (u_i - u_{i-1}) \left( \langle \nabla f(x_i), x_0 - x_i \rangle + \frac{1}{2L} \| \nabla f(x_i) \|^2 \right)$$

$$+ \sum_{i=0}^{N-1} u_i \left( \langle \nabla f(x_{i+1}), x_i - x_{i+1} \rangle + \frac{1}{2L} \| \nabla f(x_i) - \nabla f(x_{i+1}) \|^2 \right) \tag{13}$$

and

$$\mathcal{U}_N - u_N \left( f(x_N) - f_\star \right)$$

$$= \mathbf{U} + \frac{L}{2} \Big\| x_0 - x_\star + \sum_{i=0}^{N} \frac{u_i - u_{i-1}}{L} \nabla f(x_i) \Big\|^2 .$$

Since $(x_0 - x_i)$, $(x_i - x_{i+1}) \in \text{span}\{\nabla f(x_0), \ldots, \nabla f(x_N)\}$, the value of $\mathbf{U}$ is independent with $x_0, x_\star$. Thus the only term that depends on $x_0$ and $x_\star$ is $\frac{L}{2} \left\| x_0 - x_\star + \sum_{i=0}^{N} \frac{u_i - u_{i-1}}{L} \nabla f(x_i) \right\|^2$. Next, since $x_0, x_\star$ can have any value, we can take $x_0 - x_\star = \frac{u_i - u_{i-1}}{L} \nabla f(x_i)$. Thus it gives the fact that (C1) is equivalent to $\left[ \mathbf{U} \geq 0, \quad (\forall \nabla f(x_0), \ldots, \nabla f(x_N) \in \mathbb{R}^d) \right]$.

Now we calculate $\mathcal{V}_N - \frac{1}{2} \|\nabla f(y_N)\|^2$.

$$\mathcal{V}_N - \frac{1}{2} \|\nabla f(y_N)\|^2$$

$$= v_0 \left( f_\star - f(y_N) + \frac{1}{2L} \|\nabla f(y_N)\|^2 \right) + v_0 \left( f(y_0) - f_\star \right)$$

$$+ \sum_{i=0}^{N-1} v_{i+1} \left( f(y_{i+1}) - f(y_i) + \langle \nabla f(y_{i+1}), y_i - y_{i+1} \rangle + \frac{1}{2L} \|\nabla f(y_i) - \nabla f(y_{i+1})\|^2 \right)$$

$$+ \sum_{i=0}^{N-1} (v_{i+1} - v_i) \left( f(y_i) - f(y_N) + \langle \nabla f(y_i), y_N - y_i \rangle + \frac{1}{2L} \|\nabla f(y_i) - \nabla f(y_N)\|^2 \right) - \frac{1}{2L} \|\nabla f(y_N)\|^2$$

$$\overset{(\circ)}{=} \frac{v_0}{2L} \|\nabla f(y_N)\|^2 + \sum_{i=0}^{N-1} v_{i+1} \left( \langle \nabla f(y_{i+1}), y_i - y_{i+1} \rangle + \frac{1}{2L} \|\nabla f(y_i) - \nabla f(y_{i+1})\|^2 \right)$$

$$+ \sum_{i=0}^{N-1} (v_{i+1} - v_i) \left( \langle \nabla f(y_i), y_N - y_i \rangle + \frac{1}{2L} \|\nabla f(y_i) - \nabla f(y_N)\|^2 \right) - \frac{1}{2L} \|\nabla f(y_N)\|^2.$$

Note that all function values are deleted at $(\circ)$. By the calculation result,

$$\mathbf{V} = \frac{v_0}{2L} \|\nabla f(y_N)\|^2 + \sum_{i=0}^{N-1} v_{i+1} \left( \langle \nabla f(y_{i+1}), y_i - y_{i+1} \rangle + \frac{1}{2L} \|\nabla f(y_i) - \nabla f(y_{i+1})\|^2 \right)$$

$$+ \sum_{i=0}^{N-1} (v_{i+1} - v_i) \left( \langle \nabla f(y_i), y_N - y_i \rangle + \frac{1}{2L} \|\nabla f(y_i) - \nabla f(y_N)\|^2 \right) - \frac{1}{2L} \|\nabla f(y_N)\|^2.$$
$$\tag{14}$$

(C2) is equivalent to $\left[ \mathbf{V} \geq 0, \quad (\forall \nabla f(y_0), \ldots, \nabla f(y_N) \in \mathbb{R}^d) \right]$. To establish Theorem 1, demonstrating

$$\left[ \mathbf{U} \geq 0, \quad (\forall \nabla f(x_0), \ldots, \nabla f(x_N) \in \mathbb{R}^d) \right] \quad \Leftrightarrow \quad \left[ \mathbf{V} \geq 0, \quad (\forall \nabla f(y_0), \ldots, \nabla f(y_N) \in \mathbb{R}^d) \right]$$
$$\tag{15}$$

would suffice.

**Transforming U and V into a trace.** Define

$$g_x := \left[ \nabla f(x_0) | \nabla f(x_1) | \ldots | \nabla f(x_N) \right] \in \mathbb{R}^{d \times (N+1)},$$

$$g_y := \left[ \nabla f(y_0) | \nabla f(y_1) | \ldots | \nabla f(y_N) \right] \in \mathbb{R}^{d \times (N+1)}.$$

In this paragraph, we convert the $\mathbf{U}$ of (11) and the $\mathbf{V}$ of (12) into the trace of symmetric matrices. The key idea is: For each $\langle a, b \rangle$ term where $a, b \in \text{span}\{\nabla f(x_0), \ldots, \nabla f(x_N)\}$, we can write $a = g_x \mathbf{a}$, $b = g_x \mathbf{b}$ for some $\mathbf{a}, \mathbf{b} \in \mathbb{R}^{(N+1) \times 1}$. Then

$$\langle a, b \rangle = \langle g_x \mathbf{a}, g_x \mathbf{b} \rangle = \mathbf{b}^\mathsf{T} g_x^\mathsf{T} g_x \mathbf{a} = \text{Tr} \left( \mathbf{b}^\mathsf{T} g_x^\mathsf{T} g_x \mathbf{a} \right) = \text{Tr} \left( \mathbf{a} \mathbf{b}^\mathsf{T} g_x^\mathsf{T} g_x \right) = \text{Tr} \left( g_x \mathbf{a} \mathbf{b}^\mathsf{T} g_x^\mathsf{T} \right). \tag{16}$$

Also note that $(x_0 - x_i)$, $(x_i - x_{i+1}) \in \text{span}\{\nabla f(x_0), \ldots, \nabla f(x_N)\}$ and $(y_i - y_N)$, $(y_i - y_{i+1}) \in \text{span}\{\nabla f(y_0), \ldots, \nabla f(y_N)\}$. By using this technique, we observe that there exists $\mathcal{S}(H, u)$, $\mathcal{T}(H^A, v)$ that satisfy

$$(11) = \text{Tr} \left( g_x \mathcal{S}(H, u) g_x^\mathsf{T} \right), \qquad (12) = \text{Tr} \left( g_y \mathcal{T}(H^A, v) g_y^\mathsf{T} \right).$$

From now, we specifically calculate $\mathcal{S}(H, u)$ and $\mathcal{T}(H^A, v)$. Denote $\{\mathbf{e}_i\}_{i=0}^N \in \mathbb{R}^{(N+1)\times 1}$ as a unit vector which $(i+1)$-th component is 1, $\mathbf{e}_{-1} = \mathbf{e}_{N+1} = \mathbf{0}$ and define $\mathcal{H}$ as

$$\mathcal{H} = \begin{bmatrix} 0 & 0 \\ H & 0 \end{bmatrix} \in R^{(N+1)\times(N+1)}.$$

Then, by the definition of $g_x$ and $\mathcal{H}$, we have

$$g_x \mathbf{e}_i = \nabla f(x_i) \quad 0 \le i \le N, \qquad \frac{1}{L} g_x \mathcal{H}^\mathsf{T} \mathbf{e}_0 = 0, \tag{17}$$

$$\frac{1}{L} g_x \mathcal{H}^\mathsf{T} \mathbf{e}_{i+1} = \frac{1}{L} \sum_{j=0}^i h_{i,j} g_x \mathbf{e}_j = \frac{1}{L} \sum_{j=0}^i h_{i,j} \nabla f(x_j) = x_i - x_{i+1} \quad 0 \le i \le N-1. \tag{18}$$

Therefore, we can express (11) with $\mathcal{H}, \{u_i\}_{i=0}^N, g_x$ and $\{\mathbf{e}_i\}_{i=0}^N$ using (17) and (18) as

(11)

$$= -\frac{1}{2L} \left\| \sum_{i=0}^N (u_i - u_{i-1})\nabla f(x_i) \right\|^2 + \sum_{i=0}^N (u_i - u_{i-1})\left( \langle \nabla f(x_i), x_0 - x_i \rangle + \frac{1}{2L}\|\nabla f(x_i)\|^2 \right)$$

$$+ \sum_{i=0}^{N-1} u_i \left( \langle \nabla f(x_{i+1}), x_i - x_{i+1} \rangle + \frac{1}{2L}\|\nabla f(x_i) - \nabla f(x_{i+1})\|^2 \right)$$

$$= -\frac{1}{2L} \left\| \sum_{i=0}^N (u_i - u_{i-1})g_x \mathbf{e}_i \right\|^2 + \sum_{i=0}^N (u_i - u_{i-1})\left( \left\langle g_x \mathbf{e}_i, \frac{1}{L} g_x \mathcal{H}^\mathsf{T}(\mathbf{e}_0 + \cdots + \mathbf{e}_i) \right\rangle + \frac{1}{2L}\|g_x \mathbf{e}_i\|^2 \right)$$

$$+ \sum_{i=0}^{N-1} u_i \left( \left\langle g_x \mathbf{e}_{i+1}, \frac{1}{L} g_x \mathcal{H}^\mathsf{T} \mathbf{e}_{i+1} \right\rangle + \frac{1}{2L}\|g_x(\mathbf{e}_i - \mathbf{e}_{i+1})\|^2 \right).$$

Using (16) induces (11) $= \operatorname{Tr}\left(g_x \mathcal{S}(H, u) g_x^\mathsf{T}\right)$ where

$$\mathcal{S}(H, u)$$

$$= -\frac{1}{2L} \left( \sum_{i=0}^N (u_i - u_{i-1})\mathbf{e}_i \right) \left( \sum_{i=0}^N (u_i - u_{i-1})\mathbf{e}_i \right)^\mathsf{T}$$

$$+ \frac{1}{2L} \mathcal{H}^\mathsf{T} \left[ \sum_{i=0}^N u_i(\mathbf{e}_0 + \cdots + \mathbf{e}_i)(\mathbf{e}_i - \mathbf{e}_{i+1})^\mathsf{T} \right] + \frac{1}{2L} \left[ \sum_{i=0}^N u_i(\mathbf{e}_i - \mathbf{e}_{i+1})(\mathbf{e}_0 + \cdots + \mathbf{e}_i)^\mathsf{T} \right] \mathcal{H}$$

$$+ \frac{1}{2L} \left[ \sum_{i=0}^N u_i \left( (\mathbf{e}_i - \mathbf{e}_{i+1})\mathbf{e}_i^\mathsf{T} + \mathbf{e}_i(\mathbf{e}_i - \mathbf{e}_{i+1})^\mathsf{T} \right) - u_N \mathbf{e}_N \mathbf{e}_N^\mathsf{T} \right].$$

$$\tag{19}$$

Similarly, we calculate (12). Define $\mathcal{H}^A$ as a anti-transpose matrix of $\mathcal{H}$:

$$\mathcal{H}^A = \begin{bmatrix} 0 & 0 \\ H^A & 0 \end{bmatrix} \in R^{(N+1)\times(N+1)}.$$

Then, by the definition of $g_y$ and $\mathcal{H}^A$, we have

$$g_y \mathbf{e}_i = \nabla f(y_i) \quad 0 \le i \le N, \qquad \frac{1}{L} g_y \left(\mathcal{H}^A\right)^\mathsf{T} \mathbf{e}_0 = 0, \tag{20}$$

$$\frac{1}{L} g_y \left(\mathcal{H}^A\right)^\mathsf{T} \mathbf{e}_{i+1} = \frac{1}{L} \sum_{j=0}^i h_{i,j} g_y \mathbf{e}_j = \frac{1}{L} \sum_{j=0}^i h_{i,j} \nabla f(y_j) = y_i - y_{i+1} \quad 0 \le i \le N-1. \tag{21}$$

Therefore, we can express (12) with $\mathcal{H}^A, \{v_i\}_{i=0}^N, g$ and $\{\mathbf{e}_i\}_{i=0}^N$ using (20) and (21) as

(12)

$$= \frac{v_0 - 1}{2L}\|\nabla f(y_N)\|^2 + \sum_{i=0}^{N-1} v_{i+1}\left(\langle \nabla f(y_{i+1}), y_i - y_{i+1}\rangle + \frac{1}{2L}\|\nabla f(y_i) - \nabla f(y_{i+1})\|^2\right)$$

$$+ \sum_{i=0}^{N-1}(v_{i+1} - v_i)\left(\langle \nabla f(y_i), y_N - y_i\rangle + \frac{1}{2L}\|\nabla f(y_i) - \nabla f(y_N)\|^2\right)$$

$$= \frac{v_0 - 1}{2L}\|g_y\mathbf{e}_N\|^2 + \sum_{i=0}^{N-1} v_{i+1}\left(\left\langle g_y\mathbf{e}_{i+1}, \frac{1}{L}g_y\left(\mathcal{H}^A\right)^\intercal \mathbf{e}_{i+1}\right\rangle + \frac{1}{2L}\|g_y(\mathbf{e}_i - \mathbf{e}_{i+1})\|^2\right)$$

$$+ \sum_{i=0}^{N-1}(v_{i+1} - v_i)\left(-\left\langle g_y\mathbf{e}_i, \frac{1}{L}g_y\left(\mathcal{H}^A\right)^\intercal (\mathbf{e}_{i+1} + \cdots + \mathbf{e}_N)\right\rangle + \frac{1}{2L}\|g_y(\mathbf{e}_i - \mathbf{e}_N)\|^2\right).$$

We can write (12) $= \mathrm{Tr}\left(g_y\mathcal{T}(H^A, v)g_y^\intercal\right)$ where

$$\mathcal{T}(H^A, v)$$

$$= \frac{1}{2L}\left[\sum_{i=0}^N v_i\left((\mathbf{e}_{i-1} - \mathbf{e}_i)(\mathbf{e}_{i-1} - \mathbf{e}_N)^\intercal + (\mathbf{e}_{i-1} - \mathbf{e}_N)(\mathbf{e}_{i-1} - \mathbf{e}_i)^\intercal\right)\right] - \frac{v_0}{2L}\mathbf{e}_0\mathbf{e}_0^\intercal - \frac{1}{2L}\mathbf{e}_N\mathbf{e}_N^\intercal$$

$$+ \frac{1}{2L}\left[\sum_{i=0}^N v_i\left(\left(\mathcal{H}^A\right)^\intercal(\mathbf{e}_i + \cdots + \mathbf{e}_N)(\mathbf{e}_i - \mathbf{e}_{i-1})^\intercal + (\mathbf{e}_i - \mathbf{e}_{i-1})(\mathbf{e}_i + \cdots + \mathbf{e}_N)^\intercal\mathcal{H}^A\right)\right]$$

$$= \frac{1}{2L}\left[\sum_{i=0}^N \frac{1}{u_{N-i}}\left((\mathbf{e}_{i-1} - \mathbf{e}_i)(\mathbf{e}_{i-1} - \mathbf{e}_N)^\intercal + (\mathbf{e}_{i-1} - \mathbf{e}_N)(\mathbf{e}_{i-1} - \mathbf{e}_i)^\intercal\right)\right] - \frac{1}{2u_NL}\mathbf{e}_0\mathbf{e}_0^\intercal - \frac{1}{2L}\mathbf{e}_N\mathbf{e}_N^\intercal$$

$$+ \frac{1}{2L}\left[\sum_{i=0}^N \frac{1}{u_{N-i}}\left(\left(\mathcal{H}^A\right)^\intercal(\mathbf{e}_i + \cdots + \mathbf{e}_N)(\mathbf{e}_i - \mathbf{e}_{i-1})^\intercal + (\mathbf{e}_i - \mathbf{e}_{i-1})(\mathbf{e}_i + \cdots + \mathbf{e}_N)^\intercal\mathcal{H}^A\right)\right].$$

(22)

**Finding auxiliary matrix $\mathcal{M}(u)$ that gives the relation between $\mathcal{S}(H, u)$ and $\mathcal{T}(H^A, v)$.** We can show that there exists an invertible $M(u) \in \mathbb{R}^{(N+1)\times(N+1)}$ such that

$$\mathcal{S}(H, u) = \mathcal{M}(u)^\intercal \mathcal{T}(H^A, v)\mathcal{M}(u). \tag{23}$$

If we assume the above equation,

$$\mathrm{Tr}\left(g_x\mathcal{S}(H, u)g_x^\intercal\right) = \mathrm{Tr}\left(g_y\mathcal{T}(H^A, v)g_y^\intercal\right) \tag{24}$$

with $g_y = g_x\mathcal{M}(u)^\intercal$. Since $\mathcal{M}(u)$ is invertible,

$$\{g|g \in \mathbb{R}^{d\times(N+1)}\} = \{g\mathcal{M}(u)^\intercal|g \in \mathbb{R}^{d\times(N+1)}\}. \tag{25}$$

Also, note that

$$\left[\mathbf{U} \geq 0 \quad \forall \nabla f(x_0), \ldots, \nabla f(x_N)\right] \quad \Leftrightarrow \quad \left[\mathrm{Tr}\left(g_x\mathcal{S}(H, u)g_x^\intercal\right) \geq 0 \quad \forall g_x \in \mathbb{R}^{d\times(N+1)}\right]$$

and

$$\left[\mathbf{V} \geq 0 \quad \forall \nabla f(y_0), \ldots, \nabla f(y_N)\right] \quad \Leftrightarrow \quad \left[\mathrm{Tr}\left(g_y\mathcal{T}(H^A, u)g_y^\intercal\right) \geq 0 \quad \forall g_y \in \mathbb{R}^{d\times(N+1)}\right].$$

By combining (24) and (25), we obtain

$$\left[\mathrm{Tr}\left(g_x\mathcal{S}(H, u)g_x^\intercal\right) \geq 0 \quad \forall g_x \in \mathbb{R}^{d\times(N+1)}\right]$$

$$\Leftrightarrow \left[\mathrm{Tr}\left(g_y\mathcal{S}(H, u)g_y^\intercal\right) \geq 0 \quad \forall g_x \in \mathbb{R}^{d\times(N+1)}, \ g_y = g_x\mathcal{M}(u)^\intercal\right]$$

$$\Leftrightarrow \left[\mathrm{Tr}\left(g_y\mathcal{S}(H, u)g_y^\intercal\right) \geq 0 \quad \forall g_y \in \mathbb{R}^{d\times(N+1)}\right].$$

To sum up, we obtain (15)

$$\left[\mathbf{U} \geq 0 \quad \forall \nabla f(x_0), \ldots, \nabla f(x_N)\right] \quad \Leftrightarrow \quad \left[\mathbf{V} \geq 0 \quad \forall \nabla f(y_0), \ldots, \nabla f(y_N)\right],$$

which concludes the proof.

**Explicit form of $\mathcal{M}(u)$ and justification of** (23)**.** Explicit form of $\mathcal{M}(u)$ is

$$
\mathcal{M} = \begin{bmatrix}
0 & \cdots & 0 & 0 & u_N \\
0 & \cdots & 0 & u_{N-1} & u_N - u_{N-1} \\
0 & \cdots & u_{N-2} & u_{N-1} - u_{N-2} & u_N - u_{N-1} \\
\vdots & \vdots & \vdots & \vdots & \vdots \\
u_0 & \cdots & u_{N-2} - u_{N-3} & u_{N-1} - u_{N-2} & u_N - u_{N-1}
\end{bmatrix} \in \mathbb{R}^{(N+1)\times(N+1)}. \quad (26)
$$

Now, we express $\mathcal{M}(u) = \sum\limits_{0 \le i,j \le N} m_{ij}(u)\mathbf{e}_i\mathbf{e}_j^\mathsf{T}$, $\mathcal{S}(H,u) = \sum\limits_{0 \le i,j \le N} s_{ij}\mathbf{e}_i\mathbf{e}_j^\mathsf{T}$, and $\mathcal{T}(H^A, v) = \sum\limits_{0 \le i,j \le N} t_{ij}\mathbf{e}_i\mathbf{e}_j^\mathsf{T}$. Calculating $\mathcal{M}^\mathsf{T}(u)\mathcal{T}(H^A,v)\mathcal{M}(u)$ gives

$$
\mathcal{M}^\mathsf{T}(u)\mathcal{T}(H^A,v)\mathcal{M}(u) = \left(\sum_{i,j} m_{ij}(u)\mathbf{e}_j\mathbf{e}_i^\mathsf{T}\right)\left(\sum_{i,j} t_{ij}\mathbf{e}_i\mathbf{e}_j^\mathsf{T}\right)\left(\sum_{i,j} m_{ij}(u)\mathbf{e}_i\mathbf{e}_j^\mathsf{T}\right)
$$

$$
= \sum_{i,j} t_{ij}\left(\sum_k m_{ik}(u)\mathbf{e}_k\right)\left(\sum_l m_{jl}(u)\mathbf{e}_l\right)^\mathsf{T}
$$

$$
:= \sum_{i,j} t_{ij}\mathbf{f}_i(u)\mathbf{f}_j(u)^\mathsf{T}.
$$

Thus it is enough to show that $\sum\limits_{i,j} s_{ij}\mathbf{e}_i\mathbf{e}_j^\mathsf{T}$ and $\sum\limits_{i,j} t_{ij}\mathbf{f}_i(u)\mathbf{f}_j(u)^\mathsf{T}$ are the same under the basis transformation $\mathbf{f}_i(u) = \sum\limits_k m_{ik}(u)\mathbf{e}_k$. From here, we briefly write $\mathbf{f}_i$ instead $\mathbf{f}_i(u)$, and $\mathbf{f}_{-1} = \mathbf{0}$. Note that $u_i(\mathbf{e}_i - \mathbf{e}_{i+1}) = (\mathbf{f}_{N-i} - \mathbf{f}_{N-i-1})$, $0 \le i \le N$ by definition of $\mathcal{M}(u)$. Therefore, we have

$$
\frac{1}{L}\mathcal{H}^\mathsf{T}\sum_{i=0}^N u_i(\mathbf{e}_0 + \cdots + \mathbf{e}_i)(\mathbf{e}_i - \mathbf{e}_{i+1})^\mathsf{T} = \frac{1}{L}\mathcal{H}^\mathsf{T}\sum_{i=0}^N (\mathbf{e}_0 + \cdots + \mathbf{e}_i)(\mathbf{f}_{N-i} - \mathbf{f}_{N-i-1})^\mathsf{T}
$$

$$
= \frac{1}{L}\mathcal{H}^\mathsf{T}\left[\sum_{i=0}^N \mathbf{e}_i\mathbf{f}_{N-i}^\mathsf{T}\right].
$$

Therefore, we can rewrite (19) as follows:

$$
\mathcal{S}(H,u) = -\frac{1}{2L}\mathbf{f}_N\mathbf{f}_N^\mathsf{T} + \frac{1}{2L}\mathcal{H}^\mathsf{T}\left[\sum_{i=0}^N \mathbf{e}_i\mathbf{f}_{N-i}^\mathsf{T}\right] + \frac{1}{2L}\sum_{i=0}^N \mathbf{f}_{N-i}\mathbf{e}_i^\mathsf{T}\mathcal{H}
$$

$$
+ \frac{1}{2L}\sum_{i=0}^N \left[(\mathbf{f}_{N-i} - \mathbf{f}_{N-i-1})\mathbf{e}_i^\mathsf{T} + \mathbf{e}_i(\mathbf{f}_{N-i}\mathbf{f}_{N-i-1})^\mathsf{T}\right] - \frac{u_N}{2L}\mathbf{e}_N\mathbf{e}_N^\mathsf{T}
$$

$$
= \underbrace{-\frac{1}{2L}\mathbf{f}_N\mathbf{f}_N^\mathsf{T}}_{\mathbf{A}_1} + \underbrace{\frac{1}{2L}\left[\sum_{i,j} h_{i,j}\mathbf{e}_j\mathbf{f}_{N-i}^\mathsf{T}\right] + \frac{1}{2L}\left[\sum_{i,j} h_{i,j}\mathbf{f}_{N-i}\mathbf{e}_j^\mathsf{T}\right]}_{\mathbf{B}_1} \quad (27)
$$

$$
+ \underbrace{\frac{1}{2L}\sum_{i=0}^N \left[(\mathbf{f}_{N-i} - \mathbf{f}_{N-i-1})\mathbf{e}_i^\mathsf{T} + \mathbf{e}_i(\mathbf{f}_{N-i} - \mathbf{f}_{N-i-1})^\mathsf{T}\right]}_{\mathbf{C}_1} - \underbrace{\frac{u_N}{2L}\mathbf{e}_N\mathbf{e}_N^\mathsf{T}}_{\mathbf{D}_1}.
$$

Similarly, by using

$$
\mathbf{e}_{N-i} - \mathbf{e}_{N-i+1} = \frac{1}{u_{N-i}}\left(\mathbf{f}_i - \mathbf{f}_{i-1}\right) = v_i\left(\mathbf{f}_i - \mathbf{f}_{i-1}\right),
$$

we can rewrite (22) as follows:

$$\mathcal{M}^{\mathsf{T}}(u)\mathcal{T}(H^A, v)\mathcal{M}(u)$$

$$=\frac{1}{2L}\sum_{i=0}^{N}(\mathbf{e}_{N-i+1} - \mathbf{e}_{N-i})(\mathbf{f}_{i-1} - \mathbf{f}_N)^{\mathsf{T}} + (\mathbf{f}_{i-1} - \mathbf{f}_N)(\mathbf{e}_{N-i+1} - \mathbf{e}_{N-i})^{\mathsf{T}}$$

$$-\frac{1}{2u_N L}\mathbf{f}_0\mathbf{f}_0^{\mathsf{T}} - \frac{1}{2L}\mathbf{f}_N\mathbf{f}_N^{\mathsf{T}} + \frac{1}{2L}\left(\mathcal{H}^A\right)^{\mathsf{T}}\left[\sum_{i=0}^{N}\mathbf{f}_i\mathbf{e}_{N-i}^{\mathsf{T}}\right] + \frac{1}{2L}\left[\sum_{i=0}^{N}\mathbf{e}_{N-i}\mathbf{f}_i^{\mathsf{T}}\right]\mathcal{H}^A$$

$$=-\underbrace{\frac{1}{2u_N L}\mathbf{f}_0\mathbf{f}_0^{\mathsf{T}}}_{\mathbf{D}_2} + \underbrace{\frac{1}{2L}\left[\sum_{i,j}h_{N-i,N-j}\mathbf{f}_j\mathbf{e}_{N-i}^{\mathsf{T}}\right] + \frac{1}{2}\left[\sum_{i,j}h_{N-j,N-i}\mathbf{e}_{N-i}\mathbf{f}_j^{\mathsf{T}}\right]}_{\mathbf{B}_2}$$

$$+ \underbrace{\left[\frac{1}{2L}\sum_{i=0}^{N}(\mathbf{e}_{N-i+1} - \mathbf{e}_{N-i})\mathbf{f}_{i-1}^{\mathsf{T}} + \mathbf{f}_{i-1}(\mathbf{e}_{N-i+1} - \mathbf{e}_{N-i})^{\mathsf{T}} - \frac{1}{2L}\left(\mathbf{e}_0\mathbf{f}_N^{\mathsf{T}} + \mathbf{f}_N\mathbf{e}_0^{\mathsf{T}}\right)\right]}_{\mathbf{C}_2} - \underbrace{\frac{1}{2L}\mathbf{f}_N\mathbf{f}_N^{\mathsf{T}}}_{\mathbf{A}_2}.$$

$$(28)$$

For the final step, we compare (28) and (27) term-by-term, by showing $\mathbf{X}_1 = \mathbf{X}_2$ for $X = A, B, C, D$.

- $\mathbf{A}_1 = \mathbf{A}_2$ comes directly.
- $\mathbf{B}_1 = \mathbf{B}_2$ comes from changing the summation index $i \to N - i$ and $j \to N - j$.
- $\mathbf{C}_1 = \mathbf{C}_2$ comes from the expansion of summation.
- $\mathbf{D}_1 = \mathbf{D}_2$ comes from $\mathbf{f}_0 = u_N\mathbf{e}_N$.

Therefore,

$$\mathcal{S}(H, u) = \mathcal{M}^{\mathsf{T}}(u)\mathcal{T}(H^A, v)\mathcal{M}(u),$$

which concludes the proof.

**Remark 1.** We can interpret $\mathcal{M}$ as a basis transformation, where

$$u_N\mathbf{e}_N = \mathbf{f}_0, \quad u_i(\mathbf{e}_i - \mathbf{e}_{i+1}) = \mathbf{f}_{N-i} - \mathbf{f}_{N-i-1} \quad i = 0, 1, \ldots, N - 1. \tag{29}$$

**Remark 2.** To clarify, the quantifier $[\forall \nabla f(x_0), \ldots, \nabla f(x_N)]$ in (C1) means $\nabla f(x_0), \ldots, \nabla f(x_N)$ can be any arbitrary vectors in $\mathbb{R}^d$. This is different from $[\nabla f(x_0), \ldots, \nabla f(x_N)$ be gradient of some $f \colon \mathbb{R}^d \to \mathbb{R}]$. The same is true for (C2).

## B  Omitted calculation of Section 2

### B.1  Calculation of $H$ matrices

**OGM and OGM-G.**  [26, Proposition 3] provides a recursive formula for (OGM) as follows:

$$h_{k+1,i} = \begin{cases} \frac{\theta_k - 1}{\theta_{k+1}}h_{k,i} & i = 0, \ldots, k - 2 \\ \frac{\theta_k - 1}{\theta_{k+1}}\left(h_{k,k-1} - 1\right) & i = k - 1 \\ 1 + \frac{2\theta_k - 1}{\theta_{k+1}} & i = k. \end{cases} \tag{30}$$

If $k > i$,

$$h_{k+1,k-1} = \frac{\theta_k - 1}{\theta_{k+1}}\frac{2\theta_{k-1} - 1}{\theta_k},$$

$$h_{k+1,i} = \frac{\theta_k - 1}{\theta_{k+1}}h_{k,i} = \cdots = \left(\prod_{l=i+2}^{k}\frac{\theta_l - 1}{\theta_{l+1}}\right)h_{i+1,i-1} = \left(\prod_{l=i+1}^{k}\frac{\theta_l - 1}{\theta_{l+1}}\right)\frac{2\theta_i - 1}{\theta_{i+1}}.$$

Thus $H_{\text{OGM}}$ can be calculated as

$$H_{\text{OGM}}(k+1, i+1) = \begin{cases} 0 & i > k \\ 1 + \frac{2\theta_k - 1}{\theta_{k+1}} & i = k \\ \left( \prod_{l=i+1}^{k} \frac{\theta_l - 1}{\theta_{l+1}} \right) \frac{2\theta_i - 1}{2\theta_{i+1}} & i < k. \end{cases}$$

Recursive formula [29] of (OGM-G) is as following:

$$h_{k+1,i} = \begin{cases} \frac{\theta_{N-i-1} - 1}{\theta_{N-i}} h_{k+1,i+1} & i = 0, \dots, k-2 \\ \frac{\theta_{N-k} - 1}{\theta_{N-k+1}} \left( h_{k+1,i} - 1 \right) & i = k-1 \\ 1 + \frac{2\theta_{N-k-1} - 1}{\theta_{N-k}} & i = k. \end{cases} \tag{31}$$

If $k > i$,

$$h_{k+1,k-1} = \frac{\theta_{N-k} - 1}{\theta_{N-k+1}} \frac{2\theta_{N-k-1} - 1}{\theta_{N-k}},$$

$$h_{k+1,i} = \frac{\theta_{N-i-1} - 1}{\theta_{N-i}} h_{k+1,i+1} = \cdots = \left( \prod_{l=N-k+1}^{N-i-1} \frac{\theta_l - 1}{\theta_{l+1}} \right) h_{k+1,k-1} = \left( \prod_{l=N-k}^{N-i-1} \frac{\theta_l - 1}{\theta_{l+1}} \right) \frac{2\theta_{N-k-1} - 1}{\theta_{N-k}}.$$

Thus $H_{\text{OGM-G}}$ can be calculated as

$$H_{\text{OGM-G}}(k+1, i+1) = \begin{cases} 0 & i > k \\ 1 + \frac{2\theta_{N-k-1} - 1}{\theta_{N-k}} & i = k \\ \left( \prod_{l=N-k}^{N-i-1} \frac{\theta_l - 1}{\theta_{l+1}} \right) \frac{2\theta_{N-k-1} - 1}{\theta_{N-k}} & i < k \end{cases},$$

which gives $H_{\text{OGM-G}} = H_{\text{OGM}}^A$.

**Gradient Descent.** For (GD), $H(i+1, k+1) = h_{i+1,k} = h\delta_{i+1,k+1}$, where $\delta_{i,j}$ is the Kronecker delta. Therefore, $H_{\text{GD}} = H_{\text{GD}}^A$.

**OBL-F$_\flat$ and OBL-G$_\flat$.** Recall $\gamma = \sqrt{\frac{N(N+1)}{2}}$. We obtain the recursive formula of the $H$ matrix of (OBL-F$_\flat$).

$$h_{k+1,i} = \begin{cases} \frac{k}{k+3} h_{k,i} & k = 0, \dots, N-2, i = 0, \dots, k-2 \\ 1 + \frac{2k}{k+3} & k = 0, \dots, N-2, i = k \\ \frac{k}{k+3} \left( h_{k,k-1} - 1 \right) & k = 1, \dots, N-2, i = k-1 \\ 1 + \frac{N-1}{\gamma+1} & k = N-1, i = N-1 \\ \frac{N-1}{2(\gamma+1)} \left( h_{N-1,N-2} - 1 \right) & k = N-1, i = N-2 \\ \frac{N-1}{2(\gamma+1)} h_{N-1,i} & k = N-1, i = 0, \dots, N-3 \end{cases}. \tag{32}$$

By using the above formula, we obtain

$$H_{\text{OBL-F}_\flat}(k+1, i+1) = \begin{cases} 1 + \frac{2k}{k+3} & k = 0, \dots, N-2, i = k \\ \frac{2i(i+1)(i+2)}{(k+1)(k+2)(k+3)} & k = 0, \dots, N-2, i = 0, \dots, k-1 \\ 1 + \frac{N-1}{\gamma+1} & k = N-1, i = N-1 \\ \frac{i(i+1)(i+2)}{(\gamma+1)N(N+1)} & k = N-1, i = 0, \dots, N-2 \end{cases}.$$

Similarly, we achieve the following recursive formula of the $H$ matrix of (OBL-G$_\flat$).

$$h_{k+1,i} = \begin{cases} \frac{N-k-1}{N-k+2}h_{k,i} & k = 1,\ldots,N-1,\ i = 0,\ldots,k-2 \\ 1 + \frac{2(N-k-1)}{N-k+2} & k = 1,\ldots,N-1,\ i = k \\ \frac{N-k-1}{N-k+2}(h_{k,k-1}-1) & k = 1,\ldots,N-1,\ i = k-1 \\ 1 + \frac{N-1}{\gamma+1} & k = 0,\ i = 0 \end{cases} \tag{33}$$

By using the above recursive formula, we obtain

$$H_{\text{OBL-G}_\flat}(k+1,i+1) = \begin{cases} 1 + \frac{N-1}{\gamma+1} & k = 0,\ i = 0 \\ \frac{(N-k-1)(N-k)(N-k+1)}{(\gamma+1)N(N+1)} & k = 1,\ldots,N-1,\ i = 0 \\ 1 + \frac{2(N-k-1)}{N-k+2} & k = 1,\ldots,N-1,\ i = k \\ \frac{2(N-k-1)(N-k)(N-k+1)}{(N-i)(N-i+1)(N-i+2)} & k = 1,\ldots,N-1,\ i = 1,\ldots,k-1 \end{cases}$$

Thus $H_{\text{OBL-F}_\flat} = H^A_{\text{OBL-G}_\flat}$.

## B.2 Calculation of energy functions

**Calculation of U and V with $H$ matrix** In this paragraph, we calculate **U** and **V**. Recall (13) and (14).

First, we put $x_{k+1} - x_k = -\frac{1}{L}\sum_{i=0}^{k} h_{k+1,i}\nabla f(x_i)$ to (11). We have

$$\begin{aligned}
\mathbf{U} &= -\frac{1}{2L}\left\|\sum_{i=0}^{N}(u_i - u_{i-1})\nabla f(x_i)\right\|^2 + \sum_{i=0}^{N}(u_i - u_{i-1})\left(\langle\nabla f(x_i), x_0 - x_i\rangle + \frac{1}{2L}\|\nabla f(x_i)\|^2\right) \\
&\quad + \sum_{i=0}^{N-1} u_i\left(\langle\nabla f(x_{i+1}), x_i - x_{i+1}\rangle + \frac{1}{2L}\|\nabla f(x_i) - \nabla f(x_{i+1})\|^2\right) \\
&= -\frac{1}{2L}\left\|\sum_{i=0}^{N}(u_i - u_{i-1})\nabla f(x_i)\right\|^2 + \sum_{i=0}^{N}\frac{u_i - u_{i-1}}{L}\left(\left\langle\nabla f(x_i), \sum_{l=0}^{i-1}\sum_{j=0}^{l} h_{l+1,j}\nabla f(x_j)\right\rangle + \frac{1}{2}\|\nabla f(x_i)\|^2\right) \\
&\quad + \sum_{i=0}^{N-1}\frac{u_i}{L}\left(\left\langle\nabla f(x_{i+1}), \sum_{j=0}^{i} h_{i+1,j}\nabla f(x_j)\right\rangle + \frac{1}{2}\|\nabla f(x_i) - \nabla f(x_{i+1})\|^2\right).
\end{aligned}$$

By arranging, we obtain

$$\mathbf{U} = \sum_{0 \le j \le i \le N}\frac{s_{i,j}}{L}\langle\nabla f(x_i), \nabla f(x_j)\rangle$$

where

$$s_{i,j} = \begin{cases} -\frac{1}{2}(u_N - u_{N-1})^2 + \frac{1}{2}u_N & j = i,\ i = N \\ -\frac{1}{2}(u_i - u_{i-1})^2 + u_i & j = i,\ i = 0,\ldots,N-1 \\ u_i h_{i,i-1} - u_{i-1} - (u_i - u_{i-1})(u_{i-1} - u_{i-2}) & j = i-1 \\ (u_i - u_{i-1})\sum_{l=j}^{i-1} h_{l+1,j} + u_{i-1}h_{i,j} - (u_i - u_{i-1})(u_j - u_{j-1}) & j = 0,\ldots,i-2. \end{cases} \tag{34}$$

Recall that we defined $u_{-1} = 0$. Next, put $y_{k+1} - y_k = -\frac{1}{L} \sum_{i=0}^{k} h_{k+1,i} \nabla f(y_i)$ to (12). We have

$$\mathbf{V} = \mathcal{V}_N - \frac{1}{2L} \|\nabla f(y_N)\|^2$$

$$= \frac{v_0}{2L} \|\nabla f(y_N)\|^2 + \sum_{i=0}^{N-1} v_{i+1} \left( \langle \nabla f(y_{i+1}), y_i - y_{i+1} \rangle + \frac{1}{2L} \|\nabla f(y_i) - \nabla f(y_{i+1})\|^2 \right)$$

$$+ \sum_{i=0}^{N-1} (v_{i+1} - v_i) \left( \langle \nabla f(y_i), y_N - y_i \rangle + \frac{1}{2L} \|\nabla f(y_i) - \nabla f(y_N)\|^2 \right) - \frac{1}{2L} \|\nabla f(y_N)\|^2$$

$$= \frac{v_0 - 1}{2L} \|\nabla f(y_N)\|^2 + \sum_{i=0}^{N-1} v_{i+1} \left( \left\langle \nabla f(y_{i+1}), \frac{1}{L} \sum_{j=0}^{i} h_{i+1,j} \nabla f(y_j) \right\rangle + \frac{1}{2L} \|\nabla f(y_i) - \nabla f(y_{i+1})\|^2 \right)$$

$$+ \sum_{i=0}^{N-1} (v_{i+1} - v_i) \left( \left\langle \nabla f(y_i), -\frac{1}{L} \sum_{l=i}^{N-1} \sum_{j=0}^{l} h_{l+1,j} \nabla f(y_j) \right\rangle + \frac{1}{2L} \|\nabla f(y_i) - \nabla f(y_N)\|^2 \right).$$

By arranging, we obtain

$$\mathbf{V} = \sum_{0 \le j \le i \le N} \frac{t_{i,j}}{L} \langle \nabla f(y_i), \nabla f(y_j) \rangle$$

where

$$t_{i,j} = \begin{cases} \frac{v_1}{2} + \frac{v_1 - v_0}{2} - (v_1 - v_0) \sum_{l=0}^{N-1} h_{l+1,0} & i = 0, j = i \\ \frac{v_{i+1} + v_i}{2} + \frac{v_{i+1} - v_i}{2} - (v_{i+1} - v_i) \sum_{l=i}^{N-1} h_{l+1,i} & i = 1, \ldots, N-1, j = i \\ \frac{v_0 - 1}{2} + \frac{v_N}{2} + \sum_{i=0}^{N-1} \frac{v_{i+1} - v_i}{2} & i = N, j = i \\ v_i h_{i,i-1} - v_i - (v_{i+1} - v_i) \sum_{l=i}^{N-1} h_{l+1,i-1} - (v_i - v_{i-1}) \sum_{l=i}^{N-1} h_{l+1,i} & i = 1, \ldots, N-1, j = i-1 \\ v_N h_{N,N-1} - v_N - (v_N - v_{N-1}) & i = N, j = i-1 \\ v_i h_{i,j} - (v_{i+1} - v_i) \sum_{l=i}^{N-1} h_{l+1,j} - (v_{j+1} - v_j) \sum_{l=i}^{N-1} h_{l+1,i} & i = 2, \ldots, N-1, j = 0, \ldots, i-2 \\ v_N h_{N,j} - (v_{j+1} - v_j) & i = N, j = 0, \ldots, N-2 \end{cases}$$

$$\tag{35}$$

Now we calculate $\{s_{ij}\}$ and $\{t_{ij}\}$ for [(OGM), (OGM-G)] [(OBL-F$_\flat$), (OBL-G$_\flat$)] and [(GD), (GD)].

### B.2.1 Calculation of energy function of (OGM) and (OGM-G)

We will show $s_{ij} = 0$ and $t_{ij} = 0$ for all $i, j$. By the definition of $\{u_i\}_{i=-1}^{N}$ and $\{\theta_i\}_{i=-1}^{N}$,

$$u_i - u_{i-1} = 2\theta_i^2 - 2\theta_{i-1}^2 = 2\theta_i \qquad 0 \le i \le N-1,$$
$$u_N - u_{N-1} = \theta_N^2 - 2\theta_{N-1}^2 = \theta_N.$$

Therefore, we have

$$s_{i,i} = \begin{cases} -\frac{1}{2}(u_N - u_{N-1})^2 + \frac{1}{2} u_N = -\theta_N + \theta_N = 0 & j = i, i = N \\ -\frac{1}{2}(u_i - u_{i-1})^2 + u_i = -\theta_i + \theta_i = 0 & j = i, i = 0, \ldots, N-1. \end{cases}$$

Now we claim that $s_{ij} = 0$ when $j \neq i$. In the case that $i = j + 1$, we have

$$s_{i,i-1} = u_i h_{i,i-1} - u_{i-1} - (u_i - u_{i-1})(u_{i-1} - u_{i-2})$$

$$= u_i \left( \frac{2\theta_{i-1} - 1}{\theta_i} + 1 \right) - u_{i-1} - (u_i - u_{i-1})(u_{i-1} - u_{i-2})$$

$$= \begin{cases} 2\theta_i^2 \left( \frac{2\theta_{i-1}-1}{\theta_i} + 1 \right) - 2\theta_{i-1}^2 - 4\theta_i \theta_{i-1} & 0 \leq i \leq N - 1 \\ \theta_N^2 \left( \frac{2\theta_{N-1}-1}{\theta_N} + 1 \right) - 2\theta_{N-1}^2 - 2\theta_{N-1}\theta_N & i = N \end{cases}$$

$$= 0.$$

We show $s_{ij} = 0$ for $j \neq i$ with induction on $i$, i.e., proving

$$\left[ s_{i,j} = (u_i - u_{i-1}) \sum_{l=j}^{i-1} h_{l+1,j} + u_{i-1} h_{i,j} - (u_i - u_{i-1})(u_j - u_{j-1}) = 0, \quad j = 0, \ldots, i - 2 \right].$$
(36)

First we prove (36) for $i = j + 2$.

$$(u_{j+2} - u_{j+1})(h_{j+1,j} + h_{j+2,j}) + u_{j+1} h_{j+2,j} - (u_{j+2} - u_{j+1})(u_j - u_{j-1})$$

$$= (u_{j+2} - u_{j+1})h_{j+1,j} + u_{j+2}h_{j+2,j} - (u_{j+2} - u_{j+1})(u_j - u_{j-1})$$

$$= \begin{cases} 2\theta_{j+2}h_{j+1,j} + 2\theta_{j+2}^2 h_{j+2,j} - 4\theta_{j+2}\theta_j & 0 \leq j \leq N - 3 \\ \theta_N h_{N-1,N-2} + \theta_N^2 h_{N,N-2} - 2\theta_N \theta_{N-2} & j = N - 2 \end{cases}$$

$$= 0.$$

Next, assume (36) for $i = i_0$. When $i = i_0 + 1$,

$$(u_{i_0+1} - u_{i_0}) \sum_{l=j}^{i_0} h_{l+1,j} + u_{i_0} h_{i_0+1,j} - (u_{i_0+1} - u_{i_0})(u_j - u_{j-1})$$

$$= (u_{i_0+1} - u_{i_0}) \left( \sum_{l=j}^{i_0-1} h_{l+1,j} + h_{i_0+1,j} \right) + u_{i_0} h_{i_0+1,j} - (u_{i_0+1} - u_{i_0})(u_j - u_{j-1})$$

$$= (u_{i_0+1} - u_{i_0}) \left( \frac{(u_{i_0} - u_{i_0-1})(u_j - u_{j-1}) - u_{i_0-1}h_{i_0,j}}{u_{i_0} - u_{i_0-1}} + h_{i_0+1,j} \right) + u_{i_0} h_{i_0+1,j}$$

$$\quad - (u_{i_0+1} - u_{i_0})(u_j - u_{j-1})$$

$$= u_{i_0+1}h_{i_0+1,j} - \frac{u_{i_0-1}(u_{i_0+1} - u_{i_0})}{u_{i_0} - u_{i_0-1}}h_{i_0,j}$$

$$= \begin{cases} 2\theta_{i_0+1}^2 h_{i_0+1,j} - \frac{4\theta_{i_0-1}^2 \theta_{i_0+1}}{2\theta_{i_0}} h_{i_0,j} & 0 \leq i_0 \leq N - 2 \\ \theta_N^2 h_{i_0+1,j} - \frac{2\theta_{N-2}^2 \theta_N}{2\theta_{N-1}} h_{i_0,j} & i_0 = N - 1 \end{cases}$$

$$= 0$$

where the second equality comes from the induction hypothesis, and the third equality comes from (30). In sum, we proved $s_{ij} = 0$ for every $i$ and $j$, which implies $\mathbf{U} = 0$.

Next, we will claim that $t_{ij} = 0$ for all $i, j$. Firstly, explicit formula of $H_{\text{OGM-G}}(k + 1, i + 1)$ first. When $k > i$,

$$H_{\text{OGM-G}}(k + 1, i + 1) = \frac{\theta_{N-k} - 1}{\theta_{N-k+1}} \frac{\theta_{N-k+1} - 1}{\theta_{N-k+2}} \cdots \frac{\theta_{N-i-1} - 1}{\theta_{N-i}} \frac{2\theta_{N-k-1} - 1}{\theta_{N-k}}$$

$$= \frac{\theta_{N-k}^2 - \theta_{N-k}}{\theta_{N-k}\theta_{N-k+1}} \frac{\theta_{N-k+1}^2 - \theta_{N-k+1}}{\theta_{N-k+1}\theta_{N-k+2}} \cdots \frac{\theta_{N-i-1}^2 - \theta_{N-i-1}}{\theta_{N-i-1}\theta_{N-i}} \frac{2\theta_{N-k-1} - 1}{\theta_{N-k}}$$

$$= \frac{\theta_{N-k-1}^2}{\theta_{N-k}\theta_{N-k+1}} \frac{\theta_{N-k}^2}{\theta_{N-k+1}\theta_{N-k+2}} \cdots \frac{\theta_{N-i-2}^2}{\theta_{N-i-1}\theta_{N-i}} \frac{2\theta_{N-k-1} - 1}{\theta_{N-k}}$$

$$= \frac{\theta_{N-k-1}^2(2\theta_{N-k-1} - 1)}{\theta_{N-i-1}^2 \theta_{N-i}}.$$

To calculate $\{t_{i,j}\}$, it is enough to deal with the sum $\sum_{l=i}^{N-1} h_{l+1,j}$, which can be expressed as

$$\sum_{l=i}^{N-1} h_{l+1,j} = \begin{cases} \frac{\theta_N + 1}{2} & i = 0,\, j = i \\ \theta_{N-i} & i = 1,\ldots,N-1,\, j = i \\ \frac{\theta_{N-i-1}^4}{\theta_{N-j}\theta_{N-j-1}^2} & i = 1,\ldots,N-1,\, j = 0,\ldots,i-1 \end{cases}. \tag{37}$$

By inserting (37) in (35), $[t_{ij} = 0, \forall i, j]$ is obtained, which implies $\mathbf{V} = 0$. (37) and (35) are also stated in [29, Lemma 6.1].

### B.2.2 Calculation of energy function of (OBL-F$_\flat$) and (OBL-G$_\flat$)

First we calculate $\{s_{ij}\}$ for (OBL-F$_\flat$). Recall $u_i = \frac{(i+1)(i+2)}{2}$ for $0 \le i \le N-1$ and $u_N = \gamma^2 + \gamma$ where $\gamma = \sqrt{N(N+1)/2}$. When $j = i$,

$$\begin{aligned}
s_{i,i} &= \begin{cases} -\frac{1}{2}(u_N - u_{N-1})^2 + \frac{1}{2}u_N & i = N \\ -\frac{1}{2}(u_i - u_{i-1})^2 + u_i & 0 \le i \le N-1 \end{cases} \\
&= \begin{cases} \frac{\gamma}{2} = \frac{u_N - u_{N-1}}{2} & i = N \\ -\frac{1}{2}(i+1)^2 + \frac{(i+1)(i+2)}{2} = \frac{u_i - u_{i-1}}{2} & 0 \le i \le N-1 \end{cases}.
\end{aligned}$$

Now we claim that $s_{ij} = 0$ when $j \ne i$. In the case $j = i - 1$, we have

$$\begin{aligned}
s_{i,i-1} &= u_i h_{i,i-1} - u_{i-1} - (u_i - u_{i-1})(u_{i-1} - u_{i-2}) \\
&= \begin{cases} \frac{(i+1)(i+2)}{2}h_{i,i-1} - \frac{i(i+1)}{2} - (i+1)i & 0 \le i \le N-1 \\ (\gamma^2 + \gamma)h_{N,N-1} - \frac{N(N+1)}{2} - \gamma N & i = N \end{cases} \\
&= 0.
\end{aligned}$$

We show $s_{ij} = 0$ for $j \ne i$ with induction on $i$, i.e., proving

$$\left[ s_{i,j} = (u_i - u_{i-1})\sum_{l=j}^{i-1} h_{l+1,j} + u_{i-1}h_{i,j} - (u_i - u_{i-1})(u_j - u_{j-1}) = 0 \quad j = 0,\ldots,i-2 \right]. \tag{38}$$

(38) holds when $i = j + 2$ since

$$\begin{aligned}
& (u_{j+2} - u_{j+1})(h_{j+1,j} + h_{j+2,j}) + u_{j+1}h_{j+2,j} - (u_{j+2} - u_{j+1})(u_j - u_{j-1}) \\
=& (u_{j+2} - u_{j+1})h_{j+1,j} + u_{j+2}h_{j+2,j} - (u_{j+2} - u_{j+1})(u_j - u_{j-1}) \\
=& \begin{cases} (j+3)h_{j+1,j} + \frac{(j+3)(j+4)}{2}h_{j+2,j} - (j+3)(j+1) & 0 \le j \le N-3 \\ \gamma h_{N-1,N-2} + (\gamma^2 + \gamma)h_{N,N-2} - \gamma & j = N-2 \end{cases} \\
=& 0.
\end{aligned}$$

Assume (38) for $i = i_0$. For $i = i_0 + 1$,

$$(u_{i_0+1} - u_{i_0}) \sum_{l=j}^{i_0} h_{l+1,j} + u_{i_0} h_{i_0+1,j} - (u_{i_0+1} - u_{i_0})(u_j - u_{j-1})$$

$$= (u_{i_0+1} - u_{i_0}) \left( \sum_{l=j}^{i_0-1} h_{l+1,j} + h_{i_0+1,j} \right) + u_{i_0} h_{i_0+1,j} - (u_{i_0+1} - u_{i_0})(u_j - u_{j-1})$$

$$= (u_{i_0+1} - u_{i_0}) \left( \frac{(u_{i_0} - u_{i_0-1})(u_j - u_{j-1}) - u_{i_0-1} h_{i_0,j}}{u_{i_0} - u_{i_0-1}} + h_{i_0+1,j} \right) + u_{i_0} h_{i_0+1,j}$$

$$\quad - (u_{i_0+1} - u_{i_0})(u_j - u_{j-1})$$

$$= u_{i_0+1} h_{i_0+1,j} - \frac{u_{i_0-1}(u_{i_0+1} - u_{i_0})}{u_{i_0} - u_{i_0-1}} h_{i_0,j}$$

$$= \begin{cases} \frac{(i_0+2)(i_0+3)}{2} h_{i_0+1,j} - \frac{i_0(i_0+1)(i_0+2)}{2(i_0+1)} h_{i_0,j} & 0 \le i_0 \le N - 2 \\ \left( \gamma^2 + \gamma \right) h_{N,j} - \frac{(N-1)N\gamma}{2N} h_{N-1,j} & i_0 = N - 1 \end{cases}$$

$$= 0.$$

Next, we calculate $\{t_{ij}\}$ for (OBL-G$_\flat$). We need to deal with the sum $\sum_{l=k}^{N-1} h_{l+1,i}$, which can be expressed as

$$\sum_{l=i}^{N-1} h_{l+1,j} = \begin{cases} 1 + \frac{(N+2)(N-1)}{4(\gamma+1)} & i = 0, \ j = 0 \\ \frac{(N-i+2)(N-i+1)(N-i)(N-i-1)}{4(\gamma+1)N(N+1)} & i = 1, \dots, N-1, \ j = 0 \\ \frac{(N-i+2)(N-i+1)(N-i)(N-i-1)}{2(N-j)(N-j+1)(N-j+2)} & i = j+1, \dots, N-1, \ j = 1, \dots, N-1 \\ 1 + \frac{N-i-1}{2} & i = j, \ j = 1, \dots, N-1 \end{cases}.$$

By combining $v_0 = \frac{1}{\gamma^2 + \gamma}$, $v_i = \frac{1}{(N-i+1)(N-i+2)}$ for $1 \le i \le N$ and (35), we obtain

$$t_{ij} = \begin{cases} 1 & i = N, \ j = N \\ \frac{1}{2N(N+1)} - \frac{v_0}{2} & i = 0, \ j = 1 \\ v_0 - \frac{1}{N(N+1)} & i = N, \ j = 0 \\ \frac{1}{(N-i)(N-i+1)(N-i+2)} & i = 1, \dots, N-1, \ j = i \\ -\frac{2}{(N-i)(N-i+1)(N-i+2)} & i = N, \ j = 1, \dots, N-1 \\ 0 & \text{otherwise} \end{cases}$$

$$= \begin{cases} \frac{v_N}{2} & i = N, \ j = N \\ \frac{v_{i+1} - v_i}{2} & i = 0, \dots, N-1, \ j = i \\ -v_{i+1} + v_i & i = N, \ j = 0, \dots, N-1 \\ 0 & \text{otherwise} \end{cases}$$

Therefore,

$$\mathbf{V} = \sum_{0 \le j \le i \le N} \frac{t_{ij}}{L} \langle \nabla f(y_i), \nabla f(y_j) \rangle = \frac{v_0}{2L} \| \nabla f(y_N) \|^2 + \sum_{i=0}^{N-1} \frac{v_{i+1} - v_i}{2L} \| \nabla f(y_i) - \nabla f(y_N) \|^2 .$$

### B.2.3 Calculation of energy function of GD

We calculate $\{t_{ij}\}$ first. Recall that $\{v_i\}_{i=0}^N = \left(\frac{1}{2N+1}, \ldots, \frac{N+i}{(2N+1)(N-i+1)}, \ldots\right)$ and $h_{i+1,k} = \delta_{i,k}$ to (35), and making $\{t_{ij}\}$ symmetric gives us

$$
t_{ij} = \begin{cases}
\frac{1}{2} v_0 & i = j, \, i = 0 \\
v_i & i = j, \, 1 \le i \le N-1 \\
v_N - \frac{1}{2} & i = j, \, i = N \\
\frac{1}{2}\left(v_{\min(i,j)} - v_{\min(i,j)+1}\right) & i \ne j.
\end{cases}
$$

We can verify that the matrix $\{t_{ij}\}_{0 \le i,j \le N}$ is diagonally dominant: $t_{ii} = |\sum_{j \ne i} t_{ij}|$. Therefore, $\sum\limits_{0 \le i,j \le N-1} \frac{t_{ij}}{L} \langle \nabla f(y_i), \nabla f(y_j) \rangle \ge 0$ for any $\{\nabla f(y_i)\}_{i=0}^N$. This proof is essentially the same as the proof in [29], but we repeat it here with our notation for the sake of completeness.

Next, we prove that (GD) with $h = 1$ and $\{u_i\}_{i=0}^N = \left(\ldots, \frac{(2N+1)(i+1)}{2N-i}, \ldots, 2N+1\right)$ satisfies (C1), by showing more general statement:

$$
\left[\text{(GD) and } \{u_i\}_{i=0}^N = \left(\ldots, \frac{(2Nh+1)(i+1)}{2N-i}, \ldots, 2Nh+1\right) \text{ satisfies (C1)}\right] \tag{39}
$$

Note that (39) gives

$$
(2Nh+1)(f(x_N) - f^\star) \le \mathcal{U}_N \le \mathcal{U}_{-1} = \frac{L}{2}\|x_0 - x^\star\|^2. \tag{40}
$$

Later in the proof of Corollary 2, we will utilize the equation (39). The result (39) is proved in [19, Theorem 3.1], and we give the proof outline here.

In order to demonstrate (39), we will directly expand the expression $\mathcal{U}_N - u_N\left(f(x_N) - f^\star\right)$, instead of employing $\mathbf{U}$ as a intermediary step. Define $\{u_i'\}_{i=0}^N := \left(\ldots, \frac{(2N+1)(i+1)}{2N-i}, \ldots, 2N+1\right)$. Then

$$
\mathcal{U}_N - u_N\left(f(x_N) - f^\star\right) = \frac{1}{L}\operatorname{Tr}\left(g^\mathsf{T} g S\right)
$$

where $g = [\nabla f(x_0)|\ldots|\nabla f(x_N)|L(x_0 - x^\star)] \in \mathbb{R}^{d \times (N+2)}$ and $S \in \mathbb{S}^{N+2}$ is given by

$$
S = \begin{bmatrix} S' & \lambda \\ \lambda & \frac{1}{2} \end{bmatrix},
$$

$\lambda = [u_0|u_1 - u_0|\ldots|u_N - u_{N-1}]^\mathsf{T}$, $S' = \frac{2Nh+1}{2N+1}\left(hS_0 + (1-h)S_1\right)$,

$$
S_0 = \sum_{i=0}^{N-1} \frac{u_i'}{2}\left(e_{i+1}e_i^\mathsf{T} + e_i e_{i+1}^\mathsf{T} + (e_i - e_{i+1})(e_i - e_{i+1})^\mathsf{T}\right)
$$

$$
+ \sum_{i=0}^N \frac{u_i' - u_{i-1}'}{2}\left(e_i(e_0 + \cdots + e_{i-1})^\mathsf{T} + (e_0 + \cdots + e_{i-1})e_i^\mathsf{T} + e_i e_i^\mathsf{T}\right)
$$

$$
S_1 = \sum_{i=0}^{N-1} \frac{u_i'}{2}(e_i - e_{i+1})(e_i - e_{i+1})^\mathsf{T} + \sum_{i=0}^N \frac{u_i' - u_{i-1}'}{2} e_i e_i^\mathsf{T}.
$$

Now we will show $S \succeq 0$ to obtain $\left[\mathcal{U}_N - u_N\left(f(x_N) - f^\star\right) \ge 0, \forall g\right]$, which is (C1). By using Sylvester's Criterion, $S_0 \succ 0$ follows. $S_1 \succ 0$ follows from the fact that $S_1$ expressed by the sum of positive semi-definite matrices $zz^\mathsf{T}$. Since the convex sum of two positive semi-definite matrices is also positive semi-definite, $S' = hS_0 + (1-h)S_1 \succ 0$.

Next, we argue that $\det S = 0$. Indeed, take $\tau = (1, \ldots, -(2Nh+1))^\mathsf{T}$ to show $S\tau = 0$, which gives $\det S = 0$. Note that the determinant of $S$ can also be expressed by

$$
\det(S) = \left(\frac{1}{2} - \lambda^\mathsf{T}\left(S'\right)^{-1}\lambda\right)\det(S'). \tag{41}
$$

We have shown that $S' \succ 0$, (41) implies $\frac{1}{2} - \lambda^\mathsf{T}\left(S'\right)^{-1}\lambda = 0$, which is the Schur complement of the matrix $S$. By a well-known lemma on the Schur complement, we conclude $S \succeq 0$.

## B.3 Omitted proof in Section 2.5

### B.3.1 Omitted calculation of Corollary 1

Here, we will give the general formulation of H-dual FSFOM of

$$x_{k+1} = x_k + \beta_k \left( x_k^+ - x_{k-1}^+ \right) + \gamma_k \left( x_k^+ - x_k \right), \quad k = 0, \ldots, N - 1. \tag{42}$$

**Proposition 1.** The H-dual of (42) is

$$y_{k+1} = y_k + \beta_k' \left( y_k^+ - y_{k-1}^+ \right) + \gamma_k' \left( y_k^+ - y_k \right), \quad k = 0, \ldots, N - 1 \tag{43}$$

where

$$\beta_k' = \frac{\beta_{N-k}(\beta_{N-1-k} + \gamma_{N-1-k})}{\beta_{N-k} + \gamma_{N-k}}$$

$$\gamma_k' = \frac{\gamma_{N-k}(\beta_{N-1-k} + \gamma_{N-1-k})}{\beta_{N-k} + \gamma_{N-k}}$$

for $k = 0, \ldots, N - 1$ and $(\beta_N, \gamma_N)$ is any value that $\beta_N + \gamma_N \neq 0$. [3]

*Proof.* The $H$ matrix $\{h_{k,i}\}_{0 \leq i < k \leq N}$ satisfies

$$h_{k+1,i} = \begin{cases} 1 + \beta_k + \gamma_k & i = k, \ k = 0, \ldots, N - 1 \\ \beta_k \left( h_{k,i} - 1 \right) & i = k - 1, \ k = 1, \ldots, N - 1 \\ \beta_k h_{k,i} & i = k - 2, \ k = 2, \ldots, N - 1 \end{cases}.$$

Therefore,

$$h_{k+1,i} = \left( \prod_{j=i+1}^{k} \beta_j \right) (\beta_i + \gamma_i + \delta_{k,i})$$

where $\delta_{k,i}$ is a Kronecker Delta function. Similarly, $H$ matrix of (43) $\{g_{k,i}\}_{0 \leq i < k \leq N}$ satisfies

$$\begin{aligned} g_{k+1,i} &= \left( \prod_{j=i+1}^{k} \beta_j' \right) (\beta_i' + \gamma_i' + \delta_{k,i}) \\ &= \left( \prod_{j=i+1}^{k} \frac{\beta_{N-j}(\beta_{N-1-j} + \gamma_{N-1-j})}{\beta_{N-j} + \gamma_{N-j}} \right) (\beta_{N-1-i} + \gamma_{N-1-i} + \delta_{k,i}) \\ &= \left( \prod_{j=i+1}^{k} \beta_{N-j} \right) (\beta_{N-k-1} + \gamma_{N-i-1} + \delta_{N-k-1,N-i-1}). \end{aligned}$$

Thus $g_{k+1,i} = h_{N-i,N-1-k}$. $\qquad \square$

Now we derive the H-dual of (6) by applying Proposition 1. Note that

$$\beta_k = \frac{(T_k - t_k)t_{k+1}}{t_k T_{k+1}}, \qquad \gamma_k = \frac{(t_k^2 - T_k)t_{k+1}}{t_k T_{k+1}}, \quad k = 0, \ldots, N - 1.$$

Next, define $\beta_N$ and $\gamma_N$ as a same form of $\{\beta_i, \gamma_i\}_{i=0}^{N-1}$ with any $t_{N+1} > 0$. Note that

$$\beta_k + \gamma_k = \frac{t_{k+1}(t_k^2 - t_k)}{t_k T_{k+1}} = \frac{t_{k+1}(t_k - 1)}{T_{k+1}}.$$

By applying the formula at Proposition 1, we obtain

$$\beta_k' = \frac{(T_{N-k} - t_{N-k})\frac{t_{N-k}(t_{N-k-1}-1)}{T_{N-k}}}{t_{N-k}^2 - t_{N-k}} = \frac{(T_{N-k} - t_{N-k})(t_{N-k-1} - 1)}{T_{N-k}(t_{N-k} - 1)} = \frac{T_{N-k-1}(t_{N-k-1} - 1)}{T_{N-k}(t_{N-k} - 1)}$$

and

$$\gamma_k' = \frac{(t_{N-k}^2 - T_{N-k})\frac{t_{N-k}(t_{N-k-1}-1)}{T_{N-k}}}{t_{N-k}^2 - t_{N-k}} = \frac{(t_{N-k}^2 - T_{N-k})(t_{N-k-1} - 1)}{T_{N-k}(t_{N-k} - 1)}.$$

---

[3]Here, note that FSFOM (43) is independent with the any choice of $\beta_N, \gamma_N$ since $y_1 = y_0 + (\beta_0' + \gamma_0') \left( y_0^+ - y_0 \right) = y_0 + (\beta_{N-1} + \gamma_{N-1}) \left( y_0^+ - y_0 \right)$.

### B.3.2 Proof of Corollary 2

First, recall (39) when $0 < h \le 1$:

$$\left[\text{(GD) and } \{u_i\}_{i=0}^N = \left(\dots, \frac{(2Nh+1)(i+1)}{2N-i}, \dots, 2Nh+1\right) \text{ satisfies (C1)}\right]$$

Additionally, observe that the $H$ matrix of (GD) is $\text{diag}(h, \dots, h)$, which gives the fact that the H-dual of (GD) is itself.

Next, use Theorem 1 to obtain

$$\left[\text{(GD) with } 0 < h \le 1 \text{ and } \{v_i\}_{i=0}^N = \left(\frac{1}{2Nh+1}, \dots, \frac{N+i}{(2Nh+1)(N-i+1)}, \dots\right) \text{ satisfies (C2)}\right].$$
(44)

By using the same argument with (5), (44) gives

$$\frac{1}{2L}\|\nabla f(y_N)\|^2 \le \mathcal{V}_N \le \mathcal{V}_0 = \frac{1}{2Nh+1}(f(y_0)-f_\star) + \frac{1}{2Nh+1}[\![y_N, y_\star]\!] \le \frac{1}{2Nh+1}(f(y_0)-f_\star).$$
(45)

In addition, we can achieve the convergence rate of the gradient norm under the initial condition of $\|x_0 - x_\star\|^2$:

$$\frac{1}{2L}\|\nabla f(x_{2N})\|^2 \le \frac{1}{2Nh+1}(f(x_N)-f_\star) \le \frac{1}{(2Nh+1)^2}\frac{L}{2}\|x_0 - x_\star\|^2$$

and

$$\frac{1}{2L}\|\nabla f(x_{2N+1})\|^2 \le \frac{1}{2(N+1)h+1}(f(x_N)-f_\star) \le \frac{1}{(2(N+1)h+1)(2Nh+1)}\frac{L}{2}\|x_0 - x_\star\|^2.$$

The first inequality comes from (45) and the second inequality comes from (40).

### B.3.3 Proof of $\mathcal{A}_\star$-optimality of (OGM-G) and (OBL-G$_\flat$)

**Definition of $\mathcal{A}^\star$-optimal FSFOM.** For the given inequality sets $\mathcal{L}$, $\mathcal{A}^\star$-optimal FSFOM with respect to $[\mathcal{L}, \text{P1}]$ is defined as a FSFOM which $H$ matrix is the solution of following minimax problem:

$$\begin{aligned}
\underset{H \in \mathbb{R}_L^{N \times N}}{\text{minimize}} \ \underset{f}{\text{maximize}} \ & f(x_N) - f_\star \\
\text{subject.to. } & [x_0, \dots, x_N \text{ are generated by FSFOM with the matrix } H] \\
& [\forall\, l \in \mathcal{L}, f \text{ satisfies } l] \\
& \|x_0 - x_\star\|^2 \le R^2
\end{aligned}$$
(46)

Similarly, $\mathcal{A}^\star$-optimal FSFOM with respect to $[\mathcal{L}, (P2)]$ is specified with its $H$ matrix, which is the solution of the following minimax problem:

$$\begin{aligned}
\underset{H \in \mathbb{R}_L^{N \times N}}{\text{minimize}} \ \underset{f}{\text{maximize}} \ & \frac{1}{2L}\|\nabla f(y_N)\|^2 \\
\text{subject.to. } & [y_0, \dots, y_N \text{ are generated by FSFOM with the matrix } H] \\
& [\forall\, l \in \mathcal{L}, f \text{ satisfies } l] \\
& f(y_0) - f_\star \le \frac{1}{2}LR^2
\end{aligned}$$
(47)

Here $\mathbb{R}_L^{N \times N}$ is the set of lower triangular matrices. Next we denote the inner maximize problem of (46) and (47) as $\mathcal{P}_1(\mathcal{L}, H, R)$ and $\mathcal{P}_2(\mathcal{L}, H, R)$, respectively. For a more rigorous definition of $\mathcal{A}^\star$-optimal FSFOM, refer [44].

**Remark.** We discuss the minimax problems (46) and (47) and their interpretation. Specifically, we consider the meaning of the maximization problem in these minimax problems, which can be thought of as calculating the worst-case performance for a fixed FSFOM with $H$. In other words, the minimization problem in (46) and (47) can be interpreted as determining the optimal value of $H$ that minimizes the worst-case performance.

**Prior works.** The prior works for $\mathcal{A}^\star$-optimality are summarized as follows. Consider the following sets of inequalities.

$$
\begin{aligned}
\mathcal{L}_{\mathrm{F}} =& \{[\![x_i, x_{i+1}]\!]\}_{i=0}^{N-1} \cup \{[\![x_\star, x_i]\!]\}_{i=0}^{N} \\
=& \left\{ f(x_i) \geq f(x_{i+1}) + \langle \nabla f(x_{i+1}), x_i - x_{i+1}\rangle + \frac{1}{2L}\|\nabla f(x_i) - \nabla f(x_{i+1})\|^2 \right\}_{i=0}^{N-1} \\
& \bigcup \left\{ f_\star \geq f(x_i) + \langle \nabla f(x_i), x_\star - x_i\rangle + \frac{1}{2L}\|\nabla f(x_i)\|^2 \right\}_{i=0}^{N}, \\
\mathcal{L}_{\mathrm{G}} =& \{[\![y_i, y_{i+1}]\!]\}_{i=0}^{N-1} \cup \{[\![y_N, y_i]\!]\}_{i=0}^{N-1} \cup \{[\![y_N, \star]\!]\} \\
=& \left\{ f(y_i) \geq f(y_{i+1}) + \langle \nabla f(y_{i+1}), y_i - y_{i+1}\rangle + \frac{1}{2L}\|\nabla f(y_i) - \nabla f(y_{i+1})\|^2 \right\}_{i=0}^{N-1} \\
& \bigcup \left\{ f(y_N) \geq f(y_i) + \langle \nabla f(y_i), y_N - y_i\rangle + \frac{1}{2L}\|\nabla f(y_i) - \nabla f(y_N)\|^2 \right\}_{i=0}^{N-1} \\
& \bigcup \left\{ f(y_N) \geq f_\star + \frac{1}{2L}\|\nabla f(y_N)\|^2 \right\}, \\
\mathcal{L}_{\mathrm{G}'} =& \{[\![y_i, y_{i+1}]\!]\}_{i=0}^{N-1} \cup \left\{[\![y_N, y_i]\!] - \frac{1}{2L}\|\nabla f(y_i) - \nabla f(y_N)\|^2 \right\}_{i=0}^{N-1} \cup \left\{ [\![y_N, \star]\!] - \frac{1}{2L}\|\nabla f(y_N)\|^2 \right\} \\
=& \left\{ f(y_i) \geq f(y_{i+1}) + \langle \nabla f(y_{i+1}), y_i - y_{i+1}\rangle + \frac{1}{2L}\|\nabla f(y_i) - \nabla f(y_{i+1})\|^2 \right\}_{i=0}^{N-1} \\
& \bigcup \left\{ f(y_N) \geq f(y_i) + \langle \nabla f(y_i), y_N - y_i\rangle \right\}_{i=0}^{N-1} \bigcup \left\{ f(y_N) \geq f_\star \right\}, \\
\mathcal{L}_{\mathrm{F}'} =& \{[\![x_i, x_{i+1}]\!]\}_{i=0}^{N-1} \cup \left\{ [\![x_\star, x_i]\!] - \frac{1}{2L}\|\nabla f(x_i)\|^2 \right\}_{i=0}^{N} \\
=& \left\{ f(x_i) \geq f(x_{i+1}) + \langle \nabla f(x_{i+1}), x_i - x_{i+1}\rangle + \frac{1}{2L}\|\nabla f(x_i) - \nabla f(x_{i+1})\|^2 \right\}_{i=0}^{N-1} \\
& \bigcup \left\{ f_\star \geq f(x_i) + \langle \nabla f(x_i), x_\star - x_i\rangle \right\}_{i=0}^{N},
\end{aligned}
$$

and

$$
\mathcal{L}_{\mathrm{exact}} = \{[\![x_i, x_j]\!]\}_{(i,j)\in\{\star,0,1,\ldots,N\}^2}.
$$

(OGM) is $\mathcal{A}^\star$-optimal with respect to $[\mathcal{L}_{\mathrm{F}}, (\mathrm{P1})]$ [26]. Furthermore, (OGM) is also $\mathcal{A}^\star$-optimal under $[\mathcal{L}_{\mathrm{exact}}, (\mathrm{P1})]$ which implies that (OGM) is the exact optimal FSFOM with respect to (P1). In addition, the $\mathcal{A}^\star$-optimality of (OGM-G) with respect to $[\mathcal{L}_{\mathrm{G}}, (\mathrm{P2})]$ was presented as a conjecture in [29], (OBL-F$_\flat$) is $\mathcal{A}^\star$-optimal with respect to $[\mathcal{L}_{\mathrm{F}'}, (P1)]$ [44, Theorem 4], and the $\mathcal{A}^\star$-optimality of (OBL-G$_\flat$) with respect to $[\mathcal{L}_{\mathrm{G}'}, (\mathrm{P2})]$ was presented as a conjecture [44, Conjecture 8]. [4]

In the remaining parts, we give the proof of the following two theorems.

**Theorem 4.** (OGM-G) is $A^\star$-optimal with respect to $[\mathcal{L}_{\mathrm{G}}, (\mathrm{P2})]$.

**Theorem 5.** (OBL-G$_\flat$) is is $A^\star$-optimal with respect to $[\mathcal{L}_{\mathrm{G}'}, (\mathrm{P2})]$.

---

[4]Originally, the inequality set suggested that (OBL-G$_\flat$) would $\mathcal{A}^\star$-optimal is which $(f(y_N) \geq f_\star)$ is replaced with $\left(f(y_N) \geq f_\star + \frac{1}{2L}\|\nabla f(y_N)\|^2\right)$ in $\mathcal{L}_{\mathrm{G}'}$.

**Proof of $\mathcal{A}^\star$-optimality of** (OGM-G)    We provide an alternative formulation of $\mathcal{P}_2(\mathcal{L}_G, H, R)$ by using the methodology called PEP [19, 52]

$$
\begin{aligned}
\underset{f}{\text{maximize}} \quad & \frac{1}{2L}\|\nabla f(y_N)\|^2 \\[1mm]
\text{subject to} \quad & f(y_{i+1}) - f(y_i) + \langle \nabla f(y_{i+1}), y_i - y_{i+1}\rangle + \frac{1}{2L}\|\nabla f(y_i) - \nabla f(y_{i+1})\|^2 \leq 0, \quad i = 0,\dots,N-1 \\[1mm]
& f(y_i) - f(y_N) + \langle \nabla f(y_i), y_N - y_i\rangle + \frac{1}{2L}\|\nabla f(y_i) - \nabla f(y_N)\|^2 \leq 0, \quad i = 0,\dots,N-1 \\[1mm]
& f_\star + \frac{1}{2L}\|\nabla f(y_N)\|^2 \leq f(y_N) \\[1mm]
& f(y_0) - f_\star \leq \frac{1}{2}LR^2 \\[1mm]
& y_{i+1} = y_i - \frac{1}{L}\sum_{j=0}^{i} h_{i+1,j}\nabla f(y_j), \quad i = 0,\dots,N-1
\end{aligned}
$$

$(48)$

Next, define $\mathbf{F}$, $\mathbf{G}$, $\{\mathbf{e}_i\}_{i=0}^N$ and $\{\mathbf{x}_i\}_{i=0}^N$ as

$$
\mathbf{G} := \begin{pmatrix}
\langle \nabla f(y_0), \nabla f(y_0)\rangle & \langle \nabla f(y_0), \nabla f(y_1)\rangle & \cdots & \langle \nabla f(y_0), \nabla f(y_N)\rangle \\
\vdots & \vdots & \vdots & \\
\langle \nabla f(y_N), \nabla f(y_N)\rangle & \langle \nabla f(y_N), \nabla f(y_1)\rangle & \cdots & \langle \nabla f(y_N), \nabla f(y_N)\rangle
\end{pmatrix} \in \mathbb{S}^{N+1},
$$

$$
\mathbf{F} := \begin{pmatrix}
f(y_0) - f_\star \\
f(y_1) - f_\star \\
\vdots \\
f(y_N) - f_\star
\end{pmatrix} \mathbb{R}^{(N+1)\times 1},
$$

$\mathbf{e}_i \in \mathbb{R}^{N+1}$ is a unit vector which $(i+1)$-th component is 1, and

$$
\mathbf{y}_0 := 0, \qquad \mathbf{y}_{i+1} := \mathbf{y}_i - \frac{1}{L}\sum_{j=0}^{i} h_{i+1,j}\mathbf{e}_j \quad i = 0,\dots,N-1.
$$

By using the definition of $\mathbf{F}$, $\mathbf{G}$, $\{\mathbf{e}_i\}_{i=0}^N$ and $\{\mathbf{y}_i\}_{i=0}^N$, $\mathcal{P}_2(\mathcal{L}_G, H, R)$ can be converted into

$$
\begin{aligned}
\underset{\mathbf{F}, \mathbf{G}\succeq 0}{\text{minimize}} \quad & -\frac{1}{2L}\,\mathrm{Tr}\left(\mathbf{G}\mathbf{e}_N\mathbf{e}_N^\mathsf{T}\right) \\[1mm]
\text{subject to} \quad & \mathbf{F}(\mathbf{e}_{i+1} - \mathbf{e}_i)^\mathsf{T} + \mathrm{Tr}\left(\mathbf{G}\mathbf{A}_i\right) \leq 0, \quad i = 0,\dots,N-1 \\[1mm]
& \mathbf{F}(\mathbf{e}_i - \mathbf{e}_N)^\mathsf{T} + \mathrm{Tr}\left(\mathbf{G}\mathbf{B}_i\right) \leq 0, \quad i = 0,\dots,N-1 \\[1mm]
& -\mathbf{F}\mathbf{e}_N^\mathsf{T} + \frac{1}{2}\,\mathrm{Tr}\left(\mathbf{G}\mathbf{e}_N\mathbf{e}_N^\mathsf{T}\right) \leq 0 \\[1mm]
& \mathbf{F}\mathbf{e}_0^\mathsf{T} - \frac{1}{2}LR^2 \leq 0
\end{aligned}
$$

$(49)$

where

$$
\mathbf{A}_i := \frac{1}{2}\mathbf{e}_{i+1}(\mathbf{y}_i - \mathbf{y}_{i+1})^\mathsf{T} + \frac{1}{2}(\mathbf{y}_i - \mathbf{y}_{i+1})\mathbf{e}_{i+1}^\mathsf{T} + \frac{1}{2L}\left(\mathbf{e}_i - \mathbf{e}_{i+1}\right)\left(\mathbf{e}_i - \mathbf{e}_{i+1}\right)^\mathsf{T}
$$

$$
\mathbf{B}_i := \frac{1}{2}\mathbf{e}_i(\mathbf{y}_N - \mathbf{y}_i)^\mathsf{T} + \frac{1}{2}(\mathbf{y}_N - \mathbf{y}_i)\mathbf{e}_i^\mathsf{T} + \frac{1}{2L}\left(\mathbf{e}_i - \mathbf{e}_N\right)\left(\mathbf{e}_i - \mathbf{e}_N\right)^\mathsf{T}.
$$

Moreover, under the condition $d \geq N + 2$, we can take the Cholesky factorization of $\mathbf{G}$ to recover the triplet $\{(y_i, f(y_i), \nabla f(y_i))\}_{i=0}^N$.[5] Thus (48) and (49) are equivalent. The next step is calculating the Lagrangian of (49) and deriving the Lagrangian dual problem of it. In [54], they argued about the strong duality of (49).

---

[5]Cholesky factorization is unique if $\mathbf{G} > 0$ but it may be not unique if $\mathbf{G} \succeq 0$. In this case, we choose one representation of Cholesky factorization.

**Fact 1.** Assume $h_{i+1,i} \neq 0$ for $0 \leq i \leq N-1$. Then the strong duality holds between (48) and (49).

Denote the dual variables of each constraints as $\{\delta_i\}_{i=0}^{N-1}$, $\{\lambda_i\}_{i=0}^{N-1}$, $\delta_N$ and $\tau$. Then Lagrangian becomes

$$\mathcal{L}(\delta, \lambda, \tau, \mathbf{F}, \mathbf{G}) = \mathbf{F} \cdot \mathbf{X}^\intercal + \text{Tr}\left(\mathbf{G} \cdot \mathbf{T}\right) - \frac{\tau L}{2} R^2$$

where

$$\mathbf{X} = \sum_{i=0}^{N-1} \delta_i(\mathbf{e}_{i+1} - \mathbf{e}_i) + \sum_{i=0}^{N-1} \lambda_i(\mathbf{e}_i - \mathbf{e}_N) - \delta_N \mathbf{e}_N + \tau \mathbf{e}_0,$$

$$\mathbf{T} = \sum_{i=0}^{N-1} \delta_i \mathbf{A}_i + \sum_{i=0}^{N-1} \lambda_i \mathbf{B}_i + \frac{\delta_N}{2} \mathbf{e}_N \mathbf{e}_N^\intercal - \frac{1}{2} \mathbf{e}_N \mathbf{e}_N^\intercal.$$

If $[\mathbf{X} = 0$ and $\mathbf{T} \succeq 0]$ is false, we can choose $\mathbf{F}$ and $\mathbf{G}$ that makes the value of Lagrangian to be $-\infty$. Thus the convex dual problem of (49) is

$$\underset{\delta, \lambda, \tau}{\text{maximize}} \; \underset{\mathbf{F}, \mathbf{G}}{\text{minimize}} \; \mathcal{L}(\delta, \lambda, \tau, \mathbf{F}, \mathbf{G}) = \begin{cases} \underset{\delta, \lambda, \tau}{\text{maximize}} & -\frac{\tau L}{2} R^2 \\ \text{subject to} & \mathbf{X} = 0, \; \mathbf{T} \succeq 0 \\ & \delta_i \geq 0, \quad \lambda_i \geq 0, \quad \tau \geq 0 \end{cases}. \tag{50}$$

For the constraint $\mathbf{X} = 0$, $\lambda_i = \delta_i - \delta_{i-1}$ for $1 \leq i \leq N-1$, $\tau = \delta_N$ and $-\delta_0 + \lambda_0 + \delta_N = 0$. By substituting $v_{i+1} = \delta_i$ for $0 \leq i \leq N-1$ and $v_0 = \delta_N$, (50) becomes

$$\underset{v_0, \ldots, v_N}{\text{maximize}} \; \underset{\mathbf{F}, \mathbf{G}}{\text{minimize}} \; \mathcal{L}(\delta, \lambda, \tau, \mathbf{F}, \mathbf{G}) = \begin{cases} \underset{\delta, \lambda, \tau}{\text{maximize}} & -\frac{v_0 L}{2} R^2 \\ \text{subject to} & \mathbf{T} \succeq 0 \\ & 0 \leq v_0 \leq \cdots \leq v_N \end{cases}. \tag{51}$$

Therefore if strong duality holds, $\mathcal{P}_2(\mathcal{L}_\mathbf{G}, H, R)$ becomes

$$\underset{v_0, \ldots, v_N}{\text{minimize}} \; \frac{v_0 L}{2} R^2$$
$$\text{subject.to. } \mathbf{T} \succeq 0, \; 0 \leq v_0 \leq \cdots \leq v_N. \tag{52}$$

We can apply a similar argument for $\mathcal{P}_1(\mathcal{L}_\mathbf{F}, H, R)$. To begin with, define $\{\mathbf{f}_i\}_{i=-1}^N$ as a unit vector of length $N+2$ which $(i+2)$-component is 1. Additionally, define $\{\mathbf{x}_i\}_{i=0}^N$, $\{\mathbf{C}_i\}_{i=0}^{N-1}$ and $\{\mathbf{D}_i\}_{i=0}^N$ as follows:

$$\mathbf{x}_0 := \mathbf{f}_{-1}, \quad \mathbf{x}_{i+1} := \mathbf{x}_i - \frac{1}{L}\sum_{j=0}^i h_{i+1,j}\mathbf{f}_j \quad i = 0, \ldots, N-1,$$

$$\mathbf{C}_i := \frac{1}{2}\mathbf{f}_{i+1}(\mathbf{x}_i - \mathbf{x}_{i+1})^\intercal + \frac{1}{2}(\mathbf{x}_i - \mathbf{x}_{i+1})\mathbf{f}_{i+1}^\intercal + \frac{1}{2L}\left(\mathbf{f}_i - \mathbf{f}_{i+1}\right)\left(\mathbf{f}_i - \mathbf{f}_{i+1}\right)^\intercal, \tag{53}$$

$$\mathbf{D}_i := -\frac{1}{2}\mathbf{f}_i\mathbf{x}_i^\intercal - \frac{1}{2}\mathbf{x}_i\mathbf{f}_i^\intercal + \frac{1}{2L}\mathbf{f}_i\mathbf{f}_i^\intercal.$$

Then, if the strong duality holds, the problem

$$\underset{f}{\text{maximize}} \; f(x_N) - f_\star$$

$$\text{subject.to. } [x_0, \ldots, x_N \text{ are generated by FSFO with the matrix } H]$$
$$[\forall l \in \mathcal{L}_\mathbf{F}, f \text{ satisfies } l] \tag{54}$$
$$\|x_0 - x_\star\|^2 \leq R^2$$

is equivalent to

$$\underset{u_0, \ldots, u_N}{\text{maximize}} \; \frac{L}{2u_N} R^2$$
$$\text{subject.to. } \mathbf{S} \succeq 0, \; 0 \leq u_0 \leq \cdots \leq u_N, \tag{55}$$

where

$$\mathbf{S} = \frac{L}{2}\mathbf{f}_{-1}\mathbf{f}_{-1}^{\mathsf{T}} + \sum_{i=0}^{N-1} u_i \mathbf{C}_i + \sum_{i=0}^{N}(u_i - u_{i-1})\mathbf{D}_i.$$

By using Schur's Complement,

$$\mathbf{S} \succeq 0 \quad \Leftrightarrow \quad \mathbf{S}' \succeq 0$$

where

$$\mathbf{S}' = \sum_{i=0}^{N-1} u_i \mathbf{C}_i + \sum_{i=0}^{N}(u_i - u_{i-1})\mathbf{D}_i - \frac{1}{2L}\left(\sum_{i=0}^{N}(u_i - u_{i-1})\mathbf{f}_i\right)\left(\sum_{i=0}^{N}(u_i - u_{i-1})\mathbf{f}_i\right)^{\mathsf{T}}.$$

Hence (55) is equivalent to

$$\begin{aligned} \underset{u_0,\dots,u_N}{\text{maximize}} \quad & \frac{L}{2u_N}R^2 \\ \text{subject.to. } & \mathbf{S}' \succeq 0, \ 0 \le u_0 \le \cdots \le u_N. \end{aligned} \tag{56}$$

Now we will prove the following proposition.

**Proposition 2.** Consider a matrix $H = \{h_{i,j}\}_{0 \le j < i \le N}$ and $H^A$. If $h_{i+1,i} \ne 0$ for $0 \le i \le N-1$ and treating the solution of infeasible maximize problem as 0, the optimal values of $\mathcal{P}_1(\mathcal{L}_{\mathrm{F}}, H, R)$ and $\mathcal{P}_2(\mathcal{L}_{\mathrm{G}}, H^A, R)$ are same.

*Proof.* To simplify the analysis, we can consider $\{\mathbf{f}_i\}_{i=0}^N$ as a length $N+1$ unit vector, as all terms with $\mathbf{f}_{-1}$ can be eliminated by using $\mathbf{S}'$ instead of $\mathbf{S}$. With this simplification, both $\mathbf{S}'$ and $\mathbf{T}$ belong to $\mathbb{S}^{N+1}$.

Next, let $0 < u_0 \le u_1 \le \cdots \le u_N$ and $v_i = \frac{1}{u_{N-i}}$ for $0 \le i \le N$, noting that $0 < v_0 \le \cdots \le v_N$. It is important to observe that $\mathbf{S}'$ and $\mathbf{T}$ can be expressed in terms of $\mathcal{S}(H, u)$ and $\mathcal{T}(H^A, v)$, respectively. Furthermore, in the proof of Theorem 1 (A), we proved that $\mathcal{S}(H, u) = \mathcal{M}(u)^{\mathsf{T}}\mathcal{T}(H^A, v)\mathcal{M}(u)$ for some invertible $\mathcal{M}(u)$. Thus,

$$\mathcal{S}(H, u) \succeq 0 \quad \Leftrightarrow \quad \mathcal{T}(H^A, v) \succeq 0.$$

Therefore, we obtain

$$(a_0, \dots, a_N) \in \{(u_0, \dots, u_N) | \mathbf{S}' \succeq 0, 0 < u_0 \le \cdots \le u_N\} \tag{57}$$

if and only if

$$\left(\frac{1}{a_N}, \dots, \frac{1}{a_0}\right) \in \{(v_0, \dots, v_N) | \mathbf{T} \succeq 0, 0 < v_0 \le \cdots \le v_N\}. \tag{58}$$

For the next step, we claim that the optimal value of (56) and

$$\begin{aligned} \underset{u_0,\dots,u_N}{\text{maximize}} \quad & \frac{L}{2u_N}R^2 \\ \text{subject.to. } & \mathbf{S}' \succeq 0, \ 0 < u_0 \le \cdots \le u_N \end{aligned} \tag{59}$$

are same, i.e., consider the case when all $u_i$ are positive is enough. To prove that, assume there exists $0 = u_0 = \cdots = u_k$, $0 < u_{k+1} \le \cdots \le u_N$ and $H$ that satisfy $\mathbf{S}' \succeq 0$. Next, observe that the $\mathbf{f}_k\mathbf{f}_k^{\mathsf{T}}$ component of $\mathbf{S}'$ is 0 but $\mathbf{f}_{k+1}\mathbf{f}_k^{\mathsf{T}}$ component of $\mathbf{S}'$ is $u_{k+1}h_{k+1,k} \ne 0$, which makes $\mathbf{S}' \succeq 0$ impossible.

When the optimal value of (52) is 0, it implies $\{(v_0, \dots, v_N) | \mathbf{T} \succeq 0, 0 < v_0 \le \cdots \le v_N\}$ is an empty set. Therefore, $\{(u_0, \dots, u_N) | \mathbf{S}' \succeq 0, 0 < u_0 \le \cdots \le u_N\}$ is also empty set. Since (59) and (56) have the same optimal value, the optimal value of (56) is 0.

If the optimal value of (52) is positive, the optimal values of (56) and (52) are the same since (57) and (58) are equivalent. $\qquad\square$

*Proof of Theorem 4.* $H_{\text{OGM}}$ is the solution of (46) since (OGM) is $\mathcal{A}^\star$-optimal with respect to $\mathcal{L}_\text{F}$ and (P1). Additionally, if

$$(46) = \underset{H, h_{i+1,i} \neq 0}{\text{maximize}} \ \mathcal{P}_1(\mathcal{L}_\text{F}, H, R) \tag{60}$$

holds, $H_{\text{OGM}}$ is the solution to the following problem due to the strong duality.

$$\underset{H, h_{i+1,i} \neq 0}{\text{maximize}} \ \underset{u_0, \ldots, u_N}{\text{maximize}} \ \frac{L}{2u_N} R^2 \tag{61}$$
$$\text{subject.to. } \mathbf{S} \succeq 0, \ 0 \leq u_0 \leq \cdots \leq u_N.$$

Applying Proposition 2 and using the fact $H_{\text{OGM-G}} = H_{\text{OGM}}^A$ provides that $H_{\text{OGM-G}}$ is the solution of

$$\underset{H, h_{i+1,i} \neq 0}{\text{maximize}} \ \underset{v_0, \ldots, v_N}{\text{minimize}} \ \frac{v_0 L}{2} R^2 \tag{62}$$
$$\text{subject.to. } \mathbf{T} \succeq 0, \ 0 \leq v_0 \leq \cdots \leq v_N.$$

Finally, if

$$(47) = \underset{H, h_{i+1,i} \neq 0}{\text{maximize}} \ \mathcal{P}_1(\mathcal{L}_\text{F}, H, R) \tag{63}$$

holds, the optimal solution of (47) is the $H_{\text{OGM-G}}$, which proves the $\mathcal{A}^\star$-optimality of (OGM-G) with respect to $\mathcal{L}_\text{G}$ and (P2). The proof of (60) and (63) uses the continuity argument with $H$ and please refer [44, Claim 4]. □

**Remark of Proof of $\mathcal{A}^\star$-optimality of** (OGM-G). We proved that (OGM-G) is $\mathcal{A}^\star$-optimal if we use the subset of cocoercivity inequalities. Therefore, it is still open whether (OGM-G) is optimal or not among the $L$-smooth convex function's gradient minimization method.

**Proof of $\mathcal{A}^\star$-optimality of** (OBL-G$_\flat$). We provide the proof that the $H$ matrix of (OBL-G$_\flat$) is the solution of

$$\underset{H \in \mathbb{R}_L^{N \times N}}{\text{minimize}} \ \mathcal{P}_2(\mathcal{L}_{\text{G}'}, H, R) \tag{64}$$

To begin with, bring to mind the $\mathcal{A}^\star$-optimality of (OBL-F$_\flat$): (OBL-F$_\flat$) is $\mathcal{A}^\star$-optimal with respect to the $[\mathcal{L}_{\text{F}'}, P1]$, i.e., the $H$ matrix of (OBL-F$_\flat$) is the solution of

$$\underset{H \in \mathbb{R}_L^{N \times N}}{\text{minimize}} \ \mathcal{P}_1(\mathcal{L}_{\text{F}'}, H, R). \tag{65}$$

To prove the $\mathcal{A}^\star$-optimality of (OBL-G$_\flat$), we use the $\mathcal{A}^\star$-optimality of (OBL-F$_\flat$).

Under the assumption of strong duality, we could change $\mathcal{P}_1(\mathcal{L}_{\text{F}'}, H, R)$ into the following SDP:

$$\underset{u_0, \ldots, u_N}{\text{maximize}} \ \frac{L}{2u_N} R^2 \tag{66}$$
$$\text{subject.to. } \mathbf{S}_1 \succeq 0, \ 0 \leq u_0 \leq \cdots \leq u_N$$

where

$$\mathbf{S}_1 = \frac{L}{2} \mathbf{f}_{-1} \mathbf{f}_{-1}^\mathsf{T} + \sum_{i=0}^{N-1} u_i \mathbf{C}_i + \sum_{i=0}^{N} (u_i - u_{i-1}) \left( \mathbf{D}_i - \frac{1}{2L} \mathbf{f}_i \mathbf{f}_i^\mathsf{T} \right).$$

Here we used the same notation with (53). Each $\frac{1}{2L} \mathbf{f}_i \mathbf{f}_i^\mathsf{T}$ term is subtracted from the original $\mathbf{S}$ since we consider the inequality $[\![x_i, x_\star]\!] - \frac{1}{2L} \|\nabla f(x_i)\|^2$ instead of $[\![x_i, x_\star]\!]$ for $\mathcal{L}_{\text{F}'}$. Moreover, (66) is equivalent to

$$\underset{u_0, \ldots, u_N}{\text{maximize}} \ \frac{L}{2u_N} R^2 \tag{67}$$
$$\text{subject.to. } \mathbf{S}_1' \succeq 0, \ 0 \leq u_0 \leq \cdots \leq u_N$$

where

$$\mathbf{S}'_1 = \sum_{i=0}^{N-1} u_i \mathbf{C}_i + \sum_{i=0}^{N} (u_i - u_{i-1}) \left( \mathbf{D}_i - \frac{1}{2L} \mathbf{f}_i \mathbf{f}_i^\mathsf{T} \right) - \frac{1}{2L} \left( \sum_{i=0}^{N} (u_i - u_{i-1}) \mathbf{f}_i \right) \left( \sum_{i=0}^{N} (u_i - u_{i-1}) \mathbf{f}_i \right)^\mathsf{T}.$$

Similarly, under the assumption of strong duality, $\mathcal{P}_2(\mathcal{L}_{\mathrm{G}'}, H, R)$ is equivalent to

$$\begin{aligned} &\underset{v_0,\ldots,v_N}{\text{minimize}} && \frac{v_0 L}{2} R^2 \\ &\text{subject.to.} && \mathbf{T}_1 \succeq 0, \ \ 0 \le v_0 \le \cdots \le v_N \end{aligned} \tag{68}$$

where

$$\mathbf{T}_1 = \sum_{i=0}^{N-1} v_{i+1} \mathbf{A}_i + \sum_{i=0}^{N-1} (v_{i+1} - v_i) \left( \mathbf{B}_i - \frac{1}{2L} (\mathbf{e}_i - \mathbf{e}_N)(\mathbf{e}_i - \mathbf{e}_N)^\mathsf{T} \right) - \frac{1}{2} \mathbf{e}_N \mathbf{e}_N^\mathsf{T}.$$

Now we will prove the following proposition.

**Proposition 3.** Consider a matrix $H = \{h_{k,i}\}_{0 \le i < k \le N}$ and $H^A$. If $h_{i+1,i} \ne 0$ for $0 \le i \le N-1$ and treating the solution of infeasible maximize problem as 0, the optimal values of $\mathcal{P}_1(\mathcal{L}_{\mathrm{F}'}, H, R)$ and $\mathcal{P}_2(\mathcal{L}_{\mathrm{G}'}, H^A, R)$ are same.

*Proof.* The proof structure is the same as the proof of Proposition 2. First consider $\{\mathbf{f}_i\}_{i=0}^{N}$ as length $N+1$, which gives $\mathbf{S}'_1, \mathbf{T}_1 \in \mathbb{S}^{N+1}$. Furthermore, in the proof of Appendix A, we proved that

$$\mathbf{S}'_1 = \mathcal{M}(u)^\mathsf{T} \mathbf{T}_1 \mathcal{M}(u).$$

Therefore,

$$\mathbf{S}'_1 \succeq 0 \quad \Leftrightarrow \quad \mathbf{T} \succeq 0.$$

The other steps are the same as the proof of Proposition 2. $\square$

*Proof of Theorem 5.* $H_{\mathrm{OGM}}$ is the solution of (46) since (OGM) is $\mathcal{A}^\star$-optimal with respect to $\mathcal{L}_{\mathrm{F}}$ and (P1). Additionally, if

$$(46) = \underset{H, h_{i+1,i} \ne 0}{\text{maximize}} \ \mathcal{P}_1(\mathcal{L}_{\mathrm{F}}, H, R) \tag{69}$$

holds, $H_{\mathrm{OGM}}$ is the solution to the following problem due to the strong duality.

$$\begin{aligned} &\underset{H, h_{i+1,i} \ne 0}{\text{maximize}} \ \underset{u_0,\ldots,u_N}{\text{maximize}} && \frac{L}{2u_N} R^2 \\ &\text{subject.to.} && \mathbf{S} \succeq 0, \ \ 0 \le u_0 \le \cdots \le u_N. \end{aligned} \tag{70}$$

Applying Proposition 2 and using the fact $H_{\mathrm{OGM\text{-}G}} = H^A_{\mathrm{OGM}}$ provides that $H_{\mathrm{OGM\text{-}G}}$ is the solution of

$$\begin{aligned} &\underset{H, h_{i+1,i} \ne 0}{\text{maximize}} \ \underset{v_0,\ldots,v_N}{\text{minimize}} && \frac{v_0 L}{2} R^2 \\ &\text{subject.to.} && \mathbf{T} \succeq 0, \ \ 0 \le v_0 \le \cdots \le v_N. \end{aligned} \tag{71}$$

Finally, if

$$(47) = \underset{H, h_{i+1,i} \ne 0}{\text{maximize}} \ \mathcal{P}_1(\mathcal{L}_{\mathrm{F}}, H, R) \tag{72}$$

holds, the optimal solution of (47) is the $H_{\mathrm{OGM\text{-}G}}$, which proves the $\mathcal{A}^\star$-optimality of (OGM-G) with respect to $\mathcal{L}_{\mathrm{G}}$ and (P2). The proof of (69) and (72) uses the continuity argument with $H$ and please refer [44, Claim 4]. $\square$

## C    Omitted parts in Section 3

### C.1    Omitted calculations in Section 3.1

Strictly speaking, (7) is not a differential equation but rather a diffeo-integral equation. However, if $H$ is separable, i.e., when $H(t,s) = e^{\beta(s)-\gamma(t)}$ for some $\beta, \gamma \colon [0,T) \to \mathbb{R}$, then (7) can be reformulated as an ODE.

Assume the process (7) is well-defined. We can alternatively write (7) as

$$X(0) = x_0, \quad \dot{X}(t) = -e^{-\gamma(t)} \int_0^t e^{\beta(s)} \nabla f(X(s)) ds. \tag{73}$$

By multiplying $e^{\gamma(t)}$ each side and differentiating, we obtain

$$\ddot{X}(t) + \dot{\gamma}(t) \dot{X}(t) + e^{\beta(t)-\gamma(t)} \nabla f(X(t)) = 0.$$

The H-dual of (73) is

$$Y(0) = x_0, \quad \dot{Y}(t) = -e^{\beta(T-t)} \int_0^t e^{-\gamma(T-s)} \nabla f(Y(s)) ds.$$

Under the well-definedness, by multiplying $e^{-\beta(T-t)}$ each side and differentiating, we obtain

$$\ddot{Y}(t) + \dot{\beta}(T-t) \dot{Y}(t) + e^{\beta(T-t)-\gamma(T-t)} \nabla f(Y(t)) = 0.$$

When $\beta(s) = \gamma(t) = r \log t$, two ODEs become

$$\ddot{X}(t) + \frac{r}{t} \dot{X}(t) + \nabla f(X(t)) = 0 \tag{74}$$

and

$$\ddot{Y}(t) + \frac{r}{T-t} \dot{X}(t) + \nabla f(X(t)) = 0, \tag{75}$$

which are introduced in Section 3.2.

For the calculations of energy functions of (74) and (75), refer [51, Section 3.1, Section 4.2]. They proved that

$$(r-1) \|X(0) - x_\star\|^2 = T^2 \left( f(X(T)) - x_\star \right) + \frac{1}{2} \left\| T\dot{X}(T) + 2(X(T) - x_\star) \right\|^2$$

$$+ (r-3) \|X(T) - x_\star\|^2 + \int_0^T (r-3)s \left\| \dot{X}(s) \right\|^2 ds - \int_0^T 2s[X(s), x_\star] ds$$

holds for (74) and

$$\frac{1}{T^2} \left( f(Y(0)) - f(Y(T)) \right) + \frac{-r+3}{T^4} \|Y(0) - Y(T)\|^2$$

$$= \frac{1}{4(r-1)} \|\nabla f(Y(T))\|^2 + \int_0^T \frac{r-3}{(T-s)^5} \left\| (T-s)\dot{Y}(s) + 2(Y(s) - Y(T)) \right\|^2 ds - \int_0^T \frac{2}{(T-s)^3} [Y(s), Y(T)] ds$$

holds for (75). After arranging terms, we obtain the results in Section 3.2. [6]

### C.2    Proof of Theorem 2

To begin with, we give a formal version of Theorem 2. Consider two following conditions.

$$u(T) \left( f(X(T)) - f^\star \right) \le \mathcal{U}(T), \quad (\forall X(0), x^\star, \{\nabla f(X(s))\}_{s \in [0,T]} \in \mathcal{A}_1). \tag{C3$'$}$$

$$\frac{1}{2} \|\nabla f(Y(T))\|^2 \le \mathcal{V}(T), \quad (\forall Y(0), \{\nabla f(Y(s))\}_{s \in [0,T]} \in \mathcal{A}_2). \tag{C4$'$}$$

where $\mathcal{A}_1$ and $\mathcal{A}_2$ are family of vectors which makes Fubini's Theorem can be applied and will be defined later in this section.

---

[6]In [51], they considered the ODE which gradient term is $2\nabla f(Y(t))$ instead of $\nabla f(Y(t))$.

**Theorem 6** (Formal version of Theorem 2). Assume the C-FSFOMs (7) with $H$ and $H^A$ are well-defined in the sense that solutions to the diffeo-integral equations exist. Consider differentiable functions $u, v \colon (0, T) \to \mathbb{R}$ that $v(t) = \frac{1}{u(T-t)}$ for $t \in [0, T]$ and

(i) $\lim_{s \to 0} u(t) = 0$

(ii) $\lim_{s \to -T} v(s) \left( f(Y(s)) - f(Y(T)) + \langle \nabla f(Y(T)), Y(T) - Y(s) \rangle \right) = 0$.

(iii) $f$ is $L$-smooth and convex.

Then the following holds.

$$\left[ (C3') \text{ is satisfied with } u(\cdot) \text{ and } H \right] \Leftrightarrow \left[ (C4') \text{ is satisfied with } v(\cdot) \text{ and } H^A \right].$$

**Calculations of energy functions via transformation** Frist of all, we calculate $\mathcal{U}(T) - u(T)(f(X(T)) - f^\star)$.

$$\mathcal{U}(T) - u(T) \left( f(X(T)) - f^\star \right)$$

$$= \frac{1}{2} \|X(0) - x^\star\|^2 + \int_0^T u'(s) \left( f(X(s)) - f^\star + \langle \nabla f(X(s)), x^\star - X(s) \rangle \right) ds - u(T) \left( f(X(T)) - f^\star \right) ds$$

$$= \frac{1}{2} \|X(0) - x^\star\|^2 + \int_0^T u'(s) \langle \nabla f(X(s)), x^\star - X(s) \rangle ds - \int_0^T u(s) \left\langle \nabla f(X(s)), \dot{X}(s) \right\rangle ds$$

$$= \frac{1}{2} \|X(0) - x^\star\|^2 + \int_0^T u'(s) \langle \nabla f(X(s)), x^\star - X(0) \rangle ds$$

$$+ \int_0^T u'(s) \langle \nabla f(X(s)), X(0) - X(s) \rangle ds - \int_0^T u(s) \left\langle \nabla f(X(s)), \dot{X}(s) \right\rangle ds$$

$$= \frac{1}{2} \left\| X(0) - x^\star - \int_0^T u'(s) \nabla f(X(s)) ds \right\|^2 - \frac{1}{2} \left\| \int_0^T u'(s) \nabla f(X(s)) ds \right\|^2$$

$$+ \int_0^T u'(s) \langle \nabla f(X(s)), X(0) - X(s) \rangle ds - \int_0^T u(s) \left\langle \nabla f(X(s)), \dot{X}(s) \right\rangle ds.$$

We used parts of integration and $u(0) \colon = \lim_{s \to 0} u(s) = 0$. Since $X(0) - x^\star$ can have any value, (C3) is equivalent to

$$- \frac{1}{2} \left\| \int_0^T u'(s) \nabla f(X(s)) ds \right\|^2$$

$$+ \int_0^T u'(s) \langle \nabla f(X(s)), X(0) - X(s) \rangle ds - \int_0^T u(s) \left\langle \nabla f(X(s)), \dot{X}(s) \right\rangle ds \geq 0 \quad (76)$$

holds for any $\{\nabla f(X(s))\}_{s \in [0,T]}$.

Now define a transformation $\{f_s \in \mathbb{R}^d\}_{s \in [0,T]} \to \{g_s \in \mathbb{R}^d\}_{s \in [0,T]}$ as

$$g_s \colon = u(s) f_s + \int_s^T u'(z) f_z dz, \quad s \in [0, T] \quad (77)$$

and $\{g_s \in \mathbb{R}^d\}_{s \in [0,T]} \to \{f_s \in \mathbb{R}^d\}_{s \in [0,T]}$.

$$f_s \colon = \frac{1}{u(T)} g_0 + \frac{1}{u(s)} (g_s - g_0) - \int_s^T \frac{u'(b)}{u(b)^2} (g_b - g_0) db, \quad s \in [0, T]. \quad (78)$$

One can show that the above two transformations are in the inverse relationship. Next, we calculate $\mathcal{U}(T)$. Define $f_s \colon = \nabla f(X(s))$. Under the transformation (77), we can find a simple expression of

(76).

$$
(76) = -\frac{1}{2}\|g_0\|^2 - \int_0^T u'(s)\langle f_s, X(s) - X(0)\rangle\, ds - \int_0^T u(s)\left\langle f_s, \dot{X}(s)\right\rangle ds
$$

$$
= -\frac{1}{2}\|g_0\|^2 - \int_0^T u'(s)\left\langle f_s, \int_0^s \dot{X}(a)da\right\rangle ds + \int_0^T u(s)\left\langle f_s, \int_0^s \dot{X}(a)da\right\rangle ds
$$

$$
= -\frac{1}{2}\|g_0\|^2 - \int_0^T u'(s)\left\langle f_s, \int_0^s \dot{X}(a)da\right\rangle ds + \int_0^T\int_0^s H(s,a)\langle u(s)f_s, f_a\rangle\, dads \quad (79)
$$

$$
\overset{(\circ)}{=} -\frac{1}{2}\|g_0\|^2 - \int_0^T\left\langle \dot{X}(a), \int_a^T u'(s)f_s ds\right\rangle da + \int_0^T\int_0^s H(s,a)\langle u(s)f_s, f_a\rangle\, dads
$$

$$
= -\frac{1}{2}\|g_0\|^2 + \int_0^T\int_0^s H(s,a)\langle f_a, g_s\rangle\, dads.
$$

We used Fubini's Theorem at $(\circ)$. Next we calculate $\mathcal{V}(T)$.

$$
\mathcal{V}(T) = \frac{1}{u(T)}\left(f(Y(0)) - f(Y(T))\right) + \int_0^T \frac{d}{ds}\frac{1}{u(T-s)}\left(f(Y(s)) - f(Y(T)) + \langle\nabla f(Y(s)), Y(T) - Y(s)\rangle\right) ds
$$

$$
= \frac{1}{u(T)}\left(f(Y(0)) - f(Y(T))\right) + \int_0^T \frac{u'(T-s)}{u(T-s)^2}\langle Y(T) - Y(s), \nabla f(Y(s)) - \nabla f(Y(T))\rangle\, ds
$$

$$
+ \int_0^T \frac{d}{ds}\frac{1}{u(T-s)}\left(f(Y(s)) - f(Y(T)) + \langle\nabla f(Y(T)), Y(T) - Y(s)\rangle\right) ds
$$

$$
\overset{(\circ)}{=} \frac{1}{u(T)}\langle\nabla f(Y(T)), Y(0) - Y(T)\rangle + \int_0^T \frac{u'(T-s)}{u(T-s)^2}\langle Y(T) - Y(s), \nabla f(Y(s)) - \nabla f(Y(T))\rangle\, ds
$$

$$
- \int_0^T \frac{1}{u(T-s)}\left\langle \dot{Y}(s), \nabla f(Y(s)) - \nabla f(Y(T))\right\rangle ds.
$$

$(\circ)$ comes from parts of integration and the assumption

$$
\lim_{s\to T} v(s)\left(f(Y(s)) - f(Y(T)) + \langle\nabla f(X(T)), Y(T) - Y(s)\rangle\right) = 0.
$$

To clarify, $u'(T-s) = u'(z)|_{z=T-s}$. Now briefly write $g_s := \nabla f(Y(T-s))$. Then

$$
\mathcal{V}(T) - \frac{1}{2}\|\nabla f(Y(T))\|^2
$$

$$
= -\frac{1}{2}\|g_0\|^2 - \frac{1}{u(T)}\int_0^T\left\langle g_0, \dot{Y}(s)\right\rangle ds + \int_0^T \frac{u'(T-s)}{u(T-s)^2}\langle Y(T) - Y(s), g_{T-s} - g_0\rangle\, ds
$$

$$
- \int_0^T \frac{1}{u(T-s)}\left\langle \dot{Y}(s), g_{T-s} - g_0\right\rangle ds
$$

$$
= -\frac{1}{2}\|g_0\|^2 + \int_0^T \frac{u'(T-s)}{u(T-s)^2}\left\langle \int_s^T \dot{Y}(a)da, g_{T-s} - g_0\right\rangle ds
$$

$$
- \frac{1}{u(T)}\int_0^T\left\langle g_0, \dot{Y}(s)\right\rangle ds - \int_0^T \frac{1}{u(T-s)}\left\langle \dot{Y}(s), g_{T-s} - g_0\right\rangle ds
$$

$$
\overset{(\circ)}{=} -\frac{1}{2}\|g_0\|^2 + \int_0^T\left\langle -\frac{1}{u(T)}g_0 - \frac{1}{u(T-s)}(g_{T-s} - g_0) + \int_0^s \frac{u'(T-b)}{u(T-b)^2}(g_{T-b} - g_0)\, db, \dot{Y}(s)\right\rangle ds.
$$

$$
\tag{80}
$$

We use Fubini's Theorem at $(\circ)$. Finally, using (78), we obtain

$$\mathcal{V}(T) - \frac{1}{2}\|\nabla f(Y(T))\|^2$$

$$= -\frac{1}{2}\|g_0\|^2 - \int_0^T \left\langle f_{T-s}, \dot{Y}(s) \right\rangle ds$$

$$= -\frac{1}{2}\|g_0\|^2 + \int_0^T \left\langle f_{T-s}, \int_0^s H^A(s,a)g_{T-a}da \right\rangle ds \qquad (81)$$

$$= -\frac{1}{2}\|g_0\|^2 + \int_0^T \left\langle f_{T-s}, \int_0^s H(T-a,T-s)g_{T-a}da \right\rangle ds$$

$$= -\frac{1}{2}\|g_0\|^2 + \int_0^T \int_0^s H(s,a) \left\langle f_a, g_s \right\rangle dads.$$

**Proof of Theorem 2**  Define $\mathcal{A}_1$ and $\mathcal{A}_2$ as follows.

$$\mathcal{A}_1 = \{\{f_s\}_{s\in[0,T]}|\text{Analysis in the previous paragraph holds}\},$$
$$\mathcal{A}_2 = \{\{g_s\}_{s\in[0,T]}|\text{Analysis in the previous paragraph holds}\}.$$

Now we prove Theorem 2. We have shown that (C3′) is equivalent to (76) $\geq 0$ for all $\{\nabla f(X(s))\}_{s\in[0,T]} \in \mathcal{A}_1$. By definition of $\mathcal{A}_1$, it is equivalent to

$$-\frac{1}{2}\|g_0\|^2 + \int_0^T \int_0^s H(s,a) \left\langle f_a, g_s \right\rangle dads \geq 0$$

for any $\{g_s\}_{s\in[0,T]} \in \mathcal{A}_2$. Moreover, by (81), it is also equivalent to

$$-\frac{1}{2}\|g_0\|^2 + \int_0^T \int_0^s H(s,a) \left\langle f_a, g_s \right\rangle dads = \mathcal{V}(T) - \frac{1}{2}\|\nabla f(Y(T))\|^2 \geq 0$$

for any $\{g_s\}_{s\in[0,T]} \in \mathcal{A}_2$, which is (C4′).

### C.3    Omitted parts in Section 3.3

**Regularity of** (10) **at** $t = -T$.    To begin with, note that ODE (10) can be expressed as

$$\begin{cases} \dot{W}(t) = -\frac{2p-1}{T-t}W(t) - Cp^2(T-t)^{p-2}\nabla f(Y(t)) \\ \dot{Y}(t) = W(t) \end{cases}$$

for $t \in (0,T)$. Since right hand sides are Lipschitz continuous with respect to $W$ and $Y$ in any closed interval $[0,s] \in [0,T)$, solution $(Y,W)$ uniquely exists that satisfies above ODE with initial condition $(Y(0),W(0)) = (y_0,0)$. Next, we give the proof of regularity of ODE (10) at terminal time $T$ in the following order. The proof structure is based on the regularity proof in [51]:

(i)  $\sup_{t\in[0,T)} \left\|\dot{Y}(t)\right\|$ is bounded

(ii)  $Y(t)$ can be continuously extended to $T$

(iii)  $\lim_{t\to T_-} \left\|\dot{Y}(t)\right\| = 0$

(iv)  $\lim_{t\to T_-} \frac{\dot{Y}(t)}{(T-t)^{p-1}} = Cp\nabla f(Y(T)).$

**Step (i):** $\sup_{t\in[0,T)} \left\|\dot{Y}(t)\right\|$ **is bounded.**    ODE (10) is equivalent to

$$\frac{1}{(T-s)^{p-2}}\ddot{Y}(s) + \frac{2p-1}{(T-s)^{p-1}}\dot{Y}(s) + Cp^2\nabla f(Y(s)) = 0. \qquad (82)$$

By multiplying $\dot{Y}(s)$ and integrating from $0$ to $t$, we obtain

$$\int_0^t \frac{1}{(T-s)^{p-2}} \left\langle \ddot{Y}(s), \dot{Y}(s) \right\rangle ds + \int_0^t \frac{2p-1}{(T-s)^{p-1}} \left\| \dot{Y}(s) \right\|^2 ds + Cp^2 \int_0^t \left\langle \dot{Y}(s), \nabla f(Y(s)) \right\rangle ds = 0$$

and integration by parts gives us

$$\frac{1}{2(T-t)^{p-2}} \left\| \dot{Y}(t) \right\|^2 - \frac{1}{2T^{p-2}} \left\| \dot{Y}(0) \right\|^2 + \int_0^t \frac{2p-1}{(T-s)^{p-1}} \left\| \dot{Y}(s) \right\|^2 ds + Cp^2 \left( f(Y(t)) - f(Y(0)) \right) = 0.$$

Define $\Psi(t)\colon [0, T) \to \mathbb{R}$ as

$$\Psi(t) = \frac{1}{2(T-t)^{p-2}} \left\| \dot{Y}(t) \right\|^2 + Cp^2 \left( f(Y(t)) - f(Y(0)) \right).$$

Since

$$\dot{\Psi}(t) = -\frac{2p-1}{(T-s)^{p-1}} \left\| \dot{Y}(s) \right\|^2 ds,$$

$\Psi(t)$ is a nonincreasing function. Thus

$$\left\| \dot{Y}(t) \right\|^2 = 2(T-t)^{p-2} \left( \Psi(t) + Cp^2 \left( f(Y(0)) - f(Y(t)) \right) \right) \leq 2T^{p-2} \left( \Psi(0) + Cp^2 \left( f(Y(0)) - f_\star \right) \right),$$
(83)

and the right hand side of (83) is constant, which implies $M := \sup_{t \in [0,T)} \left\| \dot{Y}(t) \right\| < \infty$.

**Step (ii):** $Y(t)$ **can be continuously extended to** $T$. We can prove $Y(t)$ is uniformly continuous due to the following analysis.

$$\| Y(t+\delta) - Y(\delta) \| = \left\| \int_t^{t+\delta} \dot{Y}(s) ds \right\| \leq \int_t^{t+\delta} \left\| \dot{Y}(s) \right\| ds \leq \delta M.$$

Since a uniformly continuous function $g\colon D \to \mathbb{R}^d$ can be extended continuously to $\overline{D}$, $Y\colon [0, T) \to \mathbb{R}^d$ can be extended to $[0, T]$.

**Step (iii):** $\lim_{t \to T-} \left\| \dot{Y}(t) \right\| = 0$. We first prove the limit $\lim_{t \to T-} \left\| \dot{Y}(t) \right\| = 0$ exists. From C.3, we know $\lim_{t \to T-} Y(t)$ exists and by continuity of $f$, $\lim_{t \to T-} f(Y(t))$ also exists. Moreover, $\Psi(t)$ is non-increasing and

$$\Psi(t) = \frac{1}{2(T-t)^{p-2}} \left\| \dot{Y}(t) \right\|^2 + Cp^2 \left( f(Y(t)) - f(Y(0)) \right) \geq Cp^2 \left( f_\star - f(Y(0)) \right),$$

thus $\lim_{t \to T-} \Psi(t)$ exists. Therefore, $\lim_{t \to T-} \frac{\| \dot{Y}(t) \|^2}{(T-t)^{p-2}}$ exists, which implies $\lim_{t \to T-} \left\| \dot{Y}(t) \right\| = 0$ when $p > 2$. For the case $p = 2$, $\lim_{t \to T-} \left\| \dot{Y}(t) \right\|$ exists, and $\lim_{t \to T-} \left\| \dot{Y}(t) \right\| = 0$ follows since we have

$$\Psi(t) = \frac{1}{2T^p} \left\| \dot{Y}(0) \right\|^2 - \int_0^t \frac{3}{(T-s)^{p-1}} \left\| \dot{Y}(s) \right\|^2 ds$$

is bounded below. Thus the value of integration is finite.

**Step (iv):** $\lim_{t \to T-} \frac{\dot{Y}(t)}{(T-t)^{p-1}} = Cp \nabla f(Y(T))$. The (7) form of process $Y$ is

$$\dot{Y}(t) = -\int_0^t \frac{Cp^2(T-t)^{2p-1}}{(T-s)^{p+1}} \nabla f(Y(s)) ds.$$

By dividing $(T-t)^{p-1}$ each side, we obtain

$$\frac{\dot{Y}(t)}{(T-t)^{p-1}} = -(T-t)^p \int_0^t \frac{Cp^2}{(T-s)^{p+1}} \nabla f(Y(s)) ds.$$

By the result of C.3, we can apply L'Hopital's rule, which gives

$$\lim_{t\to T-} \frac{\dot{Y}(t)}{(T-t)^{p-1}} = -\lim_{t\to T-} \frac{\int_0^t \frac{Cp^2}{(T-s)^{p+1}} \nabla f(Y(s)) ds}{(T-t)^{-p}} = Cp\nabla f(Y(T)).$$

Also, since $\nabla f$ is $L$-l=Lipshitz, $\|\nabla f(Y(t)) - \nabla f(Y(T))\| \le L\|Y(t) - Y(T)\|$. Therefore,

$$\lim_{t\to T-} \frac{\nabla f(Y(t)) - \nabla f(Y(T))}{(T-t)^\beta} = 0 \tag{84}$$

for any $\beta < p$.

**Applying Theorem 2 to** (10).   For the case $p = 2$, refer [51]. Now we consider the case $p > 2$, with $u(t) = Ct^p$ and $v(t) = \frac{1}{C(T-t)^p}$. First, we verify conditions (i) and (ii) in Theorem 6. (i) holds since $u(0) = 0$, and (ii) holds since

$$\lim_{s\to T-} v(s)\left(f(Y(s)) - f(Y(T)) + \langle \nabla f(Y(T)), Y(T) - Y(s)\rangle\right)$$

$$= \frac{1}{C} \lim_{s\to T-} \frac{f(Y(s)) - f(Y(T)) + \langle \nabla f(Y(T)), Y(T) - Y(s)\rangle}{(T-s)^p}$$

$$\overset{(\circ)}{=} \frac{1}{C} \lim_{s\to T-} \frac{\left\langle \dot{Y}(s), \nabla f(Y(s)) - \nabla f(Y(T))\right\rangle}{-p(T-s)^{p-1}}$$

$$\overset{(\bullet)}{=} -\frac{1}{pC} \lim_{s\to T-} \left\langle \frac{\dot{Y}(s)}{(T-s)^{p-1}}, \nabla f(Y(s)) - \nabla f(Y(T))\right\rangle$$

$$= 0.$$

($\circ$) uses L'Hospital's rule and ($\bullet$) uses the limit results at the previous paragraph.

To prove

$$\frac{1}{2}\|\nabla f(Y(T))\|^2 \le \frac{1}{CT^p}\left(f(Y(0)) - f(Y(T))\right),$$

we carefully check that for any $L$-smooth convex function $f$, $\{\nabla f(Y(s))\}_{s\in[0,T]} \in \mathcal{A}_2$.

**Verification of** (80).   Fubini's Theorem is used for

$$\int_{0\le s,a\le T} \mathbf{1}_{s\le a} \frac{u'(T-s)}{u(T-s)^2}\left\langle \dot{Y}(a), (\nabla f(Y(s)) - \nabla f(Y(T)))\right\rangle$$

$$= \int_{0\le s,a\le T} \mathbf{1}_{s\le a}\dot{Y}(a)\left(\nabla f(Y(s)) - \nabla f(Y(T))\right)\frac{p}{C(T-s)^{p+1}}.$$

The above function is continuous in $0 \le s, a \le T$ due to $\lim_{t\to T-} \frac{\dot{Y}(t)}{(T-t)^{p-1}} = Cp\nabla f(Y(T))$ and (84) with $\beta = 2 < p$.

**Verification of** (79).   First,

$$\lim_{t\to T-} \frac{Y(t) - Y(T)}{(T-t)^p} = \lim_{t\to T-} \frac{\dot{Y}(t)}{-p(T-t)^{p-1}} = -C\nabla f(Y(T))$$

holds from (iv). Thus $\sup_{b\in(0,T]}\left\|\frac{Y(T-b)-Y(T)}{b^p}\right\| < \infty$. Now we will show

$$\lim_{a\to+0} a^{1+\epsilon} f_a = 0, \quad \forall \epsilon > 0.$$

Observe

$$f_a = \frac{1}{CT^p} g_0 + \frac{1}{CT^p} (g_a - g_0) - \int_a^T \frac{p}{Cb^{p+1}} (g_b - g_a) \, db$$

and

$$\left\| a^{1+\epsilon} f_a \right\| = \left\| \frac{a^{1+\epsilon}}{CT^p} g_0 + \frac{a^{1+\epsilon}}{CT^p} (g_a - g_0) - \int_a^T \frac{pa^{1+\epsilon}}{Cb^{p+1}} (g_b - g_a) \, db \right\|$$

$$\leq \left\| \frac{a^{1+\epsilon}}{CT^p} g_0 + \frac{a^{1+\epsilon}}{CT^p} (g_a - g_0) \right\| + \int_a^T \frac{pa^{1+\epsilon}}{Cb^{p+1}} \|g_b - g_0\| \, db + \int_a^T \frac{pa^{1+\epsilon}}{Cb^{p+1}} \|g_a - g_0\| \, db$$

$$\leq \left\| \frac{a^{1+\epsilon}}{CT^p} g_0 + \frac{a^{1+\epsilon}}{CT^p} (g_a - g_0) \right\| + \int_a^T \frac{Lpa^{1+\epsilon}}{Cb^{p+1}} \|Y(T-b) - Y(T)\| \, db + \int_a^T \frac{Lpa^{1+\epsilon}}{Cb^{p+1}} \|Y(T-a) - Y(T)\| \, db$$

$$\leq \left\| \frac{a^{1+\epsilon}}{CT^p} g_0 + \frac{a^{1+\epsilon}}{CT^p} (g_a - g_0) \right\| + \int_a^T Lpa^{\epsilon} \left\| \frac{Y(T-b) - Y(T)}{b^p} \right\| \, db$$

$$+ \frac{La^{1+\epsilon}}{C} \left( \frac{1}{a^p} - \frac{1}{T^p} \right) \|Y(T-a) - Y(T)\| .$$

By using the boundness of $\frac{Y(T-b)-Y(T)}{b^p}$, we obtain the desired result.

Next,

$$\left\| a^{2\epsilon - p + 2} \dot{X}(a) \right\| = \left\| \int_0^a a^{\epsilon - p + 2} H(a, b) f_b \, db \right\|$$

$$\leq \int_0^a \frac{Cp^2 b^{2p-1}}{a^{2p-1-2\epsilon}} \|f_b\| \, db$$

$$= \int_0^a \frac{Cp^2 b^{2p-2-\epsilon}}{a^{2p-1-2\epsilon}} \left\| b^{1+\epsilon} f_b \right\| \, db$$

$$= \left( \sup_{b \in [0,a]} \left\| b^{1+\epsilon} f_b \right\| \right) \frac{Cp^2}{2p - 1 - \epsilon} a^{\epsilon},$$

which gives

$$\lim_{a \to +0} a^{2\epsilon - p + 2} \dot{X}(a) = 0, \quad \forall \, 0 < \epsilon < 2p - 1.$$

For the final step, recall that Fubini's Theorem is used for

$$\int_{0 \leq a, s \leq T} \mathbf{1}_{a \leq s} s^{p-1} \left\langle f_s, \dot{X}(a) \right\rangle \, da \, ds.$$

To prove that the above function is continuous, take any $0 < \epsilon < \frac{p-2}{2}$. Then

$$\lim_{0 \leq a \leq s, s \to +0} \left\| s^{p-1} \left\langle f_s, \dot{X}(a) \right\rangle \right\| \leq \lim_{0 \leq a \leq s, s \to +0} \left\| \frac{s^{-2\epsilon + 2p - 3}}{a^{-2\epsilon + p - 2}} \left\langle \dot{X}(a), f_s \right\rangle \right\| = \lim_{s \to +0} \left\| s^{-2\epsilon + 2p - 3} f_s \right\| \cdot \lim_{a \to +0} \left\| \frac{\dot{X}(a)}{a^{-2\epsilon + p - 2}} \right\| = 0.$$

which gives the desired result.

# D   Omitted calculation in Section 4

## D.1   Preliminaries

**General proximal step and composite FSFOM**   For $a > 0$, we newly define a prox-grad step with step size $\frac{1}{L} \frac{1}{\alpha}$ as follows:

$$y^{\oplus, \alpha} := \underset{z \in \mathbb{R}^d}{\operatorname{argmin}} \left( f(y) + \langle \nabla f(y), z - y \rangle + g(z) + \frac{\alpha L}{2} \|z - y\|^2 \right). \tag{85}$$

First, we will provide a generalized version of prox-grad inequality; Prox-grad inequality is originally used to prove the convergence of FISTA as below [11]:

$$F(y^{\oplus}) - L \left\langle y^{\oplus, 1} - y, x - y^{\oplus, 1} \right\rangle \leq F(x) + \frac{L}{2} \|y^{\oplus, 1} - y\|^2.$$

**Proposition 4** (Generalized version of prox-grad inequality). For every $x, y \in \mathbb{R}^d$, the following holds:

$$(\!(x, y)\!)^\alpha := F(y^{\oplus,\alpha}) - F(x^{\oplus,\alpha}) - La \left\langle y^{\oplus,\alpha} - y, x^{\oplus,\alpha} - y^{\oplus,\alpha} \right\rangle - \frac{L}{2} \|y^{\oplus,\alpha} - y\|^2 \le 0.$$

*Proof.* The optimality condition of (85) provides

$$L\alpha(y^{\oplus,\alpha} - y) + \nabla f(y) + u = 0, \quad u \in \partial g(y^{\oplus,\alpha}). \tag{86}$$

In addition, $L$-smoothness of $f$ and convexity inequality of $f$ and $g$ give

$$F(y^{\oplus,\alpha}) \le f(y) + \left\langle \nabla f(y), y^{\oplus,\alpha} - y \right\rangle + \frac{L}{2}\|y^{\oplus,\alpha} - y\|^2 + g(y^{\oplus,\alpha})$$

$$g(y^{\oplus,\alpha}) + \left\langle u, x^{\oplus,\alpha} - y^{\oplus,\alpha} \right\rangle = g(y^{\oplus,\alpha}) + \left\langle -L\alpha(y^{\oplus,\alpha} - y) - \nabla f(y), x^{\oplus,\alpha} - y^{\oplus,\alpha} \right\rangle \le g(x^{\oplus,\alpha}),$$

$$f(y) + \left\langle \nabla f(y), x^{\oplus,\alpha} - y \right\rangle \le f(x^{\oplus,\alpha}).$$

Summing the above inequalities provides the generalized version of prox-grad inequality. $\qquad \square$

**Two problem settings for the composite optimization problem.** For the composite optimization problem, we consider the following two problems.

(P1′) Efficiently reduce $F(x_N^{\oplus,\alpha}) - F_\star$ assuming $x_\star$ exists and $\|x_0 - x_\star\| \le R$.

(P2′) Efficiently reduce $\min_{v \in \partial F(y_N^{\oplus,\alpha})} \|v\|^2$ assuming $F_\star > -\infty$ and $F(y_0) - F_\star \le R$.

Note that when $g = 0$, (P1′) and (P2′) collapse to (P1) and (P2), respectively.

**Parameterized FSFOM that reduces the composite function value : GFPGM [27]** First define $x_\star := \operatorname{argmin}_{x \in \mathbb{R}^d} F(x)$. We recall the composite function minimization FSFOMs with $x_i^{\oplus,\alpha}$ by a lower triangular matrix $\{h_{k,i}\}_{0 \le i < k \le N}$ as follows:

$$x_{k+1} = x_k - \sum_{i=0}^{k} \alpha h_{k+1,i} \left( x_i - x_i^{\oplus,\alpha} \right), \quad \forall k = 0, \dots, N-1. \tag{87}$$

In the case $g = 0$, (87) collapses to (1) as $\alpha \left( z - z^{\oplus,\alpha} \right) = (z - z^+) = \frac{1}{L}\nabla f(z)$. The iterations of GFGPM are defined as

$$x_{k+1} = x_k^{\oplus,1} + \frac{(T_k - t_k)t_{k+1}}{t_k T_{k+1}} \left( x_k^{\oplus,1} - x_{k-1}^{\oplus,1} \right) + \frac{(t_k^2 - T_k)t_{k+1}}{t_k T_{k+1}} \left( x_k^{\oplus,1} - x_k \right), \quad k = 0, \dots, N-1 \tag{88}$$

where $x_{-1}^{\oplus,1} = x_0$ and $[t_i > 0, T_i = \sum_{j=0}^{i} t_j \le t_i^2$ for $0 \le i \le N]$. (88) reduces the composite function value as follows:

$$F(x_N^{\oplus,1}) - F_\star \le \frac{1}{T_N} \frac{1}{2} \|x_0 - x_\star\|^2. \tag{89}$$

We can reconstruct the convergence proof of [27, Theorem 3.3] via our energy function scheme, which is defined as

$$\mathcal{U}_k = \frac{L}{2} \|x_0 - x_\star\|^2 + \sum_{i=0}^{k-1} u_i (\!(x_i, x_{i+1})\!)^1 + \sum_{i=0}^{k} (u_i - u_{i-1})(\!(x_\star, x_i)\!)^1 \tag{90}$$

for $k = -1, \dots, N$. The convergence rate result (89) is proved by using that $\{\mathcal{U}_k\}_{k=-1}^{N}$ is dissipative, and

$$\mathcal{U}_N - T_N \left( F(x_N^{\oplus,1}) - F_\star \right) = \frac{L}{2} \left\| x_0 - x_\star + \sum_{i=0}^{N} (u_i - u_{i-1})(x_i - x_i^{\oplus,1}) \right\|^2 + \sum_{i=0}^{N} \frac{L(T_i - t_i^2)}{2} \left\| x_i^{\oplus,1} - x_i \right\|^2 \ge 0. \tag{91}$$

Here, we can make a crucial observation.

**The formulation of** (90) **is same with** (2)**, which all** $[\![\cdot, \cdot]\!]$ **are changed into** $(\!|\cdot, \cdot|\!)^1$ **and** $u_i = T_i$.

$$(\diamond)$$

Moreover, we can express the recursive formula of the $H$ matrix of (88) as

$$h_{k+1,i} = \begin{cases} 1 + \frac{(t_k-1)t_{k+1}}{T_{k+1}} & i = k \\ \frac{(T_k-t_k)t_{k+1}}{t_k T_{k+1}} (h_{k,k-1} - 1) & i = k-1 \\ \frac{(T_k-t_k)t_{k+1}}{t_k T_{k+1}} h_{k,i} & i = 0, \dots, k-2 \end{cases}. \tag{92}$$

As we used a parameterized family in the convex function (see Section 2.5), we will use these convergence results to construct the method for minimizing $\min_{v \in \partial F(y_N^{\oplus,\alpha})} \|v\|^2$.

**Relationship between minimizing** $\frac{\alpha^2}{2L} \|y_N^{\oplus,\alpha} - y_N\|^2$ **and minimizing** $\min_{v \in \partial F(y_N^{\oplus,\alpha})} \|v\|^2$**.**
Note that $L\alpha(y_N^{\oplus,\alpha} - y_N) + \nabla f(y_N) + u = 0$, $u \in \partial g(y_N^{\oplus,\alpha})$, which gives

$$\min_{v \in \partial F(y_N^{\oplus,\alpha})} \|v\|^2 \underset{(\bullet)}{\leq} \left(\left\|\nabla f(y_N) - \nabla f(y_N^{\oplus,\alpha})\right\| + L\alpha \left\|y_N - y_N^{\oplus,\alpha}\right\|\right)^2 \underset{(\circ)}{\leq} L^2(\alpha+1)^2 \left\|y_N - y_N^{\oplus,\alpha}\right\|^2. \tag{93}$$

$(\bullet)$ is triangle inequality and $(\circ)$ comes from the $L$-smoothness of $f$.

### D.2 Proof of Theorem 3

In this section, we give the proof outline of Theorem 3 and discuss the construction of matrix $C$.

*Proof outline of Theorem 3.* We begin by proposing a parameterized family reduces $\frac{1}{2L} \|y_N^{\oplus,\alpha} - y_N\|^2$ under a fixed value of $a > 0$. To construct this family, take $\{t_i\}_{i=0}^N$, $T_i = \sum_{j=0}^i t_j$ that satisfies (96). We define the $H$ matrix of the family as $\left(\frac{1}{\alpha}H_0 + \frac{1}{\alpha^2}C\right)^A$, where $H_0$ has the same formulation as the $H$ matrix of GFGPM (88) and $C$ follows the recursive formula (97). We refer to this family as (SFG-family).

We can prove that (SFG-family) exhibits the convergence rate

$$\frac{\alpha L}{2} \|y_N^{\oplus,\alpha} - y_N\|^2 \leq \frac{1}{T_N} \left(F(y_0) - F_\star\right), \tag{94}$$

which is motivated by the observation $(\diamond)$. In parallel with $(\diamond)$, we consider the energy function

$$\mathcal{V}_k = \frac{F(y_0) - F(y_N^{\oplus,\alpha}) + (\!|y_{-1}, y_0|\!)^\alpha}{T_N} + \sum_{i=0}^{k-1} \frac{1}{T_{N-i-1}} (\!|y_i, y_{i+1}|\!)^\alpha + \sum_{i=0}^{k-1} \left(\frac{1}{T_{N-i-1}} - \frac{1}{T_{N-i}}\right) (\!|y_N, y_i|\!)^\alpha$$

for $k = 0, \dots, N$, which $[\![y_N, \star]\!]$ changed into $(\!|y_{-1}, y_0|\!)^\alpha$, all other $[\![\cdot, \cdot]\!]$ terms changed into $(\!|\cdot, \cdot|\!)^\alpha$ and $v_i = \frac{1}{T_{N-i}}$.

(94) follows from the fact that $\{\mathcal{V}_i\}_{i=0}^N$ is dissipative and

$$\frac{\alpha L}{2} \|y_N^{\oplus,\alpha} - y_N\|^2 \leq \mathcal{V}_N.$$

Detailed justification is given in Appendix D.3.

For the next step, we show that (SFG) is an instance of (SFG-family), under the choice of

$$\alpha = 4, \quad T_i = \frac{(i+2)(i+3)}{4}, \quad 0 \leq i \leq N.$$

The detailed derivation is given in Appendix D.4.

Finally, combining the convergence result

$$2L\|y_N^{\oplus,4} - y_N\|^2 \le \mathcal{V}_N \le \mathcal{V}_0 \le \frac{4}{(N+2)(N+3)}\left(F(y_0) - F_\star\right)$$

and (93) gives

$$\min_{v \in \partial F(y_N^{\oplus,4})} \|v\|^2 \le 25L^2 \left\|y_N^{\oplus,4} - y_N\right\|^2 \le \frac{50}{(N+2)(N+3)}\left(F(y_0) - F_\star\right).$$

$\square$

**Construction of matrix $C$.** Define

$$g: = L\alpha\big[y_0 - y_0^{\oplus,\alpha}|\ldots|y_N - y_N^{\oplus,\alpha}\big].$$

For an FSFOM (87) with matrix $H_1$, we have $\mathcal{V}_N - \frac{\alpha L}{2}\|y_N^{\oplus,\alpha} - y_N\|^2 = \frac{1}{2L\alpha^2}\operatorname{Tr}\left(g^\mathsf{T} g \mathcal{T}^\oplus\right)$ with

$$\mathcal{T}^\oplus: = \alpha^2 \underbrace{\left[\sum_{i=0}^N \frac{1}{T_{N-i}}\left(\mathcal{H}_1^\mathsf{T}(\mathbf{e}_i + \cdots + \mathbf{e}_N)(\mathbf{e}_i - \mathbf{e}_{i-1})^\mathsf{T} + (\mathbf{e}_i - \mathbf{e}_{i-1})(\mathbf{e}_i + \cdots + \mathbf{e}_N)^\mathsf{T}\mathcal{H}_1\right)\right]}_{\mathcal{A}_3}$$

$$+ \alpha \underbrace{\left[\sum_{i=0}^N \frac{1}{T_{N-i}}\left((\mathbf{e}_{i-1} - \mathbf{e}_i)(\mathbf{e}_{i-1} - \mathbf{e}_N)^\mathsf{T} + (\mathbf{e}_{i-1} - \mathbf{e}_N)(\mathbf{e}_{i-1} - \mathbf{e}_i)^\mathsf{T}\right) - \frac{1}{T_N}\mathbf{e}_0\mathbf{e}_0^\mathsf{T} - \mathbf{e}_N\mathbf{e}_N^\mathsf{T}\right]}_{\mathcal{B}_3}$$

$$+ \left[\frac{\alpha}{T_N}\mathbf{e}_0\mathbf{e}_0^\mathsf{T} - \sum_{i=0}^{N-1} \frac{1}{T_{N-1-i}}\mathbf{e}_i\mathbf{e}_i^\mathsf{T} - \frac{1}{T_0}\mathbf{e}_N\mathbf{e}_N^\mathsf{T}\right]$$

where $e_{-1} = \mathbf{0}$, $e_i$ is a unit vector which $(i+1) - th$ component is 1 and $\mathbb{R}^{N+1}$ and $\mathcal{H}_1: = \begin{bmatrix} 0 & 0 \\ H_1 & 0 \end{bmatrix}$. If we take $H_1$ that makes $\mathcal{T}^\oplus \succeq 0$, $\mathcal{V}_N \ge \frac{\alpha L}{2}\|y_N^{\oplus,\alpha} - y_N\|^2$ follows. Next we observe $\frac{1}{\alpha^2}\mathcal{A}_3 + \frac{1}{\alpha}\mathcal{B}_3 = \mathcal{T}(H_1, v)$ for $v_i = \frac{1}{T_{N-i}}$ which is defined as (22). By expanding (91) and defining $g': = L\big[x_0 - x_0^{\oplus,\alpha}|\ldots|x_N - x_N^{\oplus,\alpha}\big]$,

$$\mathcal{U}_N - T_N\left(F(x_N^{\oplus,1}) - F_\star\right) - \frac{L}{2}\left\|x_0 - x_\star + \sum_{i=0}^N (u_i - u_{i-1})(x_i - x_i^{\oplus,1})\right\|^2 = \operatorname{Tr}\left(g'(g')^\mathsf{T}\mathcal{S}^\oplus\right)$$

follows where

$$\mathcal{S}^\oplus: = \mathcal{S}(H_0, u) - \sum_{i=0}^{N-1}(T_i - t_i^2)\mathbf{e}_i\mathbf{e}_i^\mathsf{T}.$$

Here, $u_i = T_i$, and $\mathcal{S}(H_0, u)$ is given by the formulation (19). Using (91), we obtain

$$\mathcal{S}(H_0, u) = \sum_{i=0}^{N-1}(2T_i - t_i^2)\mathbf{e}_i\mathbf{e}_i^\mathsf{T} + (T_N - t_N^2)\mathbf{e}_N\mathbf{e}_N^\mathsf{T}.$$

Now recall matrix $\mathcal{M}(u)$ (26) and the result of Theorem 1:

$$\mathcal{S}(H_0, u) = \mathcal{M}(u)^\mathsf{T}\mathcal{T}(H_0^A, v)\mathcal{M}(u). \tag{95}$$

By substituting $H_1 = \frac{1}{\alpha}H_0^A + X$ and using (95), the result is

$$\mathcal{M}(u)^\mathsf{T}\mathcal{T}^\oplus\mathcal{M}(u) = \alpha\left[\sum_{i=0}^{N-1}(2T_i - t_i^2)\mathbf{e}_i\mathbf{e}_i^\mathsf{T} + (T_N - t_N^2)\mathbf{e}_N\mathbf{e}_N^\mathsf{T}\right]$$

$$+ \underbrace{\alpha^2\mathcal{M}(u)^\mathsf{T}\left[\sum_{i=0}^N \frac{1}{T_{N-i}}\left(\mathcal{X}^\mathsf{T}(\mathbf{e}_i + \cdots + \mathbf{e}_N)(\mathbf{e}_i - \mathbf{e}_{i-1})^\mathsf{T} + (\mathbf{e}_i - \mathbf{e}_{i-1})(\mathbf{e}_i + \cdots + \mathbf{e}_N)^\mathsf{T}\mathcal{X}\right)\right]\mathcal{M}(u)}_{\mathcal{A}_4}$$

$$+ \underbrace{\mathcal{M}(u)^\mathsf{T}\left[\frac{\alpha}{T_N}\mathbf{e}_0\mathbf{e}_0^\mathsf{T} - \sum_{i=0}^{N-1}\frac{1}{T_{N-1-i}}\mathbf{e}_i\mathbf{e}_i^\mathsf{T} - \frac{1}{T_0}\mathbf{e}_N\mathbf{e}_N^\mathsf{T}\right]\mathcal{M}(u)}_{\mathcal{C}_3}$$

where $\mathcal{X} := \begin{bmatrix} 0 & 0 \\ X & 0 \end{bmatrix}$.

Next, consider $\mathcal{A}_4$ as a function of the lower triangular matrix $\mathcal{X}$. The key observation is that if we choose $\mathcal{X}$ appropriately, all non-diagonal terms of $\mathcal{C}_3$ can be eliminated. With this choice of $\mathcal{X}$, we have

$$\mathcal{T}^\oplus = \alpha \sum_{i=0}^{N} (2T_i - t_i^2) \mathbf{e}_i \mathbf{e}_i^\mathsf{T} - T_0 \mathbf{e}_0 \mathbf{e}_0^\mathsf{T} - \sum_{i=1}^{N} \left( \frac{T_i^2}{T_{i-1}} + t_i^2 \sum_{j=0}^{i-2} \frac{1}{T_j} \right) \mathbf{e}_i \mathbf{e}_i^\mathsf{T}.$$

Note that $\mathcal{T}^\oplus$ contains only diagonal terms. Therefore, $\mathcal{T}^\oplus \succeq 0$ is equivalent to all coefficients of $\mathbf{e}_i \mathbf{e}_i^\mathsf{T}$ being nonnegative, which can be formulated using $t_i$ and $T_i$.

**Remark.** In the proof of Theorem 1 ( $\mathbf{B}_2$ and $\mathbf{B}_1$ at (27) and (28)), we have shown that

$$\mathcal{A}_4 = \alpha^2 \left[ \sum_{i=0}^{N} \left( \mathcal{X}^A \right)^\mathsf{T} T_i (\mathbf{e}_i - \mathbf{e}_{i+1})(\mathbf{e}_0 + \cdots + \mathbf{e}_i)^\mathsf{T} + T_i (\mathbf{e}_0 + \cdots + \mathbf{e}_i)(\mathbf{e}_i - \mathbf{e}_{i+1})^\mathsf{T} \mathcal{X}^A \right].$$

Observe that $(\mathbf{e}_i - \mathbf{e}_{i+1})(\mathbf{e}_0 + \cdots + \mathbf{e}_i)^\mathsf{T}$ is a lower triangular matrix and $\mathcal{X}^A$ is a strictly lower triangular matrix, i.e., all diagonal component is 0. Thus it is worth noting that $\mathcal{A}_4$ cannot induce any diagonal term $\mathbf{e}_i \mathbf{e}_i^\mathsf{T}$, choosing matrix $C$ as the optimal one.

### D.3 Parameterized family reduces the gradient mapping norm

**FSFOM that reduces $\frac{1}{2L} \left\| y_N^{\oplus,\alpha} - y_N \right\|^2$ in the composite minimization.** We propose the parameterized family that reduces $\frac{1}{2L} \left\| y_N^{\oplus,\alpha} - y_N \right\|^2$. For fixed $a > 0$, take $\{t_i\}_{i=0}^N$, $T_i = \sum_{j=0}^i t_j$ that satisfies the following conditions:

$$\alpha(2T_0 - t_0^2) \geq T_0,$$

$$\alpha(2T_k - t_k^2) \geq \frac{T_k^2}{T_{k-1}} + t_k^2 \left( \frac{1}{T_0} + \sum_{i=0}^{k-2} \frac{1}{T_i} \right) \quad k = 1, \ldots, N \tag{96}$$

where we define $\sum_{i=0}^{k-2} \frac{1}{T_i} = 0$ for $k = 1$ case. We define an lower triangular matrix $C = \{c_{k,i}\}_{0 \leq i < k \leq N}$ as

$$c_{k+1,i} = \begin{cases} \frac{t_1}{T_1} & i = 0, \ k = 0 \\ \frac{t_{k+1}}{T_{k+1}} \left( \frac{t_k}{T_0} + t_k \sum_{j=0}^{k-2} \frac{1}{T_j} + \frac{T_k}{T_{k-1}} \right) & i = k, \ k = 1, \ldots, N-1 \\ \frac{t_{k+1}(T_k - t_k)}{t_k T_{k+1}} c_{k,i} & i = 0, \ldots, k-1, \ i = 2, \ldots, N-1 \end{cases} \tag{97}$$

Matrix $C$ has a crucial role in the correspondence between composite function value minimization and composite function gradient norm minimization with $H \to \left( \frac{1}{\alpha} H + \frac{1}{\alpha^2} C \right)^A$.

**Proposition 5** (SFG family). Assume that we choose $H_0$ matrix of (88), and define $C$ as (97). Consider an FSFOM (87) which $H$ matrix is $\left( \frac{1}{\alpha} H_0 + \frac{1}{\alpha^2} C \right)^A$. Such FSFOM can be expressed as

$$y_{k+1} = y_k^{\oplus,\alpha} + \beta_k' \left( y_k^{\oplus,\alpha} - y_{k-1}^{\oplus,\alpha} \right) + \gamma_k' \left( y_k^{\oplus,\alpha} - y_k \right), \quad k = 0, \ldots, N-1,$$

$$\beta_k' = \frac{\beta_{N-k}(\beta_{N-1-k} + \gamma_{N-1-k})}{\beta_{N-k} + \gamma_{N-k}}, \qquad \gamma_k' = \frac{\gamma_{N-k}(\beta_{N-1-k} + \gamma_{N-1-k})}{\beta_{N-k} + \gamma_{N-k}}, \qquad \text{(SFG-family)}$$

$$\beta_k = \frac{t_{k+1}(T_k - t_k)}{t_k T_{k+1}}, \qquad \gamma_k = \frac{t_{k+1}(t_k^2 - T_k)}{t_k T_{k+1}} + \frac{1}{\alpha} c_{k+1,k}.$$

and $y_{-1}^{\oplus,\alpha} = y_0$. Then it exhibits the convergence rate

$$\frac{\alpha L}{2} \| y_N^\oplus - y_N \|^2 \leq \frac{1}{T_N} \left( F(y_0) - F_\star \right).$$

*Proof of Proposition 5.* To start, we give the claim about matrix $C$.

**Claim 1.** $\mathcal{C} := \begin{bmatrix} 0 & 0 \\ C & 0 \end{bmatrix}$ satisfies the following equality.

$$2 \left( \sum_{i=0}^{N} T_i (\mathbf{e}_i - \mathbf{e}_{i+1})(e_0 + \cdots + e_i)^\intercal \right) \mathcal{C}$$
$$= \sum_{i=1}^{N} \frac{1}{T_{i-1}} \mathbf{f}_{N-i} \mathbf{f}_{N-i}^\intercal + \frac{1}{T_0} \mathbf{f}_N \mathbf{f}_N^\intercal - T_0 e_0 e_0^\intercal - \sum_{i=1}^{N} \left( \frac{T_i^2}{T_{i-1}} + t_i^2 \sum_{j=0}^{i-2} \frac{1}{T_j} \right) \mathbf{e}_i \mathbf{e}_i^\intercal \tag{98}$$

where $\{\mathbf{e}_i\}_{i=0}^{N}$ is a any basis of $\mathbb{R}^{N+1}$, $\mathbf{e}_{N+1}$, $\mathbf{f}_{-1}$ are zero vectors and $\{\mathbf{f}_i\}_{i=0}^{N}$ are another basis of $\mathbb{R}^{N+1}$ which is defined as

$$T_i \left( \mathbf{e}_i - \mathbf{e}_{i+1} \right) = \mathbf{f}_{N-i} - \mathbf{f}_{N-i-1}, \quad i = 0, 1, \ldots, N.$$

The proof of Claim 1 will be provided after the proof of Proposition 5. To begin with, for the FSFOM with matrix $\left( \frac{1}{\alpha} H_0 + \frac{1}{\alpha^2} C \right)^A$, define the energy function

$$\mathcal{V}_k = \frac{F(y_0) - F(y_N^{\oplus,\alpha}) + (\!|y_{-1}, y_0|\!)^\alpha}{T_N} + \sum_{i=0}^{k-1} \frac{1}{T_{N-i-1}} (\!|y_i, y_{i+1}|\!)^\alpha + \sum_{i=0}^{k-1} \left( \frac{1}{T_{N-i-1}} - \frac{1}{T_{N-i}} \right) (\!|y_N, y_i|\!)^\alpha,$$

for $k = 0, \ldots, N$. $\{\mathcal{V}_k\}_{k=0}^{N}$ is dissipative since $(\!|\cdot, \cdot|\!)^\alpha \leq 0$. This energy function is inspired by (3). Note that if we can prove

$$\frac{\alpha L}{2} \|y_N^{\oplus,\alpha} - y_N\|^2 \leq \mathcal{V}_N, \tag{99}$$

then

$$\frac{\alpha L}{2} \left\| y_N^{\oplus,\alpha} - y_N \right\|^2 \leq \mathcal{V}_N \leq \mathcal{V}_0 \leq \frac{1}{T_N} \left( F(y_0) - F(y_N^{\oplus,\alpha}) \right) \leq \frac{1}{T_N} \left( F(y_0) - F_\star \right).$$

Thus, it is enough to show (99). Defining $g_i := L\alpha(y_i - y_i^{\oplus,\alpha})$ for $0 \leq i \leq N$ gives

$$(\!|y_i, y_j|\!)^\alpha = F(y_j^{\oplus,\alpha}) - F(y_i^{\oplus,\alpha}) + \langle g_j, y_i - y_j \rangle + \frac{1}{La} \langle g_j, g_j - g_i \rangle - \frac{1}{2L\alpha^2} \|g_j\|^2,$$
$$(\!|y_{-1}, y_0|\!)^\alpha = F(y_0^{\oplus,\alpha}) - F(y_0) + \frac{1}{L} \frac{2\alpha - 1}{2\alpha^2} \|g_0\|^2.$$

Plugging the above equalities provides

$$2L\alpha^2 \left( \mathcal{V}_N - \frac{\alpha L}{2} \left\| y_N^{\oplus,\alpha} - y_N \right\|^2 \right)$$

$$= 2L\alpha^2 \left[ \sum_{i=0}^{N-1} \frac{1}{T_{N-i-1}} \langle g_{i+1}, y_i - y_{i+1} \rangle + \sum_{i=0}^{N-1} \left( \frac{1}{T_{N-i-1}} - \frac{1}{T_{N-i}} \right) \langle g_i, y_N - y_i \rangle \right] + \frac{2\alpha - 1}{T_N} \|g_0\|^2 - \alpha \|g_N\|^2$$

$$+ \sum_{i=0}^{N-1} \frac{1}{T_{N-i-1}} \left( 2\alpha \langle g_{i+1}, g_{i+1} - g_i \rangle - \|g_{i+1}\|^2 \right) + \sum_{i=0}^{N-1} \left( \frac{1}{T_{N-i-1}} - \frac{1}{T_{N-i}} \right) \left( 2\alpha \langle g_i, g_i - g_N \rangle - \|g_i\|^2 \right)$$

$$= 2L\alpha^2 \left( \sum_{i=0}^{N-1} \frac{1}{T_{N-i-1}} \langle g_{i+1}, y_i - y_{i+1} \rangle + \sum_{i=0}^{N-1} \left( \frac{1}{T_{N-i-1}} - \frac{1}{T_{N-i}} \right) \langle g_i, y_N - y_i \rangle \right)$$

$$+ \left[ \sum_{i=0}^{N-1} \frac{\alpha}{T_{N-i-1}} \|g_{i+1} - g_i\|^2 + \sum_{i=0}^{N-1} \left( \frac{\alpha}{T_{N-i-1}} - \frac{\alpha}{T_{N-i}} \right) \|g_i - g_N\|^2 - a\|g_N\|^2 + \frac{\alpha}{T_N} \|g_N\|^2 \right]$$

$$+ \sum_{i=0}^{N-1} \frac{1}{T_{N-i-1}} \left( -\alpha\|g_i\|^2 + (a - 1)\|g_{i+1}\|^2 \right) + \sum_{i=0}^{N-1} \left( \frac{1}{T_{N-i-1}} - \frac{1}{T_{N-i}} \right) \left( -\alpha\|g_N\|^2 + (\alpha - 1)\|g_i\|^2 \right)$$

$$+ \frac{2\alpha - 1}{T_N} \|g_0\|^2 - \frac{\alpha}{T_N} \|g_N\|^2$$

$$= 2L\alpha^2 \left( \sum_{i=0}^{N-1} \frac{1}{T_{N-i-1}} \langle g_{i+1}, y_i - y_{i+1} \rangle + \sum_{i=0}^{N-1} \left( \frac{1}{T_{N-i-1}} - \frac{1}{T_{N-i}} \right) \langle g_i, y_N - y_i \rangle \right)$$

$$+ \left[ \sum_{i=0}^{N-1} \frac{\alpha}{T_{N-i-1}} \|g_{i+1} - g_i\|^2 + \sum_{i=0}^{N-1} \left( \frac{\alpha}{T_{N-i-1}} - \frac{\alpha}{T_{N-i}} \right) \|g_i - g_N\|^2 - \alpha\|g_N\|^2 + \frac{\alpha}{T_N} \|g_N\|^2 \right]$$

$$+ \frac{\alpha}{T_N} \|g_0\|^2 - \sum_{i=0}^{N-1} \frac{1}{T_{N-1-i}} \|g_i\|^2 - \frac{1}{T_0} \|g_N\|^2 .$$

Next, define $g := [g_0|g_1 \dots |g_N]$ and

$$\mathcal{H}_1 := \begin{bmatrix} 0 & 0 \\ \left( \frac{1}{\alpha} H_0 + \frac{1}{\alpha^2} C \right)^A & 0 \end{bmatrix}, \quad \mathcal{H}_0 := \begin{bmatrix} 0 & 0 \\ H_0 & 0 \end{bmatrix}.$$

By the same procedure as the proof of Theorem 1, we obtain $2L\alpha^2 \left( \mathcal{V}_N - \frac{\alpha L}{2} \left\| y_N^{\oplus,\alpha} - y_N \right\|^2 \right) =$ $\mathrm{Tr}\left( g^\intercal g \mathcal{T}^\oplus \right)$ where

$$\mathcal{T}^\oplus = \alpha^2 \left[ \sum_{i=0}^{N} \frac{1}{T_{N-i}} (\mathcal{H}_1^\intercal (\mathbf{f}_i + \cdots + \mathbf{f}_N)(\mathbf{f}_i - \mathbf{f}_{i-1})^\intercal + (\mathbf{f}_i - \mathbf{f}_{i-1})(\mathbf{f}_i + \cdots + \mathbf{f}_N)^\intercal \mathcal{H}_1) \right]$$

$$+ \alpha \left[ \sum_{i=0}^{N} \frac{1}{T_{N-i}} ((\mathbf{f}_{i-1} - \mathbf{f}_i)(\mathbf{f}_{i-1} - \mathbf{f}_N)^\intercal + (\mathbf{f}_{i-1} - \mathbf{f}_N)(\mathbf{f}_{i-1} - \mathbf{f}_i)^\intercal) - \frac{1}{T_N} \mathbf{f}_0 \mathbf{f}_0^\intercal - \mathbf{f}_N \mathbf{f}_N^\intercal \right]$$

$$+ \left[ \frac{\alpha}{T_N} \mathbf{f}_0 \mathbf{f}_0^\intercal - \sum_{i=0}^{N-1} \frac{1}{T_{N-1-i}} \mathbf{f}_i \mathbf{f}_i^\intercal - \frac{1}{T_0} \mathbf{f}_N \mathbf{f}_N^\intercal \right]$$

And consider the vector $\mathbf{e}_i \in \mathbb{R}^{(N+1)\times 1}$ which is same with (29).

In the proof of Theorem 1 $\big((27), (28)\big)$, we have shown that

$$\frac{1}{2L} \sum_{i=0}^{N} \frac{1}{T_{N-i}} (\mathcal{H}_1^\intercal (\mathbf{f}_i + \cdots + \mathbf{f}_N)(\mathbf{f}_i - \mathbf{f}_{i-1})^\intercal + (\mathbf{f}_i - \mathbf{f}_{i-1})(\mathbf{f}_i + \cdots + \mathbf{f}_N)^\intercal \mathcal{H}_1)$$

$$= \frac{1}{2L} \left( \frac{\mathcal{H}_0^\intercal}{\alpha} + \frac{\mathcal{C}^\intercal}{\alpha^2} \right) \left[ \sum_{i=0}^{N} T_i (\mathbf{e}_0 + \cdots + \mathbf{e}_i)(\mathbf{e}_i - \mathbf{e}_{i+1})^\intercal \right] + \frac{1}{2L} \left[ \sum_{i=0}^{N} T_i (\mathbf{e}_i - \mathbf{e}_{i+1})(\mathbf{e}_0 + \cdots + \mathbf{e}_i)^\intercal \right] \left( \frac{\mathcal{H}_0}{\alpha} + \frac{\mathcal{C}}{\alpha^2} \right)$$

and

$$\frac{1}{2L}\sum_{i=0}^{N}\frac{1}{T_{N-i}}\left((\mathbf{f}_{i-1}-\mathbf{f}_i)(\mathbf{f}_{i-1}-\mathbf{f}_N)^\intercal+(\mathbf{f}_{i-1}-\mathbf{f}_N)(\mathbf{f}_{i-1}-\mathbf{f}_i)^\intercal\right)-\frac{1}{2T_NL}\mathbf{f}_0\mathbf{f}_0^\intercal-\frac{1}{2L}\mathbf{f}_N\mathbf{f}_N^\intercal$$

$$=-\frac{1}{2L}\left(\sum_{i=0}^{N}(T_i-T_{i-1})\mathbf{e}_i\right)\left(\sum_{i=0}^{N}(T_i-T_{i-1})\mathbf{e}_i\right)^\intercal$$

$$+\frac{1}{2L}\left[\sum_{i=0}^{N}T_i\left((\mathbf{e}_i-\mathbf{e}_{i+1})\mathbf{e}_i^\intercal+\mathbf{e}_i(\mathbf{e}_i-\mathbf{e}_{i+1})^\intercal\right)-T_N\mathbf{e}_N\mathbf{e}_N^\intercal\right]$$

under the above transformation. Therefore, $\mathcal{T}^\oplus$ can be expressed as

$$\mathcal{T}^\oplus=\alpha^2\left(\left(\frac{\mathcal{H}_0^\intercal}{\alpha}+\frac{\mathcal{C}^\intercal}{\alpha^2}\right)\left[\sum_{i=0}^{N}T_i(\mathbf{e}_0+\cdots+\mathbf{e}_i)(\mathbf{e}_i-\mathbf{e}_{i+1})^\intercal\right]+\left[\sum_{i=0}^{N}T_i(\mathbf{e}_i-\mathbf{e}_{i+1})(\mathbf{e}_0+\cdots+\mathbf{e}_i)^\intercal\right]\left(\frac{\mathcal{H}_0}{\alpha}+\frac{\mathcal{C}}{\alpha^2}\right)\right)$$

$$+\alpha\left(-\left(\sum_{i=0}^{N}(T_i-T_{i-1})\mathbf{e}_i\right)\left(\sum_{i=0}^{N}(T_i-T_{i-1})\mathbf{e}_i\right)^\intercal+\left[\sum_{i=0}^{N}T_i\left((\mathbf{e}_i-\mathbf{e}_{i+1})\mathbf{e}_i^\intercal+\mathbf{e}_i(\mathbf{e}_i-\mathbf{e}_{i+1})^\intercal\right)-T_N\mathbf{e}_N\mathbf{e}_N^\intercal\right]\right)$$

$$+\left[\frac{\alpha}{T_N}\mathbf{f}_0\mathbf{f}_0^\intercal-\sum_{i=0}^{N-1}\frac{1}{T_{N-1-i}}\mathbf{f}_i\mathbf{f}_i^\intercal-\frac{1}{T_0}\mathbf{f}_N\mathbf{f}_N^\intercal\right]$$

$$=\alpha\mathcal{A}+\mathcal{B}+\frac{\alpha}{T_N}\mathbf{f}_0\mathbf{f}_0^\intercal$$

where

$$\mathcal{A}=\mathcal{H}_0^\intercal\left[\sum_{i=0}^{N}T_i(\mathbf{e}_0+\cdots+\mathbf{e}_i)(\mathbf{e}_i-\mathbf{e}_{i+1})^\intercal\right]+\left[\sum_{i=0}^{N}T_i(\mathbf{e}_i-\mathbf{e}_{i+1})(\mathbf{e}_0+\cdots+\mathbf{e}_i)^\intercal\right]\mathcal{H}_0$$

$$-\left(\sum_{i=0}^{N}(T_i-T_{i-1})\mathbf{e}_i\right)\left(\sum_{i=0}^{N}(T_i-T_{i-1})\mathbf{e}_i\right)^\intercal+\left[\sum_{i=0}^{N}T_i\left((\mathbf{e}_i-\mathbf{e}_{i+1})\mathbf{e}_i^\intercal+\mathbf{e}_i(\mathbf{e}_i-\mathbf{e}_{i+1})^\intercal\right)-T_N\mathbf{e}_N\mathbf{e}_N^\intercal\right]$$

and

$$\mathcal{B}=\mathcal{C}^\intercal\left[\sum_{i=0}^{N}T_i(\mathbf{e}_0+\cdots+\mathbf{e}_i)(\mathbf{e}_i-\mathbf{e}_{i+1})^\intercal\right]+\left[\sum_{i=0}^{N}T_i(\mathbf{e}_i-\mathbf{e}_{i+1})(\mathbf{e}_0+\cdots+\mathbf{e}_i)^\intercal\right]\mathcal{C}$$

$$-\sum_{i=0}^{N-1}\frac{1}{T_{N-1-i}}\mathbf{f}_i\mathbf{f}_i^\intercal-\frac{1}{T_0}\mathbf{f}_N\mathbf{f}_N^\intercal.$$

To calculate $\mathcal{A}$, expand the energy function (91). Then we obtain

$$\mathcal{A}=\sum_{i=0}^{N-1}(2T_i-t_i^2)\mathbf{e}_i\mathbf{e}_i^\intercal+(T_N-t_N^2)\mathbf{e}_N\mathbf{e}_N^\intercal.$$

Moreover, by Claim 1,

$$\mathcal{B}=-T_0\mathbf{e}_0\mathbf{e}_0^\intercal-\sum_{i=1}^{N}\left(\frac{T_i^2}{T_{i-1}}+t_i^2\sum_{j=0}^{i-2}\frac{1}{T_j}\right)\mathbf{e}_i\mathbf{e}_i^\intercal.$$

Combining above results and $T_N\mathbf{e}_N=\mathbf{f}_0$, we achieve

$$\mathcal{T}^\oplus=\alpha\sum_{i=0}^{N}(2T_i-t_i^2)\mathbf{e}_i\mathbf{e}_i^\intercal-T_0\mathbf{e}_0\mathbf{e}_0^\intercal-\sum_{i=1}^{N}\left(\frac{T_i^2}{T_{i-1}}+t_i^2\sum_{j=0}^{i-2}\frac{1}{T_j}\right)\mathbf{e}_i\mathbf{e}_i^\intercal$$

and the condition (96) makes each coefficient of $\mathbf{e}_i\mathbf{e}_i^\intercal$ nonnegative, which gives $\mathcal{T}\succeq0$. To achieve the iteration formula (SFG-family), consider the FSFOM (87) with $\frac{1}{\alpha}H_0+\frac{1}{\alpha^2}C$, which can be

expressed as

$$x_{k+1} = x_k^{\oplus,\alpha} + \beta_k \left(x_k^{\oplus,\alpha} - x_{k-1}^{\oplus,\alpha}\right) + \gamma_k \left(x_k^{\oplus,\alpha} - x_k\right), \quad k = 0, \ldots, N-1,$$

$$\beta_k = \frac{t_{k+1}(T_k - t_k)}{t_k T_{k+1}}, \quad \gamma_k = \frac{t_{k+1}(t_k^2 - T_k)}{t_k T_{k+1}} + \frac{1}{\alpha} c_{k+1,k}.$$

For last step, We make an observation that Proposition 1 can be applied when all $z^+$ terms changed into $z^{\oplus,\alpha}$. Proposition 1 gives the H-dual. $\qquad\square$

*Proof of Claim 1.* The left hand side of (98) is

$$2 \left(\sum_{i=0}^{N} T_i(\mathbf{e}_i - \mathbf{e}_{i+1})(\mathbf{e}_0 + \cdots + \mathbf{e}_i)^{\mathsf{T}}\right) \mathcal{C}$$

$$= 2 \left(\sum_{i=0}^{N} T_i(\mathbf{e}_i - \mathbf{e}_{i+1})(\mathbf{e}_0 + \cdots + \mathbf{e}_i)^{\mathsf{T}}\right) \left(\sum_{k>i} c_{k,i} \mathbf{e}_k \mathbf{e}_i^{\mathsf{T}}\right)$$

$$= 2 \left(\sum_{i=0}^{N} (\mathbf{f}_{N-i} - \mathbf{f}_{N-i+1})(\mathbf{e}_0 + \cdots + \mathbf{e}_i)^{\mathsf{T}}\right) \left(\sum_{k>i} c_{k,i} \mathbf{e}_k \mathbf{e}_i^{\mathsf{T}}\right)$$

$$= 2 \left(\sum_{i=0}^{N} \mathbf{f}_{N-i} \mathbf{e}_i^{\mathsf{T}}\right) \left(\sum_{k>i} c_{k,i} \mathbf{e}_k \mathbf{e}_i^{\mathsf{T}}\right)$$

$$= 2 \sum_{i=0}^{N} \left(\sum_{k=i+1}^{N} c_{k,i} \mathbf{f}_{N-k}\right) \mathbf{e}_i^{\mathsf{T}}$$

Now we calculate the right-hand side of (98). By plugging

$$\mathbf{f}_{N-i} = T_i \mathbf{e}_i + t_{i+1} \mathbf{e}_{i+1} + \cdots + t_N \mathbf{e}_N,$$

$$\sum_{i=1}^{N} \frac{1}{T_{i-1}} \mathbf{f}_{N-i} \mathbf{f}_{N-i}^{\mathsf{T}} + \frac{1}{T_0} \mathbf{f}_N \mathbf{f}_N^{\mathsf{T}} - T_0 \mathbf{e}_0 \mathbf{e}_0^{\mathsf{T}} - \sum_{i=1}^{N} \left(\frac{T_i^2}{T_{i-1}} + t_i^2 \sum_{j=0}^{i-2} \frac{1}{T_j}\right) \mathbf{e}_i \mathbf{e}_i^{\mathsf{T}} = 2 \sum_{i=0}^{N-1} \mathbf{a}_i \mathbf{e}_i^{\mathsf{T}}$$

where

$$\mathbf{a}_0 = t_1 \mathbf{e}_1 + \cdots + t_N \mathbf{e}_N,$$

$$\mathbf{a}_i = \left(\frac{t_i}{T_0} + \sum_{j=0}^{i-2} \frac{t_i}{T_j} + \frac{T_i}{T_{i-1}}\right) (t_{i+1} \mathbf{e}_{i+1} + \cdots + t_N \mathbf{e}_N), \quad i = 1, \ldots, N-1.$$

Now we claim that $\mathbf{a}_i = \sum_{k=i+1}^{N} c_{k,i} \mathbf{f}_{N-k}$ for $i = 0, \ldots, N-1$. Note that

$$\sum_{k=i+1}^{N} c_{k,i} \mathbf{f}_{N-k} = \sum_{k=i+1}^{N} c_{k,i} \left(T_k \mathbf{e}_k + t_{k+1} \mathbf{e}_{k+1} + \cdots + t_N \mathbf{e}_N\right)$$

$$= \sum_{k=i+1}^{N} \left(c_{k,i} T_k + c_{k-1,i} t_k + \cdots + c_{i+1,i} t_k\right) \mathbf{e}_k.$$

Coefficients of $\mathbf{e}_{i+1}$ are coincides since

$$c_{1,0} T_1 = t_1,$$

$$c_{i+1,i} T_{i+1} = t_{i+1} \left(\frac{t_i}{T_0} + \sum_{j=0}^{i-2} \frac{t_i}{T_j} + \frac{T_i}{T_{i-1}}\right) \quad i = 1, \ldots, N-1.$$

Now assume

$$\frac{t_j}{t_{j+1}} = \frac{c_{j,i} T_j + c_{j-1,i} t_j + \cdots + c_{i+1,i} t_j}{c_{j+1,i} T_{j+1} + c_{j,i} t_{j+1} + \cdots + c_{i+1,i} t_{j+1}}. \tag{100}$$

By multiplying the above equation recursively, we obtain

$$\frac{t_{i+1}}{t_{j+1}} = \frac{c_{i+1,i}T_i}{c_{j+1,i}T_{j+1} + c_{j,i}t_{j+1} + \cdots + c_{i+1,i}t_{j+1}}$$

for $1 \le j \le N - 1$. which implies $\mathbf{a}_i = \sum_{k=i+1}^{N} c_{k,i}\mathbf{f}_{N-k}$.

To prove (100), expand and obtain

$$t_{j+1}c_{j,i}T_j = t_j\left(c_{j+1,i}T_{j+1} + c_{j,i}t_{j+1}\right).$$

Above equation holds due to the recursive formula of $C$ (97). $\qquad\square$

## D.4 Instances of SFG family

**Derivation of (SFG)** Here, we will show that (SFG) is the instance of SFG family, under the choice of

$$\alpha = 4, \quad T_i = \frac{(i+2)(i+3)}{4}, \quad 0 \le i \le N.$$

First, $t_i = \frac{i+2}{2}$ for $i \ge 1$ and $t_0 = \frac{3}{2}$. Also (96) holds since

$$4\left(3 - \frac{9}{4}\right) \ge \frac{3}{2}$$

$$4\left(\frac{(k+2)(k+3)}{2} - \frac{(k+2)^2}{4}\right) \ge \frac{\frac{(k+2)^2(k+3)^2}{16}}{\frac{(k+1)(k+2)}{4}} + \frac{(k+2)^2}{4}\left(\frac{2}{3} + 4\left(\frac{1}{2} - \frac{1}{k+1}\right)\right) \quad k = 1, \ldots, N.$$

Plugging above values into (92) and (97), we obtain

$$h_{k+1,i} = \begin{cases} 1 + \frac{1}{4} & i = k, \ k = 0 \\ 1 + \frac{k}{k+4} & i = k, \ k = 1, \ldots, N-1 \\ \frac{k+1}{k+4}(h_{k,k-1} - 1) & i = k - 1 \\ \frac{k+1}{k+4}h_{k,i} & i = 0, \ldots, k - 2 \end{cases}.$$

and

$$c_{k+1,i} = \begin{cases} \frac{1}{2} & i = 0, \ k = 0 \\ \frac{2(4k+5)}{3(k+4)} & i = k, \ k = 1, \ldots, N-1 \\ \frac{k+1}{k+4}c_{k,i} & i = 0, \ldots, k-1, \ i = 2, \ldots, N-1 \end{cases}.$$

Other terms come directly, and

$$\begin{aligned} c_{k+1,k} &= \frac{t_{k+1}}{T_{k+1}}\left(\frac{t_k}{T_0} + t_k\sum_{j=0}^{k-2}\frac{1}{T_j} + \frac{T_k}{T_{k-1}}\right) \\ &= \frac{t_{k+1}}{T_{k+1}}\left(\frac{t_k}{T_0} + t_k\sum_{j=0}^{k-1}\frac{1}{T_j} + 1\right) \\ &= \frac{\frac{k+3}{2}}{\frac{(k+3)(k+4)}{4}}\left(\frac{k+2}{2} \times \frac{2}{3} + \frac{k+2}{2}\left(\frac{4}{2 \times 3} + \cdots + \frac{4}{(k+1)(k+2)}\right) + 1\right) \\ &= \frac{2}{k+4}\left(\frac{k+2}{3} + \frac{k+2}{2}\frac{2k}{k+2} + 1\right) \\ &= \frac{2(4k+5)}{3(k+4)}. \end{aligned}$$

To sum up, the matrix $\{g_{k,i}\}_{0\leq i<k\leq N}:\ =\frac{1}{4}H_0+\frac{1}{16}C$ satisfies the following recursive formula.

$$g_{k+1,i}=\begin{cases}\frac{1}{4}\left(1+\frac{3}{8}\right) & i=0,\ k=0\\[2mm]\frac{1}{4}\left(1+\frac{10k+5}{6(k+4)}\right) & i=k,\ k=1,\ldots,N-1\\[2mm]\frac{k+1}{k+4}\left(g_{k,i}-\frac{1}{4}\right) & i=k-1\\[2mm]\frac{k+1}{k+4}g_{k,i} & i=0,\ldots,k-2\end{cases}.$$

Note that

$$g_{k+1,k}=\frac{1}{4}\left(1+\frac{k}{k+4}\right)+\frac{1}{16}\left(\frac{2(4k+5)}{3(k+4)}\right)$$

$$=\frac{1}{4}\left(1+\frac{k}{k+4}+\frac{4k+5}{6(k+4)}\right)$$

$$=\frac{1}{4}\left(1+\frac{10k+5}{6(k+4)}\right).$$

Therefore, the FSFOM with $\{g_{k,i}\}_{0\leq i<k\leq N}$ can be expressed as

$$x_1=x_0^{\oplus,4}+\frac{1}{4}\cdot\left(x_0^{\oplus,4}-x_{-1}^{\oplus,4}\right)+\frac{1}{8}\left(x_0^{\oplus,4}-x_0\right),$$

$$x_{k+1}=x_k^{\oplus,4}+\frac{k+1}{k+4}\left(x_k^{\oplus,4}-x_{k-1}^{\oplus,4}\right)+\frac{4k-1}{6(k+4)}\left(x_k^{\oplus,4}-x_k\right),\quad k=1,\ldots,N-1$$

where $x_{-1}^{\oplus,4}=x_0$. To obtain the H-dual, we apply Proposition 1.

$$y_{k+1}=y_k^{\oplus,4}+\frac{(N-k+1)(2N-2k-1)}{(N-k+3)(2N-2k+1)}\left(y_k^{\oplus,4}-y_{k-1}^{\oplus,4}\right)+\frac{(4N-4k-1)(2N-2k-1)}{6(N-k+3)(2N-2k+1)}\left(y_k^{\oplus,4}-y_k\right)$$

$$y_N=y_{N-1}^{\oplus,4}+\frac{3}{10}\left(y_{N-1}^{\oplus,4}-y_{N-2}^{\oplus,4}\right)+\frac{3}{40}\left(y_{N-1}^{\oplus,4}-y_{N-1}\right)$$

where $k=0,\ldots,N-2$ and $y_{-1}^{\oplus,4}=y_0$.

**Fastest method among the SFG family via a numerical choice of** $a$  In this section, we give simple form of SFG, when all inequality conditions in (96) holds as equalities. $\{g_{k,i}\}_{0\leq i<k\leq N}$ becomes

$$g_{1,0}=\frac{1}{\alpha}\left(1+\frac{(t_0-1)t_1}{T_1}\right)+\frac{1}{\alpha^2}\left(\frac{t_1}{T_1}\right)$$

and for $k>0$,

$$g_{k+1,k}=\frac{1}{\alpha}\left(1+\frac{(t_k-1)t_{k+1}}{T_{k+1}}\right)+\frac{1}{\alpha^2}\frac{t_{k+1}}{T_{k+1}}\left(t_k\left(\frac{1}{T_0}+\frac{1}{T_0}+\cdots+\frac{1}{T_{k-2}}\right)+\frac{T_k}{T_{k-1}}\right)$$

$$=\frac{1}{\alpha}\left(1+\frac{(t_k-1)t_{k+1}}{T_{k+1}}\right)+\frac{1}{\alpha^2}\frac{t_{k+1}}{T_{k+1}}\left(\frac{\alpha(2T_k-t_k^2)-\frac{T_k^2}{T_{k-1}}}{t_k}+\frac{T_k}{T_{k-1}}\right)$$

$$=\frac{1}{\alpha}\left(1+\frac{(t_k-1)t_{k+1}}{T_{k+1}}\right)+\frac{1}{\alpha^2}\frac{t_{k+1}}{T_{k+1}}\left(\frac{\alpha(2T_k-t_k^2)}{t_k}-\frac{T_k}{t_k}\right)$$

$$=\frac{1}{\alpha}+\frac{1}{\alpha}\frac{t_{k+1}}{T_{k+1}}\left(t_k-1+\frac{2T_k-t_k^2}{t_k}-\frac{T_k}{t_k}\right)-\left(\frac{1}{\alpha^2}-\frac{1}{\alpha}\right)\frac{t_{k+1}T_k}{T_{k+1}t_k}$$

$$=\frac{1}{\alpha}+\frac{1}{\alpha}\frac{(T_k-t_k)t_{k+1}}{t_kT_{k+1}}+\left(\frac{1}{\alpha}-\frac{1}{\alpha^2}\right)\frac{t_{k+1}T_k}{T_{k+1}t_k},$$

$$g_{k+1,k-1}=\frac{t_{k+1}(T_k-t_k)}{t_kT_{k+1}}\left(g_{k,k-1}-\frac{1}{\alpha}\right),$$

$$g_{k+1,i}=\frac{t_{k+1}(T_k-t_k)}{t_kT_{k+1}}g_{k,i},\quad i=k-2,\ldots,0.$$

Hence FSFOM with $\frac{1}{\alpha}H_0 + \frac{1}{\alpha^2}C$ is

$$x_{k+1} = x_k^{\oplus,\alpha} + \frac{T_{k-1}t_{k+1}}{t_k T_{k+1}}(x_k^{\oplus,\alpha} - x_{k-1}^{\oplus,\alpha}) + \left(1 - \frac{1}{\alpha}\right)\frac{t_{k+1}T_k}{T_{k+1}t_k}(x_k^{\oplus,\alpha} - x_k)$$

where $T_{-1} = \frac{2a-1}{\alpha^2}$. By using Proposition 1, we obtain H-dual.

$$y_{k+1} = y_k^{\oplus,\alpha} + \beta_k'(y_k^{\oplus,\alpha} - y_{k-1}^{\oplus,\alpha}) + \left(1 - \frac{1}{\alpha}\right)\frac{T_{N-k}}{T_{N-k+1}}\beta_{N-k}(y_k^{\oplus,\alpha} - y_k)$$

$$\beta_k' = \frac{T_{N-k-1}t_{N-k}\left(T_{N-k-2} + \left(1 - \frac{1}{\alpha}\right)T_{N-k-1}\right)}{t_{N-k-1}T_{N-k}\left(T_{N-k-1} + \left(1 - \frac{1}{\alpha}\right)T_{N-k}\right)}, \qquad k = 0,\ldots,N-1.$$

Since all equality holds at (96), the above FSFOM achieves the fastest rate among the SFG family under fixed $\alpha$. [7] Now we optimize $a$ to achieve the fastest convergence rate. Combine (93) and the result of Proposition 5 to obtain

$$\min_{v\in\partial F(y_N^\oplus)} \|v\|^2 \le L^2(\alpha+1)^2 \left\|y_N - y_N^{\oplus,\alpha}\right\|^2 \le L^2\frac{2(\alpha+1)^2}{\alpha T_N}\left(F(y_0) - F_\star\right).$$

To achieve the tightest convergence guarantee, we solve the following optimization problem under the fixed $N$.

$$\begin{aligned}
\underset{\alpha}{\text{minimize}} \quad & \frac{2(\alpha+1)^2}{\alpha T_N} \\
\text{subject to.} \quad & \alpha(2T_0 - t_0^2) = T_0, \\
& \alpha(2T_k - t_k^2) = \frac{T_k^2}{T_{k-1}} + t_k^2\left(\frac{1}{T_0} + \sum_{i=0}^{k-2}\frac{1}{T_i}\right) \quad k = 1,\ldots,N.
\end{aligned}$$

Denote the solution of the above optimization problem as $R(\alpha, N)$. Since $R(\alpha, N)$ depends on $a$ and cannot achieve a closed-form solution, we numerically choose $a$ and observe an asymptotic behavior of $R(\alpha, N)$. By choosing $\alpha = 3.8$, asymptotic rate of $R(3.8, N)$ is about $\frac{46}{N^2}$.

# E   Broader Impacts

Our work focuses on the theoretical aspects of convex optimization algorithms. There are no negative social impacts that we anticipate from our theoretical results.

# F   Limitations

Our analysis concerns $L$-smooth convex functions. Although this assumption is standard in optimization theory, many functions that arise in machine learning practice are neither smooth nor convex.

---

[7]In fact, when $a = 1$, the FSFOM becomes FISTA-G [31].

