# OpenReview forum: "Time-Reversed Dissipation Induces Duality Between Minimizing Gradient Norm and Function Value"
_NeurIPS.cc/2023/Conference — NeurIPS 2023 poster_

### Official Review · Reviewer_x8JQ · 2023-06-28

**Soundness:** 4 excellent
**Presentation:** 4 excellent
**Contribution:** 4 excellent
**Rating:** 9
**Confidence:** 5

**Summary:**

This paper introduces novel notion of H-duality, which shows unexpected connection between time-reversed coefficients of methods and optimality w.r.t. gradient norm or function value.

**Strengths:**

Very inspiring work, introduced notion is quite curious. I think that such an idea will be interesting to most of readers. The contribution of this paper is primarily theoretical which could make it "intermediate" for those readers who work in practical field of ML, but I believe that introduced notions can help theorists to develop the theory of optimisation taking into account more important details, which will impact on practice in the end. Especially such abstract works can affect theory the most, if the introduced notion is really helpful and fundamental, which can be confirmed only by the time, but I see it necessary to present such ideas to community to "test" them. Idea seems to be novel: my experience doesn't tell me attempts to introduced such a parallels in methods constructions in terms of duality notion.

**Weaknesses:**

I do not see significant weaknesses. Proof presented in the paper are correct, but assumption are expressful enough to make proofs simple in cases where they are satisfied. In the best case, more detailed development of these concepts should lead to simplification and weakening of assumptions by showing from when do they arise in many cases.

**Questions:**

Notion is based on phrases like "efficiently reduce function values/gradient magnitude", which is not very clear in part "efficiently". Any method which optimises function value optimises gradient norm as well for convex functions, so word "efficiently" requires formalisation. I guess, it is connected with optimality w.r.t. different measures of convergence. If it's possible to develop it better, it would be good.

**Limitations:**

Everything is okay

---

> ### Author Rebuttal · Authors · 2023-08-09
>
> We're grateful for your thorough evaluation and the perceptive observations you've shared. These will certainly assist us in enhancing our paper even more. I'd like to address a few matters in response to your questions.
>
>
> **Q1) Clarification on the phrase "efficiently reduce function values/gradient magnitude"**
>
>
> In this context, the term 'efficiently' primarily pertains to the notion of acceleration or achieving a significant convergence rate. However, it's important to note that Theorem 1 does not depend on the rate of convergence; it holds for every $H$ and $u_i$s. The use of the term 'efficiently minimizing' was intended to accommodate the general interest in acceleration methods.
>
> **Q2) Optimality with respect to Different measures**
>
> You are absolutely correct in associating H-duality with optimality. Indeed, (OGM) can be considered the exact optimal method with respect to function value measurement under an initial distance constraint, while (OGM-G) is widely presumed to be the exact optimal method with respect to gradient norm measurement under an initial function value gap constraint. It's worth noting that these two methods share the same convergence rate factor, $1/\theta_N^2$, as mentioned in lines 103 and 105. Consequently, they both "share" the same convergence rate, which contributes to their efficiency.
>
> Similarly, for the notion of $\mathcal{A}^\star$-optimality, which concerns the optimal method under a certain constrained sense, we have demonstrated that several pairs of methods exhibit interesting properties: a method is $\mathcal{A}^\star$-optimal with respect to function value measures if and only if its H-dual is $\mathcal{A}^\star$-optimal with respect to gradient norm measures. Moreover, the convergence rates of these methods also share the same factor, due to our main theorem (Theorem 1). For more in-depth insights into this topic, please refer to the last paragraph of Section 2.5, where we resolve conjectures regarding $\mathcal{A}^\star$-optimality [(1), (2)].
>
> **References**
>
> (1) D. Kim and J. A. Fessler. Optimizing the efficiency of first-order methods for decreasing
> the gradient of smooth convex functions. Journal of Optimization Theory and Applications,
> 188(1):192–219, 2021
>
> (2) C. Park and E. K. Ryu. Optimal first-order algorithms as a function of inequalities.
> arXiv:2110.11035, 2021.

---

> > ### Comment · Reviewer_x8JQ · 2023-08-19
> >
> > Dear authors, thank you for your work on the final version of your paper! The rebuttal has clarified my questions. I decided to keep my overall rating the same.

---

### Official Review · Reviewer_Fq7n · 2023-07-08

**Soundness:** 3 good
**Presentation:** 3 good
**Contribution:** 3 good
**Rating:** 7
**Confidence:** 2

**Summary:**

This work defines H-duality, which is a one-to-one correspondence between methods that minimize function values and methods that minimize gradient magnitudes, under the assumption that the objective function is convex and $L$-smooth. It is proved in both discrete and continuous time dynamics that, when one method satisfies a convergence condition, the convergence of its H-duality correspondence can also be guaranteed. Furthermore, a new class of methods is derived by applying H-duality to a known class of FSFOMs, and the H-duality convergence theorem can be used to establish the convergence of the new methods.

**Strengths:**

1. To the best of my knowledge, the concept of H-duality proposed in this work is novel, and the theorems which establish the equivalence between convergence conditions of two algorithms and their H-duality are also novel.

2. This work demonstrates that the proposed H-duality can be useful, by using it to establish a symmetry property between pairs of methods, to prove the convergence of them, and to derive new methods.

3. The paper is in general well written, and the sketch of proof helps readers more easily understand the techniques in the proof.

**Weaknesses:**

1. It should be noted that this work only applies to $L$-smooth and convex objective functions, and the value of $L$ is required be known as a prior, since it is used in the methods.

2. If I understand correctly, the number of iterations $N$ is fixed in advance, which could also limit the applicability of this work. (details in question 1)

**Questions:**

1. In practice a method is usually terminated when a certain stopping criterion is reached, and the total number of iterations $N$ is often not known or fixed in advance. However, in this work, $N$ is required to be known and fixed. Could the authors please discuss about this difference? How would it affect the applicability of the conclusions in this work?

2. In section 2.3, it is not obvious to me why $U \geq 0$ is equivalent to (C1), because there is an additional term $- \frac{L}{2} \|\|x_\star -x_0 + \frac{1}{L} \sum_{i=0}^N (u_i - u_{i-1}) \nabla f(x_i) \|\|^2$, which is not in (C1). Could the authors please explain in a high level how to bridge this gap?

**Limitations:**

As far as I can see, there is no potential negative societal impact of this work.

---

> ### Author Rebuttal · Authors · 2023-08-09
>
> We appreciate the thoughtful comments and suggestions you've offered regarding our paper.
>
> **W1) Assumption of $L$-smooth convex function**
>
> Please refer to the response for W1) of reviewer 2PtR.
>
> **W2, Q1)  What about the necessity of $N$ in the H-duality?**
>
> Please refer to the response for Q2) of reviewer  8tCz.
>
> In addition, requiring the total iteration count $N$  of the method to be determined a priori (using
>  in the deterministic stepsizes) is a common property among stochastic gradient methods with convergence rates. Since (OGM-G)'s discovery in 2018, the acceleration mechanism of (OGM-G) has been an active object of study in the field [(1),(2),(3),(4),(5),(6),(7)]. In fact, if a threshold $\epsilon$ is given, we $N$ can be selected to obtain the desired error: For example, (OGM-G) with
>  \begin{align*}
>    N  \geq \left(\frac{4L(f(y_0)-f_\star)}{ \epsilon}\right)^{1/2}
>  \end{align*}
> makes  $\|\nabla f(y_N) \|^2 \leq \mathcal{O}(1/N^2)$.
>
> **Q2) Why $ U\geq 0$ is equivalent to (C1)?**
>
> The only term depends on $(x_0, x_\star)$ in $\mathbf{U}$ is $\frac{L}{2}\left \| x_0-x_{\star} + \sum_{i=0}^{N}\frac{u_i-u_{i-1}}{L}\nabla f(x_i)\right \|^2$. Note that (C1) is saying about $(\forall x_0, x_\star, \nabla f(x_0),\dots, \nabla f(x_N))$. Next, since $x_0,x_{\star}$ can have any value, we can take $x_0-x_{\star}=-\sum_{i=0}^{N}\frac{u_i-u_{i-1}}{L}\nabla f(x_i)$. In this case, $ U_N - u_N ( f(x_N) - f_\star ) = \mathbf{U} $ Thus it gives the fact that (C1) s equivalent to $\Big[ \mathbf{U} \geq 0, \quad \left(\forall\, \nabla f(x_0),\dots,\nabla f(x_N)  \in \mathbb{R}^d \right) \Big]$. You can see the same argument in line 406 of the supplementary material.
>
> This proof technique is mainly adapted from PEP(Performance Estimation Programming) literature, which provides the necessary and sufficient conditions that $(x_i, \nabla f(x_i), f(x_i))$ can be interpolated with a $L$-smooth function. $(\forall x_0, x_\star, \nabla f(x_0),\dots, \nabla f(x_N))$ part can be understood with the same perspective of the PEP analysis.
>
>
>
> **References**
>
> (1) : Z. Allen-Zhu. How to make the gradients small stochastically: Even faster convex and noncon-
> vex SGD. NeurIPS, 2018.
>
> (2) : D. Davis and D. Drusvyatskiy. Stochastic model-based minimization of weakly convex functions.
> SIAM Journal on Optimization, 29(1):207–239, 2019.
>
> (3) : J. Diakonikolas and P. Wang. Potential function-based framework for making the gradients
> small in convex and min-max optimization. SIAM Journal on Optimization, 32(3):1668–1697,
> 2021.
>
> (4) : J. Lee, C. Park, and E. K. Ryu. A geometric structure of acceleration and its role in making
> gradients small fast. NeurIPS, 2021.
>
> (5) : Y. Nesterov, A. Gasnikov, S. Guminov, and P. Dvurechensky. Primal–dual accelerated gradient
> methods with small-dimensional relaxation oracle. Optimization Methods and Software, pages
> 1–38, 2020.
>
> (6) : C. Park and E. K. Ryu. Optimal first-order algorithms as a function of inequalities.
> arXiv:2110.11035, 2021.
>
> (7) : K. Zhou, L. Tian, A. M.-C. So, and J. Cheng. Practical schemes for finding near-stationary
> points of convex finite-sums. AISTATS, 2022.

---

### Official Review · Reviewer_JyDy · 2023-07-23

**Soundness:** 3 good
**Presentation:** 3 good
**Contribution:** 2 fair
**Rating:** 5
**Confidence:** 2

**Summary:**

This paper studies the duality for optimization algorithms. As claimed by the authors, the notion they present is distinct form any previously known duality or symmetry relations. In this work, the authors present a one-to-one correspondence between methods efficiently minimizing function values and methods efficiently minimizing gradient magnitude, which is called H-duality. Using H-duality, a clearer understanding of the symmetry between FGM/OGM and OGM-G and a new class of methods efficiently reducing gradient magnitudes can be obtained. The authors claim that their new composite minimization method achieve better results than state-of-the-art methods in efficiently reducing gradient magnitude in the composite minimization setup.

**Strengths:**

1. This paper presents a novel notion of duality.
2. The theory of H-duality seems to be sound.


**Weaknesses:**

1. The relevance to NeurIPS is a concern. Actually, this paper studies the duality for optimization algorithms, which is an important problem in Optimization and Control. I am not sure whether NeurIPS is the right place.
2. In the Section 4 “New method efficiently reducing gradient mapping norm: (SFG)”, the authors claim that their method (SFG) is faster than FISTA-G by a constant factor of 5.28 while having substantially simpler coefficients. In the paper which presented FISTA-G, two graph were given to describe different methods’ effectiveness on reducing the gradient magnitude. But the result of experiments can’t be found in the paper, which make the comparison between SFG and FISTA-G is lack of persuasiveness. Could you add the result of experiments to describe the comparison between SFG and FISTA-G?


**Questions:**

Please see my question in the "Weakness" section.

**Limitations:**

Not applicable.

---

> ### Author Rebuttal · Authors · 2023-08-09
>
> We value the considerate remarks and recommendations you've provided concerning our paper.
>
> **W1) Regarding the relativity of NeurIPS and this topic**
>
> Indeed, the primary value of our work lies in its theoretical novelty. In our view, H-duality is an unexpected discovery that may enrich our understanding of accelerated optimization methods and has the potential to have a strong practical downstream impact. We kindly ask the reviewers to consider the potential downstream value of our theoretical results.
>
> As for the concern about the exclusive focus on convex optimization, we believe that establishing a unified scheme within the realm of convex optimization is crucial to advancing toward nonconvex optimization. Prior to our paper, first-order methods of reducing the gradient norm were not fully understood. Particularly in nonconvex literature, minimizing the gradient is a frequently used objective. We posit that H-duality could offer valuable insights regarding gradient minimization first-order methods. Therefore, we view the exploration of convex settings as the first step toward unraveling the intricacies of nonconvex methods.
>
>  We also want to note that there are various convex optimization papers in the ML conference venues, including [(1), (2), (3), (4), (5), (6), (7)].
>
> **W2) SFG and FISTA-G**
>
> The term '$5.26$ faster' does not come from the specific experiment, but comes from the exact worst-case analysis. In rigorous terms, the 'worst-case' rate of a given method $\mathcal{A}$ is defined as $\sup_{f,g} (\text{Objective})$ where $f=L$-smooth convex and $g$ is convex. This worst-case analysis provides careful and risk-minimizing upper bounds.
>
> For the FISTA-G [(1)], the worst-case upper bound is $ \min_{v \in \partial (y)} \| v \|^2 \leq \frac{264 L^2}{(N+2)^2} \left(F(y_0)-F_\star\right)$. On the other hand, for (SFG), we show that the worst-case upper bound is $ \min_{v \in \partial (y)} \|v \|^2 \leq \frac{50L^2}{(N+2)(N+3)} \left(F(y_0)-F_\star\right)$  (Theorem 3). This is why we mentioned that (SFG) is 5.26 times faster than (FISTA-G).
>
> **References**
>
> (1) :  J. Lee, C. Park, and E. K. Ryu. A geometric structure of acceleration and its role in making
> gradients small fast. NeurIPS, 2021.
>
> (2) : M. B. Cohen, J. Diakonikolas, and L. Orecchia. On acceleration with noise-corrupted gradients.
> ICML, 2018.
>
> (3) : J. Kim and I. Yang. Unifying nesterov’s accelerated gradient methods for convex and strongly
> convex objective functions: From continuous-time dynamics to discrete-time algorithms.
> arXiv:2301.03576, 2023.
>
> (4) : W. Krichene, A. Bayen, and P. L. Bartlett. Accelerated mirror descent in continuous and discrete
> time. NeurIPS, 2015.
>
> (5) : C. Y. Lin, C. Song, and J. Diakonikolas. Accelerated cyclic coordinate dual averaging with
> extrapolation for composite convex optimization. ICML, 2023.
>
> (6) :  W. Su, S. Boyd, and E. Candes. A differential equation for modeling Nesterov’s accelerated
> gradient method: Theory and insights. NeurIPS, 2014.
>
> (7) :  J. J. Suh, G. Roh, and E. K. Ryu. Continuous-time analysis of accelerated gradient methods via
> conservation laws in dilated coordinate systems. ICML, 2022.

---

> > ### Comment · Reviewer_JyDy · 2023-08-16
> > **Thanks for the response**
> >
> > Thanks for the response! I have no more comments.

---

### Official Review · Reviewer_8tCz · 2023-07-23

**Soundness:** 3 good
**Presentation:** 4 excellent
**Contribution:** 3 good
**Rating:** 8
**Confidence:** 3

**Summary:**

This paper shows an intriguing symmetry in convex optimization between first-order methods that efficiently minimizes the function value and those that efficiently minimizes the squared gradient norm. In particular, they show that if a specific Lyapunov function upper bounds the function value suboptimality for Algorithm A, then a corresponding Lyapunov function would upper bound the squared gradient norm for Algorithm B, which is the "dual" of Algorithm A. This duality relation is further extended to continuous-time dynamics. By using this new tool, the authors propose a new parameterized family of methods for reducing the squared gradient norm, as well as a new composite minimization method that improves the state-of-the-art by a constant factor.

**Strengths:**

- I find the results in this paper interesting and novel. At the surface, accelerated methods for reducing the function value seem quite distinct from those for reducing the gradient norm, and the mechanism for the latter is somehow mysterious. Surprisingly, the authors uncover a one-to-one correspondence between these two families of algorithms. This new concept of H-duality could help us better understand OGM-G type algorithms and design better and practical ones for minimizing the gradient norm.
- Moreover, the paper is very well-written. While the underlying proofs require heavy notations and tedious calculations, the authors manage to make the main theorems quite accessible and clearly explain the high-level ideas.

**Weaknesses:**

Overall, I don't find any major issues with this paper. While this might be too much to ask, I feel that I still don't have a good intuition for H-duality theorem or why it makes sense to reverse the combining coefficients, except that the algebra indicates so. Still, I believe this could be the first step towards some more fundamental understanding.

**Questions:**

- Just a clarifying question, it seems to me that the H-duality is only useful when the algorithm can be analyzed using the particular Lyapunov function in (2). On the other hand, many algorithms, such as the original Nesterov's accelerated method, may not conform to such a Lyapunov function, since they do not rely on the cocoercivity inequaltiy. Is it possible to generalize (2) to a broader family of Lyapunov functions?
- By definition, it is necessary to know the time horizon $N$ to form the H-dual of a given algorithm. Is it possible to establish a duality relation between two anytime algorithms? More concretely, can we use H-duality theorem to obtain an anytime method for reducing the gradient norm?

A minor typo: in (1) it should be $\nabla f(x_i)$ instead of $\nabla f(x_k)$.

**Limitations:**

The authors have addressed the limitations.

---

> ### Author Rebuttal · Authors · 2023-08-09
>
> Thank you for your detailed review and your insightful comments. They certainly will aid us in refining our paper further. I would like to clarify a few points based on your queries.
>
>
> **W1) Why does the H-duality theorem intuitively hold?**
>
>  Let us explain the motivation of the main theorem and energy function, (2) and (3). They come from the Lagrangian of a certain SDP.  Let an FSFOM (1) with matrix H is given. Then we consider the following optimization problem :
> \begin{align*}
> \sup_{f} 	\\,\\, 	& f(x_N) - f_\star\\\\
> \text{sub. to.}\\,\\, & x_0, x_1, ... x_N \text{ are generated by } H \\\\
> 		&[[ x_i, x_{i+1} ]] \leq 0 ,\quad  i = 0,1,... N-1\\\\
> 		&[[ x_i , x_\star]] \leq 0 ,\quad  i = 0,1,... N\\\\
> 		&\| x_0 - x_\star \|^2 \leq R.
> \end{align*}
> The Lagrangian is : $L( f , a_{i,i+1}, b_i , \alpha) = - f(x_N) + f_\star + \sum_{i=0}^{N} a_{i, i+1} [[x_i  , x_{i+1} ]] + \sum_{i=0}^{N}b_i [[ x_i , x_\star ]] +  \alpha \|x_0 - x_\star\|^2$  for nonnegative dual variables $a_{i,i+1} , b_i, \alpha$ and the convex dual problem becomes
> \begin{align*}
>     \sup_{a , b} \inf_{ f(x_i), \nabla f(x_i) } L( f , a_{i,i+1} ,b_i , \alpha).
> \end{align*}
> Here, we can reduce the dual variables into $u_i - u_{i-1}$ and $u_i $. For the other cases,  $\inf_{ f(x_i), \nabla f(x_i) } L$ becomes $-\infty$. Therefore, if $L( f , a_{i,i+1} , b_i, \alpha) = 1/u_N ( U_N  - u_N ( f(x_N) - f_\star ) \geq 0$, we have the function value convergence rate $f(x_N) - f_\star \leq \frac{1}{u_N}\frac{L}{2} \| x_0 - x_\star \|^2$ . This is the motivation of the energy function (2) and the condition (C1). For the gradient norm case, we consider the following optimization problem.
> \begin{align*}
>     \sup_{f} \\,\\,	 	&\| \nabla f(y_N) \|^2 \\\\
> \text{sub. to.} \\,\\, 	&y_0, y_1, ... y_N \text{ are generated by } H' \\\\
> 		&[[ y_i, y_{i+1} ]] \leq 0 ,\quad  i = 0,1,... N-1\\\\
> 		&[[ y_i , y_N ]] \leq 0 ,\quad i = 0,1,... N-1\\\\
> 		&f(y_0) - f_\star \leq D.
> \end{align*}
> Analogously, we obtain a similar Lagrangian function, $L(f , v_i ) = V_N -\frac{1}{2L}\| \nabla f(y_N)\|^2$.
> Therefore,  $V_N - \frac{1}{2L} \| \nabla f(y_N)\|^2 \geq 0 $ , we have the gradient norm convergence rate $\frac{1}{2L} \| \nabla f(y_N) \|^2 \leq v_0 ( f(y_0) - f_\star)$ . This is the motivation of the energy function (3) and the condition (C2).
>
> Surprisingly, when $H' = H^{\text{AT}}$ and $v_i = 1/u_{N-i}$, $V_N-\frac{1}{2L} \| \nabla f(y_N)\|^2$ has the same value with $U_N  - u_N ( f(x_N) - f_\star )$  under the transformation $u_i ( \nabla f(x_i) - \nabla f(x_{i+1}) ) \leftrightarrow \nabla f(y_{N-i}) - \nabla f(y_{N-i-1})$. It leads to our main Theorem 1. For the continuous-time interpretation of H-duality (Theorem 2), we use the continuous-time version of transformation: Please refer to lines 707 and 708 in the supplementary material.
>
> It may be too long for an answer to be 'intuitive', but to the best of our knowledge, we have all shared an understanding. We believe that a simple transformation : $u_i ( \nabla f(x_i) - \nabla f(x_{i+1}) )\leftrightarrow\nabla f(y_{N-i}) - \nabla f(y_{N-i-1})$ will be the principle of H-duality. However, we firmly believe that even more profound insights underlie the duality between gradient minimization and function value minimization. Hence, we used the phrase "open the door" to convey that our discovery of this new form of duality is exact and that we anticipate numerous potential directions for future research and exploration.
>
> **Q1) Is the H-duality useful only when an algorithm can be analyzed using the Lyapunov function in (2)?**
>
> Indeed, the most well-known algorithms, including  Nesterov's FGM [35], OGM [24], G-OGM [26], GD [18], OBL-F$_\flat$ [42], and others, can be analyzed using the framework presented in (2). While some of these algorithms may have alternative analysis approaches, their convergence proofs can still be interpreted within the context of (2).
>
> Furthermore, it is worth noting that altering the form of the Lyapunov function is a possibility. For example, in composite minimization problems (where the problem reduces to a single function setting when \(g=0\)), we have explored a different form of the Lyapunov function, specifically related to SFG (as discussed in our paper), FISTA [10], G-FPGM [25], and FISTA-G [29]. As we mention in our paper, we have left this as a future research direction to explore more generalized forms of the Lyapunov function.
>
> By adopting framework (2) and allowing for different forms of the Lyapunov function, we hope to enable a deeper understanding and analysis of various optimization algorithms, opening up opportunities for further exploration and advancements in the field.
>
> **Q2) What about the necessity of $N$ in the H-duality? Is it feasible without a terminal time?**
>
> We admit that our paper does not offer a dual version without the knowledge of terminal iteration number $N$. However, we suspect it might be possible, albeit not straightforward, as conjectured in  [16, Conjecture 1], which states that there would be no any-time algorithm for minimizing gradient norm having acceleration. If we presume the conjecture, we must discover the dual of Nesterov's FGM in an unknown terminal iteration case, given that Nesterov's FGM accelerates but the conjecture suggests all-time algorithms cannot accelerate gradient norm minimization, which indicates that there would be no convergence rate counterpart.
>
>
> **References**
>
> *Every citation is from our paper, so please check our paper for reference. Apologies for the inconvenience due to the word limit in rebuttal*

---

> > ### Comment · Reviewer_8tCz · 2023-08-14
> > **Thank you for your response**
> >
> > I thank the authors for their detailed response. The observation that the potential function is motivated by the Lagrangian of the PEP is interesting and partly answers my question, though why such a transformation would connect the two Lagrangian functions is still a bit puzzling to me. In any case, I believe this is a nice work and would be of great interest to the community.

---

### Official Review · Reviewer_2PtR · 2023-07-27

**Soundness:** 4 excellent
**Presentation:** 3 good
**Contribution:** 2 fair
**Rating:** 4
**Confidence:** 4

**Summary:**

This paper presents the concept of H-duality, which refers to the duality between optimization algorithms. The notion involves two sets of algorithms: 1) those that efficiently reduce the function value, and 2) those that efficiently reduce the gradient magnitude. The authors create the H matrix, which contains the coefficients of previous gradient evaluations at time step t, for all fixed step size first-order methods. They define the anti-transpose of H, denoted as H^A, as the H-dual of H.

In the main theorem, the authors demonstrate that if there exists an algorithm that reduces the function value using matrix H, then there is another algorithm that minimizes the gradient magnitude using H^A. This duality property is applicable to both discrete time and continuous time first-order optimization algorithms.

Leveraging this duality property for composite objectives, the authors utilize a proximal-based first-order method that minimizes the function value. Subsequently, they propose a novel algorithm that minimizes the gradient value.

**Strengths:**

The introduction of duality between algorithms is a concept presented to the optimization community. This idea not only aids in establishing equivalence between algorithms but also proves valuable in conducting convergence analysis for these algorithms.

**Weaknesses:**

The proposed duality, while introducing a novel perspective, has certain limitations as it does not encompass a broader class of algorithms and loss functions commonly used in machine learning. For instance, it requires the convergence of the last iterate and overlooks convergence based on average iterates. Additionally, it assumes the objective to be convex and L-smooth, which may not always hold in practical Machine Learning scenarios. As a result, the paper primarily pertains to optimization theory rather than providing specific optimization techniques tailored to machine learning applications, especially for stochastic algorithms frequently used in this field.

The paper lacks a clear explanation of how discovering this equivalence between algorithms can facilitate the development of better optimization techniques. For instance, in cases where the objective is convex and the gradient vanishes (\nabla f* = 0), it is evident that achieving convergence based on function values can easily translate to convergence based on gradient magnitudes.

Another aspect that remains unclear in the paper is whether algorithms that are H-dual to each other exhibit similar convergence rates. For instance, in the gradient descent (GD) example, it is observed that the gradient norm can attain a rate of O(1/N^2) under certain conditions. Consequently, it is ambiguous whether two dual algorithms have identical convergence rates or if one may achieve a better rate. In the former case, it becomes necessary to conduct separate convergence analyses for each algorithm to account for potential differences in their rates.

**Questions:**

look at the weaknesses

**Limitations:**

Have been addressed

---

> ### Author Rebuttal · Authors · 2023-08-09
>
> We appreciate your thoughtful comments and suggestions regarding our paper.
>
>  First and foremost, we would like to emphasize that the primary focus of our paper lies in its theoretical contributions rather than constructing a faster method. However, we firmly believe that H-duality opens up the possibility of developing a more efficient gradient-norm-reducing approach: an aspect we will further elaborate on in response to W2.
>
> As for the concern about the exclusive focus on convex optimization, we believe that establishing a unified scheme within the realm of convex optimization is crucial to advancing toward nonconvex optimization. Prior to our paper, first-order methods of reducing the gradient norm were not fully understood. Particularly in nonconvex literature, minimizing the gradient is a frequently used objective. We posit that H-duality could offer valuable insights regarding gradient minimization first-order methods. Therefore, we view the exploration of convex settings as the first step toward unraveling the intricacies of nonconvex methods. Specifically, for a nonconvex $L$-smooth function $f$, the following holds for all $x,y$.
> $$ f(y) \geq f(x) + \langle \nabla f(x) , y-x \rangle + \frac{1}{2L}\| \nabla f(x) -\nabla f(y)\|^2 -\frac{L}{2}\| x-y-\frac{1}{L}\left(\nabla f(x) -\nabla f(y) \right)\|^2
> $$
> Utilizing the above inequality and constructing the parallel energy functions may lead to a similar conclusion.
>
> Moreover, in the realm of minimax optimization, we anticipate that a notion akin to H-duality could potentially extend to minimizing the duality gap or the operator norm. For example, a natural extension for $f(x)-f_\star$ is the duality gap and a natural extension for the $\|\nabla f(x)\|^2$ is the operator norm.
>
> H-duality has provided a clear and insightful understanding of the relationship between one-to-one correspondence gradient norm-reducing methods and function value-reducing methods. Our work stands out as unique because it offers a unified analysis of both function value minimization and gradient value minimization.
>
> **W1)  The assumption of H-duality concerning the last iterate convergence and knowledge of $L$.**
>
>
>  We acknowledge your concerns. In regard to our assumption of the objective being convex and $L$-smooth, we concur that it might not always align with practical scenarios. However, such assumptions are common in numerous papers addressing convex optimization (even those published in machine learning conference venues), including [28,49,11,29,(1),(2),(3)].
>
>  Practically, the smoothness factor $L$ can be estimated using backtracking. In fact, (OBL-F$_b$) and (OBL-G$_b$) (Section 2.3 , Example 2 in our paper ) are originated from backtracking methods.  Regarding average iterates, we think that our methodology could be adapted to use a similar scheme. This is a topic we plan to delve into in future research.
>
> **W3) The clarity of whether algorithms that are H-dual to each other exhibit similar convergence rates.**
>
> Before delving further into the topic, it is important to clarify two common initial conditions: (IDC) and (IFC), which are also written in (P1) and (P2) (lines 48 and 49) in our paper.
>
> (IDC) represents a condition where the initial point's distance from the optimal point is bounded, i.e., $\|x_0 - x_\star\| \leq R$ for some constant $R$.
>
> (IFC), on the other hand, implies that the initial point's function sub-optimality is bounded, i.e., $f(y_0)-f_\star \leq R$ for some constant $R$.
>
> To clarify further, we have established a duality between **[(IDC), function value minimization], and its H-dual method [(IFC), gradient value minimization].**
>
> For the Gradient Descent (GD), the known tightest bound of the gradient norm is $\mathcal{O}(1/N^2)$ under (IDC), but it becomes $\mathcal{O}(1/N)$ under (IFC), aligning with our results.
> Also (GD) reduces the function value with $\mathcal{O}\left(1/N\right)$ rate under (IDC), which matches the complexity bound with the gradient norm convergence rate under (IFC). Additionally, they share the same factor of convergence rate due to our main theorem.
>
> **W2) How does H-duality contribute to the development of optimization methods?**
>
> he analysis of the *concatenation* of the two methods, which provides faster methods for gradient minimization, is as follows: Suppose we perform $2N$ iterations of an algorithm, using method $\mathcal{A}$ for the first $N$ iterations, and then its H-dual method $\mathcal{B}$ for the last $N$ iterations. Assuming $\mathcal{A}$ exhibits an $\mathcal{O}(1/N^\alpha)$ rate under IDC, then $\mathcal{B}$ exhibits the same $\mathcal{O}(1/N^\alpha)$ rate under IFC.
>
> To elaborate, we have: $$
>  \|\nabla f(x_{2N})\|^2 \leq \mathcal{O}((f(x_{N}) -  f_\star) / N^\alpha)  \leq  \mathcal{O}( \|x_0 - x_\star\|^2 / N^{2\alpha}) .
> $$
>
> The first inequality arises from the convergence rate of $\mathcal{B}$, and the second one comes from the convergence rate of $\mathcal{A}$. As a result, the convergence rate becomes faster than the original method that provides only the $\mathcal{O} ( 1 / N^\alpha) $ rate of the gradient norm. For instance, running (FISTA) for the first $N$ steps and (SFG) for the last $N$ steps yields a $\mathcal{O}(1/N^4)$ rate under (IDC).
>
>
> **References**
>
> (1) : W. Krichene, A. Bayen, and P. L. Bartlett. Accelerated mirror descent in continuous and discrete
> time. NeurIPS, 2015.
>
> (2) : C. Y. Lin, C. Song, and J. Diakonikolas. Accelerated cyclic coordinate dual averaging with
> extrapolation for composite convex optimization. ICML, 2023.
>
> (3) : M. B. Cohen, J. Diakonikolas, and L. Orecchia. On acceleration with noise-corrupted gradients.
> ICML, 2018.
>
> *The other citations are from our paper, so please check our paper for reference. Apologies for the inconvenience due to the word limit in rebuttal*

---

> > ### Comment · Reviewer_2PtR · 2023-08-14
> >
> > I appreciate the authors' thoughtful replies to my questions. However, I maintain my view that this work aligns more closely with the literature on optimization theory rather than being a suitable fit for NeurIPS.

---

### Author Rebuttal · Authors · 2023-08-09


# Common Response

We would like to express our sincere appreciation to the reviewers for their time and constructive criticism.

Several reviewers (Reviewers 8tCz, Fq7n, and x8JQ) strongly appreciated the theoretical novelty and value of our paper, acknowledging the innovative aspects of H-duality. (With reviewer x8JQ providing a rating of 9!)


On the other hand, Reviewers 2PtR and JyDy had concerns about the immediate practical applicability of our work, with Reviewer 2PtR pointing out that the convexity and $L$-smoothness assumptions limit the applicability of the analysis in practical machine learning. Indeed, the primary value of our work lies in its theoretical novelty. In our view, however, H-duality is an unexpected discovery that may enrich our understanding of accelerated optimization methods and has the potential to have a strong practical downstream impact. We kindly ask the reviewers to consider the potential downstream value of our theoretical results.

---

### Decision · Program_Chairs · 2023-09-21

**Decision:**

Accept (poster)

**Comment:**

Overall the reviewers viewed the paper positively. There was some discussion over whether the work fits in the scope of NeurIPS, but ultimately it does apply to problems that have historically arised in ML.